# Biogeochemical protocols and diagnostics for the CMIP6 Ocean Model Intercomparison Project (OMIP)

James C. Orr[1], Raymond G. Najjar[2], Olivier Aumont[3], Laurent Bopp[1], John L. Bullister[4], Gokhan Danabasoglu[5], Scott C. Doney[6], John P. Dunne[7], Jean-Claude Dutay[1], Heather Graven[8], Stephen M. Griffies[7], Jasmin G. John[7], Fortunat Joos[9], Ingeborg Levin[10], Keith Lindsay[5], Richard J. Matear[11], Galen A. McKinley[12], Anne Mouchet[13,14], Andreas Oschlies[15], Anastasia Romanou[16], Reiner Schlitzer[17], Alessandro Tagliabue[18], Toste Tanhua[15], and Andrew Yool[19]

[1]LSCE/IPSL, Laboratoire des Sciences du Climat et de l'Environnement, CEA-CNRS-UVSQ, Gif-sur-Yvette, France
[2]Dept. of Meteorology, Pennsylvania State University, University Park, Pennsylvania, USA
[3]Laboratoire d'Océanographie et de Climatologie: Expérimentation et Approches Numériques, IPSL, Paris, France
[4]Pacific Marine Environmental Laboratory, NOAA, Seattle, Washington, USA
[5]National Center for Atmospheric Research, Boulder, Colorado, USA
[6]Marine Chemistry & Geochemistry, Woods Hole Oceanographic Institution, USA
[7]NOAA Geophysical Fluid Dynamics Laboratory, Princeton, New Jersey, USA
[8]Dept. of Physics, Imperial College, London, UK
[9]Climate and Environmental Physics, Physics Inst. & Oeschger Center for Climate Change Res., Univ. of Bern, Switzerland
[10]Institut fuer Umweltphysik, Universitaet Heidelberg, Heidelberg, Germany
[11]CSIRO Oceans and Atmosphere, Hobart, Tasmania 7000, Australia
[12]Atmospheric and Oceanic Sciences, University of Wisconsin-Madison, Wisconsin, USA
[13]Max Planck Institute for Meteorology, Hamburg, Germany
[14]Astrophysics, Geophysics and Oceanography Department, University of Liege, Liege, Belgium
[15]GEOMAR Helmholtz Centre for Ocean Research Kiel, Kiel, Germany
[16]Columbia University and NASA-Goddard Institute for Space Studies, New York, NY, USA
[17]Alfred Wegener Institute, Bremerhaven, Germany
[18]Earth, Ocean and Ecological Sciences, University of Liverpool, Liverpool, UK
[19]National Oceanographic Centre, Southampton, UK

*Correspondence to:* James Orr (james.orr@lsce.ipsl.fr)

**Abstract.** The Ocean Model Intercomparison Project (OMIP) focuses on the physics and biogeochemistry of the ocean component of Earth System Models participating in the sixth phase of the Coupled Model Intercomparison Project (CMIP6). OMIP aims to provide standard protocols and diagnostics for ocean models, while offering a forum to promote their common assessment and improvement. It also offers to compare solutions of the same ocean models when forced with reanalysis data (OMIP simulations) versus when integrated within fully coupled Earth System Models (CMIP6). Here we detail simulation protocols and diagnostics for OMIP's biogeochemical and inert chemical tracers. These passive-tracer simulations will be coupled to ocean circulation models, initialized with observational data or output from a model spin up, and forced by repeating the 1948–2009 surface fluxes of heat, fresh water, and momentum. These so-called OMIP-BGC simulations include three inert chemical tracers (CFC-11, CFC-12, $SF_6$) and biogeochemical tracers (e.g. dissolved inorganic carbon, carbon isotopes, alkalinity, nutrients, and oxygen). Modelers will use their preferred prognostic BGC model but should follow common guidelines for gas exchange and carbonate chemistry. Simulations include both natural and total carbon tracers. The required forced simu-

lation *(omip1)* will be initialized with gridded observational climatologies. An optional forced simulation *(omip1-spunup)* will be initialized instead with BGC fields from a long model spin up, preferably for 2000 years or more and forced by repeating the same 62-year meteorological forcing. That optional run will also include abiotic tracers of total dissolved inorganic carbon and radiocarbon, $C_\text{T}^\text{abio}$ and $^{14}C_\text{T}^\text{abio}$, to assess deep-ocean ventilation and distinguish the role of physics vs. biology. These simula-

tions will be forced by observed atmospheric histories of the three inert gases and $CO_2$ as well as carbon isotope ratios of $CO_2$. OMIP-BGC simulation protocols are founded on those from previous phases of the Ocean Carbon-Cycle Model Intercomparison Project. They have been merged and updated to reflect improvements concerning gas exchange, carbonate chemistry, and new data for initial conditions and atmospheric gas histories. Code is provided to facilitate their implementation.

## 1   Introduction

Centralized efforts to compare numerical models with one another and with data commonly lead to model improvements and accelerated development. The fundamental need for model comparison is fully embraced in Phase 6 of the Coupled Model Intercomparison Project (CMIP6), an initiative that aims to compare Earth System Models (ESMs) and their climate-model counterparts as well as their individual components. CMIP6 emphasizes common forcing and diagnostics through 21 dedicated Model Intercomparison Projects (MIPs) under a common umbrella (Eyring et al., 2016). One of these MIPs is the Ocean Model

Intercomparison Project (OMIP). OMIP focuses on comparison of global ocean models that couple circulation, sea-ice, and optional biogeochemistry, which together make up the ocean components of the ESMs used within CMIP6. OMIP works along two coordinated branches focused on ocean circulation and sea ice (OMIP-Physics) and on biogeochemistry (OMIP-BGC). The former is described in a companion paper in this same issue (Griffies et al., 2016), while the latter is described here.

Groups that participate in OMIP will use different ocean biogeochemical models coupled to different ocean general circu-

lation models (OGCMs). The skill of the latter in simulating ocean circulation affects the ability of the former to simulate ocean biogeochemistry. Thus previous efforts to compare global-scale, ocean biogeochemical models have also strived to evaluate simulated patterns of ocean circulation. For instance, the Ocean Carbon-Cycle Model Intercomparison Project (OCMIP) included efforts to assess simulated circulation along with simulated biogeochemistry. OCMIP began in 1995 as an effort to identify the principal differences between existing ocean carbon-cycle models. Its first phase (OCMIP1) included four models

and focused on natural and anthropogenic components of oceanic carbon and radiocarbon (Sarmiento et al., 2000; Orr et al., 2001). OCMIP2 was launched in 1998, comparing 12 models with common biogeochemistry, and evaluating them with physical and inert chemical tracers (Doney et al., 2004; Dutay et al., 2002; Matsumoto et al., 2004; Dutay et al., 2004; Orr et al., 2005; Najjar et al., 2007). In 2002, OCMIP3 turned its attention to evaluating simulated interannual variability in forced ocean biogeochemical models (e.g. Rodgers et al., 2004; Raynaud et al., 2006). More recently, OCMIP has focused on assessing

ocean biogeochemistry simulated by ESMs (e.g. Bopp et al., 2013).

OCMIP2 evaluated simulated circulation using the physically active tracers, temperature $T$ and salinity $S$ (Doney et al., 2004), but also with passive tracers, i.e. those having no effect on ocean circulation. For example, OCMIP2 used two anthropogenic transient tracers, CFC-11 and CFC-12 (Dutay et al., 2002). Although these are reactive gases in the atmosphere

that participate in the destruction of ozone, they remain inert once absorbed by the ocean. From an oceanographic perspective, they may be thought of as dye tracers given their inert nature and purely anthropogenic origin, increasing only since the 1930s (Fig. 1). Furthermore, precise measurements of CFC-11 and CFC-12 have been made throughout the world ocean, e.g. having been collected extensively during WOCE (World Ocean Circulation Experiment) and CLIVAR (Climate and Ocean—Variability, Predictability and Change). Hence they are well suited for model evaluation and are particularly powerful when used together to deduce decadal ventilation times of subsurface waters. Yet their combination is less useful to assess more recent ventilation, because their atmospheric concentrations have peaked and declined, since 1990 for CFC-11 and since 2000 for CFC-12, as a result of the Montreal Protocol. To fill this recent gap, oceanographers now also measure $SF_6$, another anthropogenic, inert chemical tracer whose atmospheric concentration has increased nearly linearly since the 1980s. Combining $SF_6$ with either CFC-11 or CFC-12 is optimal for assessing even the most recent ventilation time scales. Together these inert chemical tracers can be used to assess transient time distributions (TTDs). TTDs are used to infer distributions of other passive tracer distributions, such as anthropogenic carbon (e.g. Waugh et al., 2003), which cannot be measured directly.

To help assess simulated circulation fields, OCMIP also included another passive tracer, radiocarbon, focusing on both its natural and anthropogenic components. Radiocarbon ($^{14}$C) is produced naturally by cosmogenic radiation in the atmosphere, invades the ocean via air-sea gas exchange, and is mixed into the deep sea. Its natural component is useful because its horizontal and vertical gradients in the deep ocean result not only from ocean transport but also from radioactive decay (half-life of 5700 years), leaving a time signature for the slow ventilation of the deep ocean (roughly 100 to 1000 yr depending on location). Hence natural $^{14}$C provides rate information throughout the deep ocean, unlike $T$ and $S$. For example, the ventilation age of the deep North Pacific is about 1000 years, based on the depletion of its $^{14}$C/C ratio (-260‰ in terms of $\Delta^{14}$C, i.e. the fractionation-corrected ratio relative to that of the preindustrial atmosphere) when compared with that of source waters from the surface Southern Ocean (-160‰) (Toggweiler et al., 1989a). In the same vein, ventilation times of North Atlantic Deep Water and Antarctic Bottom Water have been deduced from $^{14}$C in combination with another biogeochemical tracer $PO_4^*$ ("phosphate star") (Broecker et al., 1998), facilitated by their strong regional contrasts. The natural component of radiocarbon complements the three inert chemical tracers mentioned above, which are used to assess more recently ventilated waters nearer to the surface. Yet the natural component is only half of the story.

During the industrial era, atmospheric $\Delta^{14}$C declined due to emissions of fossil $CO_2$ (Suess effect) until the 1950s when that signal was overwhelmed by the much larger spike from atmospheric nuclear weapons tests (Fig. 2). Since the latter dominates, the total change from both anthropogenic effects is often referred to as bomb radiocarbon. As an anthropogenic transient tracer, bomb radiocarbon complements CFC-11, CFC-12, and $SF_6$ because of its different atmospheric history and much longer air-sea equilibration time (Broecker and Peng, 1974). Observations of bomb radiocarbon have been used to constrain the global mean gas transfer velocity (Broecker and Peng, 1982; Sweeney et al., 2007); however, in recent decades, ocean radiocarbon changes have become more sensitive to interior transport and mixing, making it behave more like anthropogenic $CO_2$ (Graven et al., 2012). Hence it is particularly relevant to use radiocarbon observations to evaluate ocean carbon-cycle models that aim to assess uptake of anthropogenic carbon as done during OCMIP (e.g. Orr et al., 2001).

Information from the stable carbon isotope $^{13}$C also helps to constrain the anthropogenic perturbation in dissolved inorganic carbon by exploiting the Suess effect (Quay et al., 1992, 2003). Driven by the release of anthropogenic $CO_2$ produced from agriculture, deforestation, and fossil-fuel combustion, the Suess effect has resulted in a continuing reduction of the $^{13}$C/$^{12}$C ratio relative to that of the preindustrial atmosphere-ocean system. That ratio is reported relative to a standard as $\delta^{13}$C, which

is not corrected for fractionation, unlike $\Delta^{14}$C. Fractionation occurs during gas exchange and photosynthesis, and $\delta^{13}$C is also sensitive to respiration of organic material and ocean mixing. Ocean $\delta^{13}$C observations have been used to test marine ecosystem models, including processes such as phytoplankton growth rate, iron limitation, and grazing (Schmittner et al., 2013; Tagliabue and Bopp, 2008) and may also provide insight into climate-related ecosystem changes. Past changes in $\delta^{13}$C recorded in ice cores and marine sediments are likewise useful to evaluate models (Schmitt et al., 2012; Oliver et al., 2010).

Besides the aforementioned tracers to evaluate modeled circulation fields, OMIP-BGC also includes other passive tracers to compare simulated ocean biogeochemistry with data and among models, e.g. in terms of mean states, trends, and variability. Whereas all OCMIP2 groups used a common biogeochemical model (Najjar and Orr, 1998, 1999; Najjar et al., 2007), essentially testing its sensitivity to different circulation fields, OMIP will not adopt the same approach. Rather, OMIP focuses on evaluating and comparing preselected "combined" ocean models (circulation-ice-biogeochemistry) largely defined already by

individual groups planning to participate in CMIP6. Those combined ocean models will be evaluated when forced by reanalysis data as well as when coupled within the CMIP6 ESMs.

OMIP-BGC model groups will use common physical forcing for ocean-only models and common formulations for carbonate chemistry, gas exchange, gas solubilities, and Schmidt numbers. Biogeochemical models will be coupled to the ocean-ice physical models, online (active and passive tracers will be modeled simultaneously), and they will be forced with the same

atmospheric gas histories. Yet beyond those commonalities, model groups are free to choose their preferred ocean model configuration. For instance, groups may choose whether or not to include direct coupling between simulated chlorophyll and ocean dynamics. When coupled, chlorophyll is not a typical passive tracer; it is active in the sense that it affects ocean circulation. Likewise, OMIP groups are free to use their preferred boundary conditions for the different sources of nutrients and micronutrients to the ocean via atmospheric deposition, sediment mobilization, and hydrothermal sources (e.g. for Fe) as

well as lateral input of carbon from river and groundwater discharge. Biogeochemical models with riverine delivery of carbon and nutrients to the ocean usually include sediment deposition as well as loss of carbon from rivers back to the atmosphere through the air-sea exchange. Each group is free to use their preferred approach as long as mass is approximately conserved. Groups are requested to provide global integrals of these boundary conditions and to document their approach, preferably in a peer-reviewed publication.

OMIP-BGC aims to provide the technical foundation to assess trends, variability, and related uncertainties in ocean carbon and related biogeochemical variables since the onset of the industrial era and into the future. That foundation includes (1) the OMIP-BGC protocols for groups that will include inert chemical tracers and biogeochemistry in OMIP's two forced global ocean model simulations, which couple circulation, sea-ice, and biogeochemistry, and (2) the complete list of ocean biogeochemical diagnostics for OMIP, but also for CMIP6 (Eyring et al., 2016) and any ocean-related MIPs under its umbrella, e.g.

C4MIP (Jones et al., 2016).

Simulated results from OMIP-BGC will be exploited to contribute to OMIP's effort to study basic CMIP6 science questions on the origins and consequences of systematic model biases. In particular, OMIP-BGC offers a forum for ocean biogeochemical modelers and a technical framework by which they will assess and improve biases of simulated tracer and biogeochemical components of CMIP6's ESMs. OMIP-BGC will contribute to the World Climate Research Programme's (WCRP) Grand Challenges by providing fundamental information needed to improve near-term climate prediction and carbon feedbacks in the climate system. Assessments will focus on current and future changes in ocean carbon uptake and storage, acidification, deoxygenation, and changes in marine productivity.

Novel analyses are expected from OMIP, in part because of recent improvements in the physical and biogeochemical components. For example, some of the physical models will have sufficient resolution to partially resolve mesoscale eddies. When coupled to biogeochemical models, that combination should allow OMIP to provide a first assessment of how air-sea $CO_2$ fluxes and related biogeochemical variables are affected by the ocean's intrinsic variability (also known as internal, chaotic, or unforced variability). Previous studies of the ocean's internal variability have focused only on physical variables (Penduff et al., 2011). Other studies have assessed internal variablility of ocean biogeochemistry, but they account only for the component associated with turbulence in the atmosphere, i.e., they use a coarse-resolution ocean model coupled within an Earth System Model framework (Lovenduski et al., 2016). Whether internal variablity from the ocean works to enhance or reduce that from the atmosphere will depend on the variable studied, the region, and the model. OMIP aims to provide a new insight on the ocean's contribution to internal variability while also quantifying the relative importance of the contribution of internal variability to overall uncertainty of model projections.

## 2    Protocols

As described by Griffies et al. (2016), the OMIP-Physics simulations consist of forcing physical model systems (an ocean general circulation model coupled to a sea-ice model) with the interannually varying atmospheric data reanalysis known as the Coordinated Ocean-ice Reference Experiments (CORE-II) available over 1948 to 2009 (Large and Yeager, 2009). For OMIP, that 62-yr forcing will be repeated five times to make simulations of 310 yr. OMIP-BGC participants will make these simulations by coupling their prognostic models of ocean biogeochemistry, online, to their physical model systems. These OMIP-BGC simulations will be forced by observed records of atmospheric $CO_2$ and other gases during the 310-yr period, defined as equivalent to calendar years 1700 to 2009. One 310-yr OMIP simulation *(omip1)*, with models initialized by data, is required (Tier 1) for all OMIP modeling groups; another 310-yr simulation *(omip1-spunup)*, with models initialized from a previous long spin-up simulation, is only for OMIP-BGC groups. Although optional, the *omip1-spunup* simulation is strongly encouraged (Tier 2) to minimize drift, assess deep-ocean ventilation, and separate physical vs. biological components of ocean carbon. Details of these simulations are provided below.

The two forced ocean model simulations, *omip1* and *omip1-spunup* differ from but are connected to the CMIP6 DECK and historical simulations. The only differences are the intialization and the forcing. In *omip1*, the ocean model is initialized with observations and forced by reanalysis data; in *historical* the ocean model is coupled within an Earth System Model Framework

after some type of spin up. Likewise, the early portion of the *omip1-spunup* forced simulation is comparable to the CMIP6 DECK *piControl* coupled simulation. The complementarity of approaches will lead to a more thorough model evaluation.

When modeling chemical and biogeochemical tracers, it is recommended that OMIP groups use the same formulations for gas exchange and carbonate chemistry as outlined below. Little effort would be needed to modify code that is already consistent with previous phases of OCMIP. For gas exchange, model groups only need to change the value of the gas transfer coefficient, the formulations and coefficients for Schmidt numbers, and the atmospheric gas histories. For carbonate chemistry, groups should strive to use the constants recommended for best practices (Dickson et al., 2007) on the total pH scale and to avoid common modeling assumptions that lead to significant biases, notably an oversimplified alkalinity equation (Orr and Epitalon, 2015). Fortran 95 code to make these calculations will be made available to OMIP-BGC participants.

## 2.1   Passive Tracers

### 2.1.1   Inert chemistry

The inert chemistry component of OMIP includes online simulation of CFC-11, CFC-12, and $SF_6$. While CFC-12 is required (priority 1), CFC-11 and $SF_6$ are encouraged (priority 2). About the same amount of observational data in the global ocean exists for both CFC-11 and CFC-12, starting with early field programs in the 1980s. But CFC-12 has a longer atmospheric history, with its production starting a decade earlier ($\sim$1936) and a slower decline starting a decade later due to its longer atmospheric lifetime (112 vs. 52 yr) relative to CFC-11 (Rigby et al., 2013). In contrast, $SF_6$ has continued to increase rapidly in recent decades. That increase will continue for many years despite ongoing efforts to restrict production and release of this potent greenhouse gas, because $SF_6$'s atmospheric lifetime is perhaps 3000 yr (Montzka et al., 2003). Using pairs of these tracers offers a powerful means to constrain ventilation ages; if model groups are only able to model two of these tracers, the ideal combination is CFC-12 and $SF_6$.

Simulation protocols are based on the OCMIP2 design document (Najjar and Orr, 1998) and its ensuing CFC protocol (Orr et al., 1999a) and model comparison (Dutay et al., 2002). These inert passive tracers are computed online along with the active tracers (i.e. temperature and salinity in the physical simulation); they are independent of the biogeochemical model. OMIP models will be forced to follow historical atmospheric concentrations of CFC-11, CFC-12, and $SF_6$, accounting for gas exchange and their different solubilities and Schmidt numbers. The same passive tracers should be included in the forced OMIP simulations and in the coupled CMIP6 historical simulations. Both types of simulations will be analyzed within the framework of OMIP. These inert chemistry tracers are complementary to the ideal age tracer that is included in the OMIP-Physics protocols (Griffies et al., 2016).

### 2.1.2   Biogeochemistry

For the other passive tracers, referred to as biogeochemistry, the OMIP-BGC protocols build on those developed for OCMIP. These include the OCMIP2 abiotic and biotic protocols (Najjar and Orr, 1998, 1999; Orr et al., 1999b) and the OCMIP3 protocols for interannually forced simulations (Aumont et al., 2004). Each model group will implement the OMIP protocol

in their own prognostic ocean biogeochemical model as in OCMIP3, unlike the common-model approach of OCMIP2. Each OMIP biogeochemical model will be coupled online to an ocean general circulation model, forced by the CORE-II atmospheric state. Geochemical boundary conditions for the atmosphere include an imposed constant atmospheric concentration of $O_2$ (mole fraction $x_{O_2}$ of 0.20946) but a variable atmospheric $CO_2$ that follows observations (Meinshausen et al., 2016).

In addition, OMIP-BGC simulations should include a natural carbon tracer that sees a constant atmospheric mole fraction of $CO_2$ in dry air ($x_{CO_2}$) fixed at the 1 January 1850 value (284.65 ppm), the CMIP6 preindustrial reference. This can be done either in an independent simulation with identical initial conditions and forcing, except for atmospheric $x_{CO_2}$, or in the same simulation by adding one or more new tracers to the biogeochemical model, referred to here as a *dual-$C_T$* simulation. For this *dual* simulation, OMIP modelers would need to add a second dissolved inorganic carbon tracer ($C_T^{nat}$), e.g. as in Yool

et al. (2010). In OMIP, this added tracer will isolate natural $CO_2$ and keep track of model drift. Such *doubling* may also be necessary for other biogeochemical model tracers if they are directly affected by the $CO_2$ increase. For instance, expansion of the PISCES model (Aumont and Bopp, 2006) to a *dual-$C_T$* implementation resulted in doubling not only $C_T$ but also its transported $CaCO_3$ tracer, which in turn affects total alkalinity $A_T$ (Dufour et al., 2013). These natural tracers are referred to as $C_T^{nat}$, $CaCO_3^{nat}$, and $A_T^{nat}$. Calculated variables affected by $CO_2$ should also be doubled, including pH, $pCO_2$, the air-sea

$CO_2$ flux, and carbonate ion concentration. If biology depends on $CO_2$, additional tracers such as nutrients and $O_2$ would also need to be doubled, making the doubling strategy less appealing. That strategy may also be more complex in some ESMs, e.g. if $A_T$ changes abiotically due to warming-related changes in weathering and river runoff.

### 2.1.3   Abiotic carbon and radiocarbon

In the *omip1-spunup* simulation (as well as in its previously run spin up) OMIP-BGC groups will also include two abiotic

tracers to simulate total dissolved inorganic carbon $C_T^{abio}$ and corresponding radiocarbon $^{14}C_T^{abio}$. These abiotic tracers do not depend on any biotic tracers. They should be included in addition to the biotic carbon tracers mentioned above ($C_T$ and $C_T^{nat}$). The ratio of the two abiotic tracers will be used to evaluate and compare models in terms of deep-ocean ventilation ages (natural radiocarbon) and near-surface anthropogenic invasion of bomb radiocarbon. In addition, $C_T^{abio}$ will be compared to $C_T$ to distinguish physical from biogeochemical effects on total carbon. For simplicity, simulations will be made abiotically

following OCMIP2 protocols (Orr et al., 1999b). We recommend that participating groups add these two independent tracers to their biogeochemical model to simulate them simultaneously, thus promoting internal consistency while reducing costs.

    In OMIP, we will use this two-tracer approach rather than the simpler approach of modeling only the $^{14}C/C$ ratio directly (Toggweiler et al., 1989a, b). That simpler approach would be a better choice if our focus were only on comparing simulated and field-based estimates of the ocean's bomb-$^{14}C$ inventory, both of which are biased low (Naegler, 2009; Mouchet, 2013).

The simpler modeling approach underestimates the inventory, because it assumes a constant air-sea $CO_2$ disequilibrium during the industrial era; likewise, field reconstructions of the ocean's bomb-$^{14}C$ inventory (Key et al., 2004; Peacock, 2004; Sweeney et al., 2007) are biased low because they assume that ocean $C_T$ is unaffected by the anthropogenic perturbation. Yet in terms of oceanic $\Delta^{14}C$, the simple and the two-tracer approaches yield similar results (Mouchet, 2013), because the effect of increasing

$C_T$ on oceanic $\Delta^{14}C$ is negligible (Naegler, 2009). We also choose the two-tracer approach to take advantage of its $C_T^{abio}$ tracer to help distinguish physical from biological contributions to $C_T$.

To model $^{14}C$, OMIP neglects effects due to fractionation (i.e. from biology and gas exchange). Hence model results will be directly comparable to measurements reported as $\Delta^{14}C$, a transformation of the $^{14}C/C$ ratio designed to correct for frac-
tionation (Toggweiler et al., 1989a). Thus biases associated with our abiotic approach may generally be neglected. For natural $^{14}C$, Bacastow and Maier-Reimer (1990) found essentially identical results for simulations that accounted for biological frac-
tionation vs. those that did not, as long as the atmospheric $CO_2$ boundary conditions were identical. For bomb $^{14}C$, which also includes the Suess effect, neglecting biological fractionation results in small biases (Joos et al., 1997).

Hence for the *omip1-spunup* simulation, OMIP-BGC groups will simulate four flavors of dissolved inorganic carbon: biotic natural ($C_T^{nat}$), biotic total ($C_T$), abiotic total ($C_T^{abio}$), and abiotic radiocarbon ($^{14}C_T^{abio}$). Conversely for the *omip1* simulation, groups will simulate only the first two flavors, $C_T^{nat}$ and $C_T$. These tracers may be simulated simultaneously or in separate simulations, although we recommend the former.

### 2.1.4 Carbon-13

Groups that have experience modeling $^{13}C$ in their biogeochemical model are requested to include it as a tracer in the OMIP-
BGC simulations. Groups without experience should avoid adding it. It is not required to simulate $^{13}C$ in order to participate in OMIP. Modeling groups that will simulate ocean $^{13}C$ are requested to report net air-sea fluxes of $^{13}CO_2$ and concentrations of total dissolved inorganic carbon-13 ($^{13}C_T$) for the *omip1-spunup* simulation. In Sect. 2.5 we recommend how isotopic fractionation during gas exchange should be modeled. Carbon-13 is typically included in ocean models as a biotic variable influenced by fractionation effects during photosynthesis that depend on growth rate and phytoplankton type; some models
also include fractionation during calcium carbonate formation (e.g., Tagliabue and Bopp, 2008). Modeling groups should incorporate ecosystem fractionation specific to their ecosystem model formulation. We do not request modeling groups to report variables related to $^{13}C$ in phytoplankton or other organic carbon pools, only $^{13}C_T$ and net air-sea $^{13}CO_2$ fluxes.

### 2.2 Duration and initialization

As described by (Griffies et al., 2016), the physical components of the models are to be forced over 310 years, i.e. over five
repeated forcing cycles of the 62-year CORE-II forcing (1948–2009). The biogeochemistry should be included, along with the physical system, during the full 310 years (1700–2009) and the inert chemistry only during the last 74 years (1936–2009). The biogeochemical simulations will be initialized on calendar date 01 January 1700, at the start of the first CORE-II forcing cycle. The inert anthropogenic chemical tracers (CFC-11, CFC-12, $SF_6$) will be initialized to zero on 01 January 1936, during the fourth CORE-II forcing cycle at model date 01 January 0237.

For the *omip1* simulation, biogeochemical tracers will be initialized generally with observational climatologies. Fields from the 2013 World Ocean Atlas (WOA2013) will be used to initialize model fields of oxygen (Garcia et al., 2014a) as well as nitrate, total dissolved inorganic phosphorus, and total dissolved inorganic silicon (Garcia et al., 2014b). The latter two nutrients are often referred to simply as phosphate and silicate, but other inorganic P and Si species also contribute substantially to each

total concentration (Fig. 3). Indeed it is the total dissolved concentrations ($P_T$ and $Si_T$) that are both modeled and measured. OMIP will provide all these initial biogeochemical fields by merging WOA2013's means for January, available down to 500 m (for nitrate, phosphate and silicate), and down to 1500m for oxygen, with its annual mean fields below.

Model fields for $A_T$ and preindustrial $C_T$ will be initialized with gridded data from version 2 of the Global Ocean Data
Analysis Project (GLODAPv2) from Lauvset et al. (2016), based on discrete measurements during WOCE and CLIVAR (Olsen et al., 2016). For greater consistency with GLODAPv1, OMIP-BGC model groups will use the $C_T$ and $A_T$ fields from GLODAPv2's first period (1986–1999, the WOCE era). To initialize modeled dissolved organic carbon (DOC), OMIP provides fields from the adjoint model from Schlitzer (Hansell et al., 2009). For dissolved iron (Fe), OMIP simulations will not be initialized from observations because a full-depth, global 3-D data climatology is unavailable due to lack of data coverage,
particularly in the deep ocean. Hence for initial Fe fields, OMIP provides the median model result from the Iron Model Intercomparison Project (FeMIP, Tagliabue et al., 2016). Yet that initialization field may not be well suited for all Fe models, which differ greatly. Although OMIP provides initialization fields for Fe and DOC, their actual initialization is left to the discretion of each modeling group. In a previous comparison (Kwiatkowski et al., 2014), groups did not initialize modeled Fe with a common field nor approach because the complexity of the Fe cycle differed greatly among models. Likewise, there
was no common approach to initialize DOC because biogeochemical models vary greatly in the way they represent its lability. Initialization of other tracers is less critical, e.g. phytoplankton biomass is restricted to the top 200 m and equilibrates rapidly as do other biological tracers.

The *omip1* simulation is relatively short and is thus manageable by all groups, but many of its tracers will have large drifts because model initial states will be far from their equilibrium states. These drifts complicate assessment of model performance
based on model-data agreement (Séférian et al., 2015). Hence a complementary simulation, *omip1-spunup*, is proposed, where biogeochemical tracers are initialized instead with a near-equilibrium state. Model groups may generate this spun-up initial state by any means at their disposal. The classic approach would be to spin up the model. That could be done either online, repeating many times the same physical atmospheric forcing (CORE-II), or offline, repeatedly cycling the physical transport fields from a circulation model forced by a single loop of the CORE-II forcing.

If the spin-up simulation is made online, groups should reset their model's physical fields at the end of every fifth cycle of CORE-II forcing to their state at the beginning of the previous third cycle. Thus groups will avoid long-term drift in the model's physical fields, and the latter will not diverge greatly from those of the *ocmip1* simulation but be allowed to evolve freely over a period roughly equivalent to that of the transient CO2 increase (last three forcing cycles). Conversely, biogeochemical fields should not be reset. The end of the spin-up simulation will be reached only after many repetitions of the 5 consecutive forcing
cycles with the online model. That final state (i.e. the physical and biogeochemical fields from the end of the final fifth cycle) will be used to initialize the *ocmip1-spunup* simulation. Offline spin-up simulations should made in a consistent fashion. That is, groups should first integrate their circulation model over two cycles of forcing and then use the physical circulation fields generated during the third forcing cycle to subsequently drive their offline biogeochemical model, typically until they reach the criteria described below.

If possible, the spin-up should be run until it reaches the biogeochemical equilibrium criteria adopted for OCMIP2. These criteria state that the globally integrated, biotic and abiotic air-sea $CO_2$ fluxes ($F_{CO_2}$ and $F_{CO_2}^{abio}$) should each drift by less than 0.01 Pg C yr$^{-1}$ (Najjar and Orr, 1999; Orr et al., 1999b) and that abiotic $^{14}C_T$ should be stabilized to the point that 98% of the ocean volume has a drift of less than 0.001 ‰ yr$^{-1}$ (Aumont et al., 1998). The latter is equivalent to a drift of about 10 yr in the

$^{14}C$ age per 1000 yr of simulation. For most models, these drift criteria can be reached only after integrations of a few thousand model years. To reach the spun-up state with the classic approach, i.e. with the online or offline methods outlined above, we request that groups spin up their model for at least 2000 yr, if at all possible. Other approaches to obtain the spun-up state, such as using tracer-acceleration techniques or fast solvers (Li and Primeau, 2008; Khatiwala, 2008; Merlis and Khatiwala, 2008) are also permissible. If used, they should also be applied until models meet the same equilibrium criteria described above.

The spin-up simulation itself should be initialized as for the *omip1* simulation, except for the abiotic tracers and the $^{13}C_T$ tracer. The abiotic initial fields of $A_T^{abio}$ and $C_T^{abio}$ will be provided, being derived from initial fields of $T$ and $S$. Although $C_T^{abio}$ is a passive tracer carried in the model, $A_T^{abio}$ is not. The latter will be calculated from the initial 3-D salinity field as detailed below; then that calculated field will be used to compute $C_T^{abio}$ throughout the water column assuming equilibrium with the preindustrial level of atmospheric $CO_2$ at the initial T and S conditions (using OMIP's carbonate chemistry routines). For

$^{14}C_T^{abio}$, initial fields will be based on those from GLODAPv1 for natural $\Delta^{14}C$ (Key et al., 2004). OMIP will provide these fields with missing grid cells filled based on values from adjacent ocean grid points. Groups that include $^{13}C_T$ in *omip1-spunup* should initialize that in the precursor spin-up simulation to 0‰ following the approach of Jahn et al. (2015). Beware though that equilibration timescales for $^{13}C$ are longer than for $C_T$, implying the need for a much longer spin up.

## 2.3  Geochemical atmospheric forcing

The atmospheric concentration histories of the three inert chemical tracers (CFC-11, CFC-12, and $SF_6$) to be used in OMIP are summarized by Bullister (2015) and shown in Fig. 1. Their atmospheric values are to be held to zero for the first three cycles of the CORE-II forcing, then increased starting on 01 January 1936 (beginning of model year 0237) according to the OMIP protocol. To save computational resources, the inert chemical tracers may be activated only from 1936 onward, starting from zero concentrations in the atmosphere and ocean. The atmospheric $CO_2$ history used to force the OMIP models is the same

as that used for the CMIP6 historical simulation (Meinshausen et al., 2016), while carbon isotope ratios ($\Delta^{14}C$ and $\delta^{13}C$) are the same as those used by C4MIP (Jones et al., 2016). These atmospheric records of $CO_2$ and carbon isotope ratios (Fig. 2) and those for the inert chemical tracers will be made available on the CMIP6 web site. The biogeochemical tracers are to be activated at the beginning of the 310-year simulation (on 01 January 1700), but initialized differently as described above for *omip1* and *omip1-spunup*. The atmospheric concentration of $CO_2$ is to be maintained at the CMIP6 preindustrial reference

of $xCO_2^{atm} = 284.65$ ppm between calendar years 1700.0 and 1850.0, after which it must increase following observations (Meinshausen et al., 2016). The increasing $xCO_2^{atm}$ will thus affect $C_T$ but not $C_T^{nat}$, which sees only the preindustrial reference level of $xCO_2^{atm}$. The increasing $xCO_2^{atm}$ is also seen by $^{13}C_T$ and the two abiotic tracers, $C_T^{abio}$ and $^{14}C_T^{abio}$, to be modeled only in the *omip1-spunup* simulation and its spin up.

## 2.4 Conservation equation

The time evolution equation for all passive tracers is given by

$$\frac{\partial C}{\partial t} = \mathbf{L}(C) + J_C, \tag{1}$$

where $C$ is the tracer concentration; $\mathbf{L}$ is the 3-D transport operator, which represents effects due to advection, diffusion, and convection; and $J_C$ is the internal source-sink term. Conservation of volume is assumed in Eq. 1 and standard units of $\mathrm{mol\,m^{-3}}$ are used for all tracers. For the inert chemical tracers (CFC-11, CFC-12, and $SF_6$), $J_C = 0$. For the abiotic carbon tracers, in the *omip1-spunup* simulation and its spin up, the same term is also null for the total carbon tracer $C_\mathrm{T}$

$$J_{C_\mathrm{T}^{\mathrm{abio}}} = 0, \tag{2}$$

but not for the total radiocarbon tracer $^{14}C_\mathrm{T}^{\mathrm{abio}}$ due to radioactive decay

$$J_{^{14}C_\mathrm{T}^{\mathrm{abio}}} = -\lambda\,^{14}C_\mathrm{T}^{\mathrm{abio}}, \tag{3}$$

where $\lambda$ is the radioactive decay constant for $^{14}C$, i.e.

$$\lambda = ln(2)/5700\,\mathrm{yr} = 1.2160 \times 10^{-4}\,\mathrm{yr^{-1}}, \tag{4}$$

converted to $\mathrm{s^{-1}}$ using the number of seconds per year in a given model. For other biogeochemical tracers $J_C$ is non-zero and often differs between models. For $^{13}C_\mathrm{T}$, $J_C$ includes isotopic fractionation effects.

## 2.5 Air-sea gas exchange

Non-zero surface boundary conditions must also be included for all tracers that are affected by air-sea gas exchange: CFC-11, CFC-12, $SF_6$, dissolved $O_2$, and dissolved inorganic carbon in its various modeled forms ($C_\mathrm{T}$, $C_\mathrm{T}^{\mathrm{nat}}$, $C_\mathrm{T}^{\mathrm{abio}}$, $^{14}C_\mathrm{T}^{\mathrm{abio}}$, and $^{13}C_\mathrm{T}$). In OCMIP2, surface boundary conditions also included a virtual-flux term for some biogeochemical tracers, namely in models that had a virtual salt flux because they did not allow water transfer across the air-sea interface. Water transfer calls for different implementations depending on the way the free-surface is treated, as discussed extensively by Roullet and Madec (2000). Groups that have implemented virtual fluxes for active tracers ($T$ and $S$) should follow the same practices to deal with virtual fluxes of passive tracers such as $C_\mathrm{T}$ and $A_\mathrm{T}$, as detailed in the OCMIP2 design document (Najjar and Orr, 1998) and in the OCMIP2 Abiotic HOWTO (Orr et al., 1999b). In OMIP, all models should report air-sea $CO_2$ fluxes due to gas exchange ($F_{CO_2}$, $F_{CO_2}^{\mathrm{nat}}$, $F_{CO_2}^{\mathrm{abio}}$, $F_{^{14}CO_2}^{\mathrm{abio}}$, and $F_{^{13}CO_2}$) without virtual fluxes included. Virtual fluxes are not requested as they do not directly represent $CO_2$ exchange between the atmosphere and ocean.

Surface boundary fluxes may be coded simply as adding source-sink terms to the surface layer, e.g.

$$J_A = \frac{F_A}{\Delta z_1}, \tag{5}$$

where for gas $A$, $J_A$ is its surface-layer source-sink term due to gas exchange ($\mathrm{mol\,m^{-3}\,s^{-1}}$) and $F_A$ is its air-to-sea flux ($\mathrm{mol\,m^{-2}\,s^{-1}}$), while $\Delta z_1$ is the surface-layer thickness (m).

In OMIP, we parameterize air-sea gas transfer of CFC-11, CFC-12, $SF_6$, $O_2$, $CO_2$, $^{14}CO_2$, and $^{13}CO_2$ using the gas transfer formulation also adopted for OCMIP2 (excluding effects of bubbles):

$$F_A = k_w \left([A]_{sat} - [A]\right),\tag{6}$$

where for gas $A$, $k_w$ is its gas transfer velocity, $[A]$ is its simulated surface-ocean dissolved concentration, and $[A]_{sat}$ is its corresponding saturation concentration in equilibrium with the water-vapor-saturated atmosphere at a total atmospheric pressure $P_a$. Concentrations throughout are indicated by square brackets and are in units of $\text{mol m}^{-3}$.

For all gases that remain purely in dissolved form in seawater, gas exchange is modeled directly with Eq. (6). However for $C_T$, only a small part remains as dissolved gas as mentioned in Sect. 2.6. Thus the dissolved gas concentration $[CO_2^*]$ must first be computed, each time step, from modeled $C_T$ and $A_T$ and then the gas exchange is computed with Eq. (6). For example, for the two abiotic tracers (in *omip1-spunup*):

$$F_{CO_2}^{abio} = k_w \left([CO_2^*]_{sat} - [CO_2^*]\right)\tag{7}$$

and

$$F_{^{14}CO_2}^{abio} = k_w \left(\left[^{14}CO_2^*\right]_{sat} - \left[^{14}CO_2^*\right]\right).\tag{8}$$

For $^{13}C$, isotopic fractionation associated with gas exchange must be included in the flux calculation. We recommend using the formulation of Zhang et al. (1995)

$$F_{^{13}CO_2} = k_w \, \alpha_k \, \alpha_{aq-g} \left(^{13}R_{atm} \, [CO_2^*]_{sat} - \frac{\left[^{13}CO_2^*\right]}{\alpha_{C_T-g}}\right),\tag{9}$$

where $\alpha_k$ is the kinetic fractionation factor, $\alpha_{aq-g}$ is the fractionation factor for gas dissolution, and $\alpha_{C_T-g}$ is the equilibrium fractionation factor between dissolved inorganic carbon and gaseous $CO_2$. $^{13}R_{atm}$ is the $^{13}C/^{12}C$ ratio in atmospheric $CO_2$. Following Zhang et al. (1995), $\alpha_{C_T-g}$ depends on $T$ and the fraction of carbonate in $C_T$, $fCO_3$:

$$\alpha_{C_T-g} = \frac{0.0144\,T_c\,fCO_3 - 0.107\,T_c + 10.53}{1000} + 1,\tag{10}$$

where $T_c$ is temperature in units of °C, while division by 1000 and addition of 1 converts the fractionation factor from $\epsilon$ in units of ‰ into $\alpha$. The $\alpha_{aq-g}$ term depends on temperature following

$$\alpha_{aq-g} = \frac{0.0049\,T_c - 1.31}{1000} + 1.\tag{11}$$

Conversely no temperature dependence was found for $\alpha_k$. Hence we recommend that OMIP modelers use a constant value for $\alpha_k$ of 0.99914 ($\epsilon_k$ of -0.86‰), the average from Zhang et al.'s measurements at 5° and 21°C.

### 2.5.1 Gas transfer velocity

OMIP modelers should use the instantaneous gas transfer velocity $k_w$ parameterization from Wanninkhof (1992), a quadratic function of the 10-m wind speed $u$

$$k_W = a \left(\frac{Sc}{660}\right)^{-1/2} u^2 \left(1 - f_i\right),\tag{12}$$

to which we have added limitation from sea-ice cover following OCMIP2. Here $a$ is a constant, $Sc$ is the Schmidt number, and $f_i$ is the sea-ice fractional coverage of each grid cell (varying from 0 to 1). Normally, the constant $a$ is adjusted so that wind speeds used to force the model are consistent with the observed global inventory of bomb $^{14}$C, e.g. as done in previous phases of OCMIP (Orr et al., 2001; Najjar et al., 2007). Here though, we choose to use one value of $a$ for all simulations, independent of whether models are used in forced (OMIP) or coupled mode (i.e. in CMIP6 DECK [Diagnostic, Evaluation and Characterization of Klima] and historical simulations). For $a$ in OMIP, we rely on the reassessment from Wanninkhof (2014) who used improved estimates of the global-ocean bomb-$^{14}$C inventory along with CCMP (Cross Calibrated Multi-Platform) wind fields in an inverse approach with the Modular Ocean Model (Sweeney et al., 2007) to derive a best value of

$$a = 0.251 \, \frac{\text{cm hr}^{-1}}{(\text{m s}^{-1})^2}, \tag{13}$$

which will give $k_w$ is in $\text{cm hr}^{-1}$ if winds speeds are in $\text{m s}^{-1}$. For model simulations where tracers are carried in $\text{mol m}^{-3}$, $k_w$ should be in units of $\text{m s}^{-1}$; thus, $a$ should be set equal to $6.97 \times 10^{-7} \, \text{m s}^{-1}$. The same value of $a$ should be adopted for the forced OMIP simulations and for Earth System Model simulations made under CMIP6.

### 2.5.2 Schmidt number

Besides $a$, the Schmidt number $Sc$ is also needed to compute the gas transfer velocity (Eq. 12). The Schmidt number is the ratio of the kinematic viscosity of water $\mu$ to the diffusion coefficient of the gas $D$ ($Sc = \mu/D$). The coefficients for the fourth-order polynomial fit of $Sc$ to *in situ* temperature over the temperature range of $-2$ to $40^\circ$C (Wanninkhof, 2014) are provided in Table 1 for each gas to be modeled in OMIP and CMIP6. Fortran 95 routines using the same formula and coefficients for all gases modeled in OMIP are available for download via the *gasx* module of the *mocsy* package (Sect. 2.6).

### 2.5.3 Atmospheric saturation concentration

The surface gas concentration in equilibrium with the atmosphere (saturation concentration) is

$$[A]_{sat} = K_0 \, f_A = K_0 \, C_f \, p_A = K_0 \, C_f \, (P_a - p\text{H}_2\text{O}) \, x_A, \tag{14}$$

where for gas $A$, $K_0$ is its solubility, $f_A$ is its atmospheric fugacity, $C_f$ is its fugacity coefficient, $p_A$ is its atmospheric partial pressure, and $x_A$ is its mole fraction in dry air, while $P_a$ is again the total atmospheric pressure (atm) and $p\text{H}_2\text{O}$ is the vapor pressure of water (also in atm) at sea surface temperature and salinity (Weiss and Price, 1980).

The combined term $K_0 C_f (P_a - p\text{H}_2\text{O})$ is available at $P_a = 1$ atm (i.e. $P_a^0$) for all modeled gases except oxygen. We denote this combined term as $\phi_A^0$ (at $P_a^0$) ; elsewhere it is known as the solubility function $F$ (e.g. Weiss and Price, 1980; Warner and Weiss, 1985; Bullister et al., 2002) but we do not use the latter notation here to avoid confusion with the air-sea flux (Eq. 6). For four of the gases to be modeled in OMIP, the combined solubility function $\phi_A^0$ has been computed using the empirical fit

$$ln\left(\phi_A^0\right) = a_1 + a_2 \left(\frac{100}{T}\right) + a_3 \, ln\left(\frac{T}{100}\right) + a_4 \left(\frac{T}{100}\right)^2 + S\left[b_1 + b_2\left(\frac{T}{100}\right) + b_3\left(\frac{T}{100}\right)^2\right], \tag{15}$$

where $T$ is the model's in-situ, absolute temperature (ITS90) and $S$ is its salinity on the practical salinity scale (PSS-78). Thus separate sets of coefficients are available for $CO_2$ (Weiss and Price, 1980, Table VI), CFC-11 and CFC-12 (Warner and Weiss, 1985, Table 5), and $SF_6$ (Bullister et al., 2002, Table 3), the values of which are detailed in Table 2. For $O_2$, it is not $\phi_A^0$ that is available but rather $[O_2]_{sat}^0$ (Garcia and Gordon, 1992) as detailed below.

Both the solubility function $\phi_A^0$ and the saturation concentration $[A]_{sat}^0$ can be used at any atmospheric pressure $P_a$, with errors of less than 0.1%, by approximating Eq. (14) as

$$[A]_{sat} = \frac{P_a}{P_a^0} \phi_A^0 \, x_A = \frac{P_a}{P_a^0} \, [A]_{sat}^0, \tag{16}$$

where $P_a^0$ is the reference atmospheric pressure (1 atm). Variations in surface atmospheric pressure must not be neglected in OMIP because they alter the regional distribution of $[A]_{sat}$. For example, the average surface atmospheric pressure between

$60°$ and $30°S$ is 3% lower than the global mean, thus reducing surface-ocean $pCO_2$ by $10 \, \mu atm$ and $[O_2]_{sat}$ by $10 \, \mu mol \, kg^{-1}$. The atmospheric pressure fields used to compute gas saturations should also be consistent with the other physical forcing. Thus for the OMIP forced simulations, modelers will use surface atmospheric pressure from CORE II, converted to atm.

For the two abiotic carbon tracers, abbreviating $K' = K_0 C_f$, we can write their surface saturation concentrations (Eq. 14) as

$$[CO_2^*]_{sat}^{abio} = K' \, (P_a - pH_2O) \, x_{CO_2} \tag{17}$$

and

$$\left[^{14}CO_2^*\right]_{sat}^{abio} = [CO_2^*]_{sat} \, {}^{14}r_{atm}'. \tag{18}$$

Here ${}^{14}r_{atm}'$ represents the normalized atmospheric ratio of $^{14}C/C$, i.e.

$${}^{14}r_{atm}' = \frac{{}^{14}r_{atm}}{{}^{14}r_{std}} = \left(1 + \frac{\Delta^{14}C_{atm}}{1000}\right), \tag{19}$$

where ${}^{14}r_{atm}$ is the atmospheric ratio of $^{14}C/C$, ${}^{14}r_{std}$ is the analogous ratio for the standard ($1.170 \times 10^{-12}$, see Appendix A), and $\Delta^{14}C_{atm}$ is the atmospheric $\Delta^{14}C$, the fractionation-corrected ratio of $^{14}C/C$ relative to a standard reference given in permil (see below). We define ${}^{14}r_{atm}'$ and use it in Eq. (18) to be able to compare $C_T^{abio}$ and $^{14}C_T^{abio}$, directly, potentially simplifying code verification and testing. With the above model formulation for the OMIP equilibrium run (where $xCO_2^{atm} = 284.65$ ppm and $\Delta^{14}C^{atm} = 0$ ‰), both $C_T^{abio}$ and $^{14}C_T^{abio}$ have identical units. Short tests with the same initialization for

both tracers can thus verify consistency. Differences in the spin-up simulation will stem only from different initializations and radioactive decay. Differences will grow further during the anthropogenic perturbation (in *omip1-spunup*, i.e. after spin up), because of the sharp contrast between the shape of the atmospheric histories of $xCO_2$ and $\Delta^{14}C_{atm}$.

For $^{13}C$, the $\delta^{13}C_{atm}$ in atmospheric $CO_2$ is incorporated into Eq. (9) through the term $^{13}R_{atm}$, which is given by

$${}^{13}R_{atm} = \left(\frac{\delta^{13}C_{atm}}{1000} + 1\right) {}^{13}R_{std}, \tag{20}$$

where $^{13}R_{std}$ is the standard ratio 0.0112372 (Craig, 1957). In this formulation, unlike for $^{14}C_T^{abio}$, $^{13}C_T$ is not normalized by the standard ratio. However, modeling groups may wish to simulate normalized $^{13}C_T$, e.g. by including a factor of $1/^{13}R_{std}$ analogous to the approach used for $^{13}C_T^{abio}$. Modeling groups that simulate $^{13}C$ in OMIP must report non-normalized values of the concentration $^{13}C_T$ and the air-sea flux $F_{13CO_2}$. No other $^{13}C$ results are requested.

For all gases simulated in OMIP, the atmospheric saturation concentration $[A]_{sat}$ is computed using Eq. (16). For all gases except oxygen, the combined solubility function $\phi_A^0$ is available, being computed each time step using modeled $T$ and $S$ with Eq. (15), the corresponding gas-specific coefficients (Table 2), and the atmospheric mole fraction of each gas $x_A$. The exception is $O_2$ because rather than $x_A$ and $\phi_A^0$, it is the reference saturation concentration $[O_2]_{sat}^0$ that is available (Garcia and Gordon, 1992, equation 8, Table 1).

In all cases, the same $P_a/P_a^0$ term is used to account for effects of atmospheric pressure (Eq. 16). For $P_a$, modelers must use the fields of surface atmospheric pressure *(sap)* from CORE II, i.e. for OMIP's forced ocean simulations (*omip1* and *omip1-spunup*), whereas for any CMIP6 coupled simulation, modelers should use *sap* from the coupled atmospheric model.

To compute $[A]_{sat}$ then, we only need one additional type of information, namely the $x_A$'s for each of $CO_2$, CFC-11, CFC-12, and $SF_6$ as well as corresponding atmospheric histories for carbon isotopes:

1. $x_{CFC-11}$, $x_{CFC-12}$, and $x_{SF_6}$: Atmospheric records for observed CFC-11 and CFC-12 (in parts per trillion - ppt) are based on station data at 41°S and 45°N from (Walker et al., 2000) with subsequent extensions as compiled by Bullister (2015). For OMIP, each station will be treated as representative of its own hemisphere, except between 10°S and 10°N where those station values will be interpolated linearly as a function of latitude. Thus there are 3 zones: 90°S–10°S, where CFC's are held to same value as at the station at 41°S; 10°S–10°N, a buffer zone where values are interpolated linearly; and 10°N–90°N, where values are held to the same value as at the measuring station at 45°N. For $SF_6$, OMIP also relies on the Bullister (2015) synthesis over the same latitudinal bands. Values for all three inert chemical tracers are given at mid-year. It is recommended that modelers linearly interpolate these mid-year values to each time step, because annual growth rates can be large and variable.

2. $x_{CO_2}$: In the *spin-up* simulation, needed to initialize *omip1-spunup* simulation, atmospheric $CO_2$ is held constant at $x_{CO_2} = 284.65$ ppm, the same preindustrial value as used for the CMIP6 *picontrol* simulation. Over the industrial era, defined as between years 1850.0 and 2010.0 for both of OMIP's transient simulations (*omip1* and *omip1-spunup*), atmospheric $x_{CO_2}$ will follow the same observed historical increase as provided for CMIP6 (Meinshausen et al., 2016). Modelers should use the record of global annual mean atmospheric $x_{CO_2}$, interpolated to each time step. That increasing $x_{CO_2}$ affects the total tracer $C_T$ in both transient simulations as well as the two abiotic tracers and $^{13}C_T$ in the *omip1-spunup* simulation. However, it does not affect the natural tracer $C_T^{nat}$, for which the atmosphere is always held at $x_{CO_2} = 284.65$ ppm.

3. $\Delta^{14}C^{atm}$: For the OMIP spin-up simulation, $\Delta^{14}C^{atm}$ is held constant at 0‰. For the *omip1-spunup* simulation, the equilibrium reference is thus year 1850.0. Then the model must be integrated until 2010.0 following the observed record of $\Delta^{14}C^{atm}$ (Levin et al., 2010), separated into three latitudinal bands (90°S–20°S, 20°S–20°N, and 20°N–90°N).

4. $\delta^{13}C^{atm}$: The atmospheric record of $\delta^{13}C$ is the same as adopted for C4MIP, a compilation of ice-core data (Rubino et al., 2013) and atmospheric measurements at Mauna Loa (Keeling, 2001).

### 2.5.4 Surface ocean concentration

The equation above for the atmospheric equilibrium (saturation) concentration of a gas (Eq. 14) should not be confused with the analogous equation for the simulated ocean concentration. The surface-ocean equation allows conversion between the simulated surface-ocean dissolved gas concentration $[A]$, the corresponding fugacity $f_O$, and the partial pressure $p_O$ of the surface ocean as follows:

$$[A] = K_0 \, f_O = K_0 \, C_f \, p_O = K' \, p_O. \tag{21}$$

This surface-ocean equation is analogous to that for the atmospheric equilibrium saturation concentration $[A]_{sat}$ (Eq. 14), except that the ocean equation omits the final portion of the atmospheric equation which computes the mole fraction, a conventional parameter only for the atmosphere. Thus the combined term that includes the atmospheric pressure and humidity corrections (last term in parentheses) in Eq. (14) is not pertinent for the surface ocean equation. It should not be used when converting between simulated oceanic $[A]$ and the corresponding $p_O$. Confusion on this point was apparent in the publicly available OMIP2 code, i.e. for the conversion from $[CO_2^*]$ to $pCO_2$, although that did not affect simulated $F_{CO_2}$.

To avoid potential confusion and redundancy, OMIP modelers may prefer to separately compute the parts of $\phi_A$ rather than computing $\phi_A^0$ and using it directly. Since

$$\phi_A = K_0 \, C_f \, (P_a - pH_2O) = K' \, (P_a - pH_2O), \tag{22}$$

modelers need only compute $K'$, and use that in both the ocean equation (Eq. 21) and the atmospheric saturation equation (Eq. 14), while for the latter also correct for atmospheric pressure and humidity, i.e. the $(P_a - pH_2O)$ term. That combined correction is to be computed with $P_a$ from the CORE II forcing and with $pH_2O$ calculated from model surface $T$ and $S$ (Weiss and Price, 1980, Eq. 10):

$$pH_2O = 24.4543 - 67.4509 \left( \frac{100}{T} \right) - 4.8489 \, ln \left( \frac{T}{100} \right) - 0.000544 \, S, \tag{23}$$

where $pH_2O$ is in atm, $T$ is the in-situ, absolute temperature and $S$ is salinity in permil. In this way, OMIP modelers may avoid using the sometimes confusing combined term $\phi_A^0$ altogether as well as its approximative pressure correction when calculating the saturation concentration (Eq. 16). Pressure corrections for $K'$ may be neglected in the surface ocean where total pressure remains close to 1 atm (Weiss, 1974).

The ocean equation (Eq. 21) converts a simulated dissolved gas concentration to a partial pressure using its combined product $K'$, which can be computed directly for some gases or via a two-step process for others. For OMIP's inert chemical tracers, tabulated coefficients can be used to compute $K'$ directly, i.e. for CFC-11 and CFC-12 (Warner and Weiss, 1985, Table 2) and for $SF_6$ (Bullister et al., 2002, Table 2) using modeled $T$ and $S$ in an equation just like Eq. 15 but without the first $T^2$ term

$(a_4 = 0)$:

$$ln(K') = a_1 + a_2 \left( \frac{100}{T} \right) + a_3 \, ln \left( \frac{T}{100} \right) + S \left[ b_1 + b_2 \left( \frac{T}{100} \right) + b_3 \left( \frac{T}{100} \right)^2 \right], \tag{24}$$

where $T$ is the in-situ absolute temperature and $S$ is salinity in permil.

For $O_2$, $K'$ is not needed for the saturation calculations, but it is necessary when using the simulated dissolved $[O_2]$ to
compute the corresponding surface ocean $pO_2$, a required variable for OMIP and CMIP6. That solubility conversion factor $K'$
can be derived by substituting its definition into Eq. (14) and rearranging, so that

$$K'_{O2} = \frac{[O_2]^0_{sat}}{x_{O_2} (P^0_a - pH_2O)}, \tag{25}$$

where the numerator is from Eq. 8 of Garcia and Gordon (1992) using coefficients from their Table 1, and the denominator is
the product of the corresponding constant atmospheric mole fraction of $O_2$ ($x_{O_2} = \text{O}.20946$) and the wet-to-dry correction at
1 atm as described above. The computed $K'_{O2}$ is then exploited to compute the partial pressure of oxygen ($pO_2 = [O_2] / K'_{O2}$).

For $CO_2$, tabulated coefficients are not available to compute $K'$, but they are available to compute $K_0$ (Weiss, 1974, Table 1).
Hence given that $K' = K_0 \, C_f$, modelers must also compute the fugacity coefficient $C_f$ from Eq. 9 of Weiss (1974):

$$C_f = \exp \left[ \left( B + 2 x_2^2 \, \delta_{12} \right) \frac{P_{ao}}{RT} \right], \tag{26}$$

where $B$ is the virial coefficient of $CO_2$ (Weiss, 1974, Eq. 6), $x_2$ is the sum of the mole fractions of all remaining gases
($1 - xCO_2$, when $xCO_2 \ll 1$), and $\delta_{12} = 57.7 - 0.118T$. Here $P_{ao}$ is the total pressure (atmospheric + hydrostatic) in atm, $R$
is the gas constant ($82.05736 \, \text{cm}^3 \, \text{atm} \, \text{mol}^{-1} \, \text{K}^{-1}$), and $T$ is the in-situ absolute temperature (K).

Although the surface ocean concentration of dissolved carbon dioxide gas $[CO_2^*]$ is needed to compute air-sea $CO_2$ ex-
change, it is not that inorganic carbon species that is carried as a tracer in ocean carbon models (Sect. 2.6). Instead the $[CO_2^*]$
concentration ($\text{mol} \, \text{m}^{-3}$) must be computed each time step from a model's simulated surface $C_T$, $A_T$, $T$, and $S$ as well as nu-
trient concentrations (total dissolved inorganic phosphorus $P_T$ and silicon $Si_T$) as detailed in the following section. All OMIP
biogeochemical models will carry $C_T$ and $A_T$ as passive tracers. Most if not all will also carry at least one inorganic nutrient,
nitrogen or phosphorus. Some will carry silicon. For models that carry only nitrogen, it is preferred that they compute $P_T$ by
dividing the total dissolved inorganic nitrogen concentration by 16, the constant N:P ratio from Redfield et al. (1963). For
models without $Si_T$, it is preferred that they use climatological $Si_T$ data interpolated to their model grid (i.e. annual average
data from WOA2013). These options offer a better alternative than assuming nutrient concentrations are zero, which lead to
systematic shifts of order of 10 $\mu$atm in calculated surface-water $pCO_2$.

The abiotic portion of the biogeochemical simulation, carries only two tracers, $C_T^{abio}$ and $^{14}C_T^{abio}$, which are not connected
to other biogeochemical tracers. Hence to compute corresponding abiotic $[CO_2^*]$ and $\left[ ^{14}CO_2^* \right]$ concentrations, we also need
abiotic alkalinity. Following OCMIP2, the abiotic alkalinity in OMIP will be calculated simply as a normalized linear function
of salinity:

$$A_T^{abio} = \overline{A_T} \left( \frac{S}{\overline{S}} \right), \tag{27}$$

where $\overline{A_T}$ is the global mean of surface observations 2297 $\mu\mathrm{mol\,kg}^{-1}$ (Lauvset et al., 2016) and $\overline{S}$ is the model's global- and annual-mean surface salinity. In practice, it is recommended that $\overline{S}$ is first computed as the global mean of the initial salinity field and then, after one year of simulation, from the annual mean salinity of the previous year. Also needed are two other input arguments, $P_T$ and $Si_T$. Although accounting for both of their acid systems makes a difference, these abiotic tracers are not included along with abiotic $C_T$. Hence we take their concentrations as being constant, equal to the global mean of surface observations for $P_T$ of 0.5 $\mu\mathrm{mol\,kg}^{-1}$ and for $Si_T$ of 7.5 $\mu\mathrm{mol\,kg}^{-1}$. The assumption of constant nutrient distributions applies only to the carbonate chemistry calculations for abiotic $C_T$.

For the abiotic simulation's radiocarbon tracer, we must likewise compute its surface-ocean dissolved gas concentration $\left[^{14}\mathrm{CO}_2^*\right]$. The latter is related to the calculated dissolved gas concentration of the stable abiotic carbon tracer as follows:

$$\left[^{14}\mathrm{CO}_2^*\right]^{\mathrm{abio}} = \left[\mathrm{CO}_2^*\right]^{\mathrm{abio}}\,^{14}r'_{\mathrm{ocn}}, \tag{28}$$

where

$$^{14}r'_{\mathrm{ocn}} = \frac{^{14}r_{\mathrm{ocn}}}{^{14}r_{\mathrm{std}}} = \frac{^{14}C_T^{\mathrm{abio}}}{C_T^{\mathrm{abio}}} \tag{29}$$

and $^{14}r_{\mathrm{ocn}}$ is the $^{14}\mathrm{C/C}$ of seawater. This normalization essentially means that $^{14}C_T^{\mathrm{abio}}$ represents the actual fractionation-corrected $^{14}\mathrm{C}$ concentration divided by $^{14}r_{\mathrm{std}}$. This output must be saved in normalized form. But for subsequent $^{14}\mathrm{C}$ budget calculations, it will be necessary to back-correct the normalized and fractionation-corrected modeled concentration ($^{14}C_T^{\mathrm{abio}}$) and $^{14}\mathrm{C}$ flux ($F_{^{14}\mathrm{CO}_2}^{\mathrm{abio}}$), i.e. the only two $^{14}\mathrm{C}$ variables saved in OMIP, to molar units of actual $^{14}\mathrm{C}$ (see Appendix A). For eventual comparison to ocean measurements, one can compute oceanic $\Delta^{14}C$ as

$$\Delta^{14}C_{\mathrm{ocn}}^{\mathrm{abio}} = 1000\left(^{14}r'_{\mathrm{ocn}} - 1\right). \tag{30}$$

For $^{13}\mathrm{C}$, the surface ocean dissolved gas concentration $\left[^{13}\mathrm{CO}_2^*\right]$ is given by

$$\left[^{13}\mathrm{CO}_2^*\right] = \left[\mathrm{CO}_2^*\right]\,^{13}r_{\mathrm{ocn}}, \tag{31}$$

where $^{13}r_{\mathrm{ocn}} =^{13} C_T/C_T$. Here $^{13}C_T$ is not normalized by the standard ratio, but modeling groups may wish to simulate normalized $^{13}C_T$ by including a factor of $1/^{13}r_{\mathrm{std}}$, analogous to what is done for the $^{14}C_T^{\mathrm{abio}}$ normalization above.

## 2.6 Carbonate chemistry

Unlike other modelled gases in OMIP, $CO_2$ does not occur in seawater as a simple dissolved passive tracer. Instead, it reacts with seawater forming carbonic acid ($H_2CO_3$), most of which dissociates into two other inorganic species, bicarbonate ($HCO_3^-$) and carbonate ($CO_3^{2-}$) ions. Since dissolved $CO_2$ cannot be distinguished analytically from the much less abundant $H_2CO_3$, common practice is to refer to the sum of the two, $CO_2 + H_2CO_3$, as $CO_2^*$. The sum of the three species $CO_2^* + HCO_3^- + CO_3^{2-}$ is referred to as total dissolved inorganic carbon $C_T$, while their partitioning depends on seawater pH, temperature, salinity, and pressure. The pH may be calculated from $C_T$ and seawater's ionic charge balance, formalised as

total alkalinity $A_T$. Both $C_T$ and $A_T$ are conservative with respect to mixing and changes in seawater temperature, salinity, and pressure. Hence both are carried as passive tracers in all ocean models, and both are used, along with temperature, salinity, and nutrient concentrations, to compute the dissolved concentration of $CO_2$ and the related $pCO_2$, as needed to compute air-sea $CO_2$ fluxes.

To simulate carbonate chemistry, OMIP groups should use the total pH scale and the equilibrium constants recommended for best practices (Dickson et al., 2007; Dickson, 2010). Additionally, the model's total alkalinity equation should include alkalinity from phosphoric and silicic acid systems as well as from carbonic acid, boric acid, and water, namely

$$A_T = A_C + A_B + A_W + A_P + A_{SI} + A_O, \tag{32}$$

where

$$A_C = \left[HCO_3^-\right] + 2\left[CO_3^{2-}\right], \tag{33}$$

$$A_B = \left[B(OH)_4^-\right], \tag{34}$$

$$A_W = [OH^-] - [H^+]_F - \left[HSO_4^-\right] - [HF], \tag{35}$$

$$A_P = \left[HPO_4^{2-}\right] + 2\left[PO_4^{3-}\right] - [H_3PO_4], \tag{36}$$

$$A_{SI} = \left[SiO(OH)_3^-\right], \tag{37}$$

$$A_O = [NH_3] + [HS^-] + \ldots \tag{38}$$

The right side of Eq. (32) thus separates the contributions from components of carbonic acid, boric acid, water, phosphoric acid, silicic acid, and other species, respectively. Neglect of $A_P$ and $A_{SI}$ has been common among model groups but leads to systematic errors in computed $pCO_2$, e.g. in the Southern Ocean (Najjar and Orr, 1998; Orr et al., 2015). Models with the nitrogen cycle should also account for effects of changes in the different inorganic forms of nitrogen on total alkalinity,

including changes due to denitrification and nitrogen fixation plus nitrification. Models with $P_T$ as the sole macronutrient tracer should consider accounting for the effect of nitrate assimilation and remineralization on alkalinity, effects that are 16 times larger than for those for $P_T$ (Wolf-Gladrow et al., 2007).

Although phosphorus and silicon alkalinity is included in the carbonate chemistry routines provided for OCMIP2 and OCMIP3 (Orr et al., 1999b; Aumont et al., 2004), those routines focused only on computing surface $pCO_2$ and are now

outdated. They have been replaced by *mocsy*, a new Fortran 95 package for ocean modelers (Orr and Epitalon, 2015). Relative to the former OCMIP code, *mocsy* computes derived variables (e.g. $pCO_2$, pH, $CO_3^{2-}$, and $CaCO_3$ saturation states) throughout the water column, corrects for common errors in pressure corrections, and replaces the solver of the pH-Alkalinity equation with the faster and safer SolveSaphe algorithm from Munhoven (2013). The latter converges under all conditions, even for very low salinity (low $C_T$ and $A_T$), unlike other approaches. Although by default *mocsy* uses older scales for temperature and salin-

ity (ITS90 and PSS78, respectively) for input, it now includes a new option so that modelers can choose to use the TEOS-10 standards (Conservative Temperature and Absolute Salinity) instead. The *mocsy* routines may be downloaded by issuing the following command:

```
git clone https://github.com/jamesorr/mocsy.git
```

and then installed by typing make. Alternatively, it can be dowloaded directly from the same site as a zipfile.

## 3  Diagnostics

The second goal of OMIP-BGC is to provide a complete list of diagnostics requested for the ocean simulations of in-
ert chemistry and biogeochemistry within the framework of OMIP and CMIP6. The limited diagnostics requested for the
simulations of inert chemistry are provided in Table 4. The diagnostics requested for the biogeochemical simulations are
more extensive. Hence they are given here as a series of tables separated by priority, type, and output frequency, i.e. as an-
nual means (Tables 5 to 8), monthly means (Tables 9 to 16), and daily means (Table 17). The same list of requested vari-
ables is given in a different form and with more detail in the OMIP-BGC MIP tables for CMIP6, which are available from
https://earthsystemcog.org/projects/wip/CMIP6DataRequest.

Conceptually there is no difference in output requirements for the forced ocean simulations made for OMIP and the coupled
simulations made with the ESMs that are participating in CMIP6 (e.g., DECK and historical). These simulations differ in
forcing but not in the types of output requested.

To foster analysis of the model output generated by OMIP and CMIP6, OMIP-BGC plans to encourage contributions to a
centralized list of analysis subprojects. The aim is to promote collaboration while avoiding excessive redundancy to allow the
international community to advance more quickly and to exploit a greater diversity of output. Although much analysis will be
led by OMIP members, others will also be encouraged to participate, e.g., scientists from other CMIP6 projects (e.g. C4MIP)
or projects outside of CMIP (e.g., FishMIP or MAREMIP).

## 4  Conclusions

The required OMIP simulation *(omip1)* will be performed by many groups, each of which will couple their global-ocean, sea-
ice model to a passive-tracer transport model for inert chemistry and ocean biogeochemistry, online. All groups, even those
without biogeochemistry, will include at least one inert chemistry tracer (CFC-12) to assess subsurface model ventilation;
two other tracers (CFC-11 and $SF_6$) are also requested to better assess subsurface watermass ages relative to observations.
Groups with ocean biogeochemical models should also include that component (OMIP-BGC). The physical component will
be forced with the CORE II forcing (1948–2009) over five repeated cycles (310 years) as described in the companion OMIP
paper (Griffies et al., 2016). The biogeochemical component will be connected for the full 310 yr. Each model's atmospheric
$CO_2$ will be held to the CMIP6 preindustrial level (1 January 1850) during the first 150 years (1700–1849), while for the next
160 years (1850-2009) models will be forced to follow the historical observations as defined for CMIP6. Physical analyses will
focus on the fifth cycle, while those for the chemistry and biogeochemistry will also study transient changes over the industrial
era. All OMIP-BGC simulations should include either the *natural* carbon tracer $C_T^{nat}$, or a parallel separate simulation that
accounts only for natural carbon, in order to assess and remove effects of model drift.

An optional simulation *(omip1-spunup)* is requested from all groups having biogeochemistry and able to afford a long spin up, made beforehand. Rather than using observed climatologies to initialize the biogeochemistry as in *omip1*, this simulation will be initialized with model tracer fields that have been spun up preferably for 2000 years or more. In addition, the *omip1-spunup* simulation (and its spin up) will include two simplified tracers, abiotic carbon and radiocarbon, to evaluate deep-ocean

circulation and deconvolve physical vs. biological contributions to the carbon cycle. Finally, groups already having $^{13}$C as a biogeochemical tracer are encouraged to include that in the *omip1-spunup* simulation (and its spin up), using commmon OMIP formulations for gas exchange and fractionation, to evaluate the simulated Suess effect and to compare cycling of $^{13}$C in the marine ecosystem. Besides the initial fields and the three new tracers, the *omip1* and *omip1-spunup* simulation protocols are identical.

**5    Data and code availability**

To facilitate comparison, an OMIP-BGC web page (http://omip-bgc.lsce.ipsl.fr), now under construction, will provide links to these protocols as well as links to distribute OMIP-BGC's common atmospheric gas histories, data fields for initialization, and code to compute all facets of gas exchange and carbonate chemistry. This site will be open for public use on or before the publication of the final version of the paper. All related data files and code will be made available there. The code currently

mentioned in this manuscript is available in the *mocsy* package, which can be obtained as detailed in Sect. 2.6. That package contains not only the carbonate chemistry routines, but also routines in the *gasx* module to compute Schmidt numbers, solubilities, and air-sea exchange for the gases to be modeled during OMIP ($CO_2$, $O_2$, CFC-11, CFC-12, and $SF_6$).

**Appendix A:    Converting modeled $^{14}$C fluxes to conventional units**

The $^{14}$C tracer that is adopted for OMIP from OCMIP is fractionation corrected to avoid the need to explicitly compute $^{13}$C

fluxes between modeled carbon reservoirs. It is also normalized. Both of these manipulations affect the units of modeled $^{14}$C concentrations and fluxes. These normalized, fractionation-corrected units must be used when OMIP model groups save their $^{14}$C output. The saved OMIP model output is used directly to calculate simulated $\Delta^{14}C_{ocn}$ with Eqs. (29) and (30) for comparison to observations, but for budget calculations it must be converted to atoms or moles of $^{14}$C (Naegler, 2009). Here we detail that conversion.

As mentioned in Sects. 2.5.3 and 2.5.4, modeled $^{14}$C ratios in OMIP are expressed relative to total carbon, i.e. the *fractional isotopic abundance* $^{14}r_{model} = {}^{14}C/C$; conversely, for $^{13}$C, its ratio is typically shown relative to $^{12}$C (Mook, 1986), i.e. with the *isotopic ratio* $^{13}R = {}^{13}C/{}^{12}C$. The *fractional abundance* approach is convenient for ocean carbon-cycle models, which already transport total carbon, e.g. to assess uptake of fossil $CO_2$, which includes both $^{12}$C and $^{13}$C. But whether $^{12}$C or C is the reference, there is only a small effect on simulated results. That is, $^{13}$C amounts to only about 1% of the total carbon

($^{13}R_{std} = 0.0112372$, Craig, 1957) and $^{14}$C is proportionally much less still. For $^{14}$C, we adopt as a reference the standard isotopic fractional abundance $^{14}r_{std}$ ($^{14}C/C$) of $1.170 \times 10^{-12}$, which follows from the absolute international standard activity

for $^{14}r_{\text{std}}$ of $13.56 \pm 0.07$ disintegrations per minute (dpm) per g C (Karlen et al., 1965) and a radiocarbon half-life of $5700 \pm 30$ yr (Audi et al., 2003; Bé et al., 2013). For comparison, Karlen et al. (1965) used the now outdated value for the half-life ($5730 \pm 40$ yr, Godwin, 1962) to infer that $^{14}r_{\text{std}} = 1.176 \times 10^{-12}$; both of those values should now be revised downward to the values provided in the previous sentence.

The purpose of $\Delta^{14}C$ and the fractionation-normalized ratio $^{14}r_N$ is to remove the impact of isotopic fractionation to isolate the effect of "aging" by radioactive decay. Such fractionation occurs during photosynthesis and air-sea $CO_2$ exchange, leading to differences in the $^{13}C/^{12}C$ signature in different reservoirs; without fractionation, that ratio would not differ between carbon reservoirs. Fractionation of $^{14}C$ is about twice that of $^{13}C$ in permil units, based on the atomic mass difference relative to $^{12}C$. One can approximately remove the influence of fractionation on $^{14}C$ by relying on measured $\delta^{13}C$ referenced to a common

isotopic $\delta^{13}C$ signature, taken as -25‰ (Broecker and Olson, 1961). Thus for a particular reservoir $i$ where $^{14}r_i = {}^{14}C/C$:

$$^{14}r_{\text{N,i}} = {}^{14}r_i \left[ 1 - 2 \left( \frac{\delta^{13}C_i + 25}{1000} \right) \right], \tag{A1}$$

where the two terms in the numerator in parentheses are in permil, and

$$\Delta^{14}C_i = \left( \frac{^{14}r_{\text{N,i}}}{^{14}r_{\text{std}}} - 1 \right) 1000. \tag{A2}$$

Deviations between this correction and the actual impact of fractionation on $^{14}C$ occur under non-steady state conditions.

More importantly, radioactive decay in the ocean results in a net transfer of $^{14}C$ into the ocean, unlike the case for $^{13}C$, and this net $^{14}C$ flux is not corrected for fractionation. In OMIP, we simplify equations and avoid small numerical values by defining $^{14}r' = {}^{14}r/^{14}r_{\text{std}}$, i.e. compare Eq. (A2) with Eq. (30). This normalization is further discussed in Sects. 2.5.3 and 2.5.4 (see in particular in Eqs. (19) and (29)).

    Thus OMIP simulates a $^{14}C$ concentration that is (1) fractionation corrected and (2) normalized by dividing $^{14}r$ by $^{14}r_{\text{std}}$.

These corrections must be removed to convert modeled concentrations into number of atoms or moles of $^{14}C$. Thus, we rearrange Eq. (A1) while multiplying by the common denominator (C) of both its $^{14}r$ values and then we multiply by $^{14}r_{\text{std}}$, yielding

$$^{14}C = \left( {}^{14}C_{\text{model}} \Big/ \left[ 1 - 2 \left( \frac{(\delta^{13}C + 25)}{1000} \right) \right] \right) {}^{14}r_{\text{std}}. \tag{A3}$$

Here we neglect that the $\delta^{13}C$ of the standard material ($-19$‰, Karlen et al., 1965) differs from that of ocean water ($-1$ to

2‰) because the resulting bias in computed $^{14}C$ is only 0.02 %.

    Now let us use Eq. (A3) to compute corrections for the preindustrial ocean and atmosphere by plugging in their estimated $\delta^{13}C$ values. For the preindustrial ocean, we assume that $\delta^{13}C$ was around 2‰ in surface waters and 0‰ in the deep ocean, a difference attributable to biological fractionation. Inserting those numbers into Eq. (A3) and simplifying, we thus have

$$^{14}C_S \approx {}^{14}C_{\text{model},S} \left( 1 + \frac{54}{1000} \right) {}^{14}r_{\text{std}} \tag{A4a}$$

$$^{14}C_D \approx {}^{14}C_{\text{model},D} \left( 1 + \frac{50}{1000} \right) {}^{14}r_{\text{std}}, \tag{A4b}$$

where the subscripts $S$ and $D$ indicate surface and deep waters. Thus, there is a correction of 54‰ for the surface ocean and 50‰ for the deep ocean. For the preindustrial atmosphere, using the same approach with its assumed $\delta^{13}C$ of -6.4‰, we find

$$^{14}C_A \approx {}^{14}C_{\text{model},A} \left(1 + \frac{37}{1000}\right) {}^{14}r_{\text{std}}. \tag{A5}$$

Thus, the $^{14}C_A$ correction to switch from model to conventional units for the atmosphere is about 37‰.

Turning to the gas exchange, in the model formulation the related change in the atmospheric $^{14}C$ inventory is calculated by removing the net air-to-sea flux $F^{\text{abio}}_{^{14}CO_2}$ and adding that to the ocean $^{14}C$ inventory. To convert this modeled air-sea flux into atomic units, we use the same correction as for the modeled concentrations because the change in inventory is proportional to the change in concentrations. The difference between the atmospheric and oceanic corrections ($54 - 37 = 17$‰) is related to the equilibrium fractionation factor for air-sea transfer, i.e. 8–9‰ for $^{13}C$ and double that for $^{14}C$. In the model, the impact

of fractionation on the net (non-zero) radiocarbon transfer is not taken explicitly into account giving rise to this inconsistency even under equilibrium conditions where a climatological average flux replaces the ocean sink by radioactive decay.

In the OMIP simulations, atmospheric radiocarbon is prescribed and forces the ocean. The ocean radiocarbon inventory changes in response to this forcing. Thus, a correction of about $+50$‰ (Eqs. (A4a) and (A4b)) is needed to convert ocean $^{14}C$ concentrations and net air-sea $^{14}C$ fluxes from model units into molar units. For concentrations,

$$^{14}C_{\text{T}} = {}^{14}C^{\text{abio}}_{\text{T, model}} \times 1.05 \times 1.170 \times 10^{-12}, \tag{A6}$$

and for fluxes,

$$F_{^{14}CO_2} = F^{\text{abio}}_{^{14}CO_2,\text{model}} \times 1.05 \times 1.170 \times 10^{-12}. \tag{A7}$$

In both Eqs. (A6) and (A7), units on the left-hand side are in terms of mol $^{14}C$ while those for the first term on the right-hand side are model units (*normalized* and *fractionation-corrected* mol $^{14}C$).

*Acknowledgements.* JCO and LB were supported by the EU H2020 CRESCENDO project (grant 641816). JLB was supported by the NOAA Climate Program Office—this is PMEL contribution #4476. HG was supported by a Marie Curie Career Integration Grant from the European Commission. RGN was supported by NASA's Ocean Biology and Biogeochemistry Program and NASA's Interdisciplinary Science Program.

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

**Table 1.** Seawater coefficients for fit of $Sc$ to temperature[*],[†] from Wanninkhof (2014).

| Gas | A | B | C | D | E | $Sc$ (20°C) |
|-----|---|---|---|---|---|-------------|
| CFC-11 | 3579.2 | –222.63 | 7.5749 | –0.14595 | 0.0011874 | 1179 |
| CFC-12 | 3828.1 | –249.86 | 8.7603 | –0.1716 | 0.001408 | 1188 |
| $SF_6$ | 3177.5 | –200.57 | 6.8865 | –0.13335 | 0.0010877 | 1028 |
| $CO_2$ | 2116.8 | –136.25 | 4.7353 | –0.092307 | 0.0007555 | 668 |
| $O_2$ | 1920.4 | –135.6 | 5.2122 | –0.10939 | 0.00093777 | 568 |
| $N_2O$ | 2356.2 | –166.38 | 6.3952 | –0.13422 | 0.0011506 | 697 |
| DMS | 2855.7 | –177.63 | 6.0438 | –0.11645 | 0.00094743 | 941 |

[*] Coefficients for fit to $Sc = A + B\,T_c + C\,T_c^2 + D\,T_c^3 + E\,T_c^4$, where $T_c$ is surface temperature in °C

[†] Conservative Temperature should be converted to in situ temperature before using these coefficients

**Table 2.** Coefficients for fit[*,†,‡] of solubility function $\phi_A^0$ ($\mathrm{mol\,L^{-1}\,atm^{-1}}$).

| Gas | $a_1$ | $a_2$ | $a_3$ | $a_4$ | $b_1$ | $b_2$ | $b_3$ |
|-----|-------|-------|-------|-------|-------|-------|-------|
| CFC-11 | -229.9261 | 319.6552 | 119.4471 | -1.39165 | -0.142382 | 0.091459 | -0.0157274 |
| CFC-12 | -218.0971 | 298.9702 | 113.8049 | -1.39165 | -0.143566 | 0.091015 | -0.0153924 |
| $SF_6$ | -80.0343 | 117.232 | 29.5817 | 0.0 | 0.0335183 | -0.0373942 | 0.00774862 |
| $CO_2$ | -160.7333 | 215.4152 | 89.8920 | -1.47759 | 0.029941 | -0.027455 | 0.0053407 |
| $N_2O$ | -165.8806 | 222.8743 | 92.0792 | -1.48425 | -0.056235 | 0.031619 | -0.0048472 |

[*] Fit to Eq. (15), where $T$ is in-situ, absolute temperature (K) and S is salinity (practical salinity scale).

[†] For units of $\mathrm{mol\,m^{-3}\,atm^{-1}}$, coefficients should be multiplied by 1000.

[‡] The units refer to atm of each gas, not atm of air.

[§] When using these coefficients, conservative temperature should be converted to in situ temperature (K) and absolute salinity should be converted to practical salinity.

**Table 3.** Coefficients for fit of $K'$ and $K_0$ (both in $\mathrm{mol\,L^{-1}\,atm^{-1}}$).

| Gas | $a_1$ | $a_2$ | $a_3$ | $b_1$ | $b_2$ | $b_3$ |
|---|---|---|---|---|---|---|
| | | | | $K'$ | | |
| CFC-11 | -134.1536 | 203.2156 | 56.2320 | -0.144449 | 0.092952 | -0.0159977 |
| CFC-12 | -122.3246 | 182.5306 | 50.5898 | -0.145633 | 0.092509 | -0.0156627 |
| $SF_6$ | -96.5975 | 139.883 | 37.8193 | 0.0310693 | -0.0356385 | 0.00743254 |
| | | | | $K_0$ | | |
| $CO_2$ | -58.0931 | 90.5069 | 22.2940 | 0.027766 | -0.025888 | 0.0050578 |
| $N_2O$ | -62.7062 | 97.3066 | 24.1406 | -0.058420 | 0.033193 | -0.0051313 |

[*] Fit to Eq. 24, where $T$ is in-situ, absolute temperature (K) and S is salinity (practical scale).

[†] The final three footnotes of Table 2 also apply here.

**Table 4.** Output for inert chemistry.

| Symbol | Variable name | Units | Shape | Priority | Long name |
|---|---|---|---|---|---|
| | | | Annual means | | |
| $SF_6$ | sf6 | mol m$^{-3}$ | XYZ | 2 | Mole Concentration of SF6 in Seawater |
| CFC-11 | cfc11 | mol m$^{-3}$ | XYZ | 2 | Mole Concentration of CFC-11 in Seawater |
| CFC-12 | cfc12 | mol m$^{-3}$ | XYZ | 1 | Mole Concentration of CFC-12 in Seawater |
| | | | Monthly means | | |
| $SF_6$ | sf6 | mol m$^{-3}$ | XYZ | 2 | Mole Concentration of SF6 in Seawater |
| CFC-11 | cfc11 | mol m$^{-3}$ | XYZ | 2 | Mole Concentration of CFC-11 in Seawater |
| CFC-12 | cfc12 | mol m$^{-3}$ | XYZ | 1 | Mole Concentration of CFC-12 in Seawater |
| $F_{SF_6}$ | fgsf6 | mol m$^{-2}$ s$^{-1}$ | XY | 2 | Surface Downward SF6 flux |
| $F_{CFC-11}$ | fgcfc11 | mol m$^{-2}$ s$^{-1}$ | XY | 2 | Surface Downward CFC-11 flux |
| $F_{CFC-12}$ | fgcfc12 | mol m$^{-2}$ s$^{-1}$ | XY | 1 | Surface Downward CFC-12 flux |

**Table 5.** Annual-mean biogeochemical output: Priority 1.

| Symbol | Variable name | Units | Shape | Priority | Long name |
|---|---|---|---|---|---|
| $C_\mathrm{T}$ | dissic | mol m$^{-3}$ | XYZ | 1 | Dissolved Inorganic Carbon Concentration |
| $C_\mathrm{T}^\mathrm{nat}$ | dissicnat | mol m$^{-3}$ | XYZ | 1 | Natural Dissolved Inorganic Carbon Concentration |
| $C_\mathrm{T}^\mathrm{abio}$ | dissicabio | mol m$^{-3}$ | XYZ | 1 | Abiotic Dissolved Inorganic Carbon Concentration |
| $^{14}C_\mathrm{T}^\mathrm{abio}$ | dissi14cabio | mol m$^{-3}$ | XYZ | 1 | Abiotic Dissolved Inorganic 14Carbon Concentration |
| $^{13}C_\mathrm{T}$ | dissi13c | mol m$^{-3}$ | XYZ | 1 | Dissolved Inorganic 13Carbon Concentration |
| $A_\mathrm{T}$ | talk | mol m$^{-3}$ | XYZ | 1 | Total Alkalinity |
| $A_\mathrm{T}^\mathrm{nat}$ | talknat | mol m$^{-3}$ | XYZ | 1 | Natural Total Alkalinity |
| pH | ph | 1 | XYZ | 1 | pH |
| pH$^\mathrm{nat}$ | phnat | 1 | XYZ | 1 | Natural pH |
| pH$^\mathrm{abio}$ | phabio | 1 | XYZ | 1 | Abiotic pH |
| $O_2$ | o2 | mol m$^{-3}$ | XYZ | 1 | Dissolved Oxygen Concentration |
| $NO_3^-$ | no3 | mol m$^{-3}$ | XYZ | 1 | Dissolved Nitrate Conentration |
| $P_\mathrm{T}$ | po4$^{a,*}$ | mol m$^{-3}$ | XYZ | 1 | Total Dissolved Inorganic Phosphorus Concentration |
| $Si_\mathrm{T}$ | si$^\dagger$ | mol m$^{-3}$ | XYZ | 1 | Total Dissolved Inorganic Silicon Concentration |
| Fe | dfe$^\ddagger$ | mol m$^{-3}$ | XYZ | 1 | Mole Concentration of Dissolved Iron in sea water |
| Chl | chl$^\S$ | kg m$^{-3}$ | XYZ | 1 | Mass Concentration of Total Chlorophyll in Seawater |
| $F_{CO_2^\mathrm{tot}}$ | fgco2 | kg m$^{-2}$ s$^{-1}$ | XY | 1 | Surface Downward Flux of Total CO2 |
| $F_{CO_2^\mathrm{nat}}$ | fgco2nat | kg m$^{-2}$ s$^{-1}$ | XY | 1 | Surface Downward Flux of Natural CO2 |
| $F_{CO_2^\mathrm{abio}}$ | fgco2abio | kg m$^{-2}$ s$^{-1}$ | XY | 1 | Surface Downward Flux of Abiotic CO2 |
| $F_{^{14}CO_2^\mathrm{abio}}$ | fg14co2abio | kg m$^{-2}$ s$^{-1}$ | XY | 1 | Surface Downward Flux of Abiotic 14CO2 |
| $F_{^{14}CO_2}$ | fg13co2 | kg m$^{-2}$ s$^{-1}$ | XY | 1 | Surface Downward Flux of 13CO2 |

$^a$ For models that do not carry $P_\mathrm{T}$ as a tracer, it should be computed from $NO_3^-$ assuming N:P $= 16{:}1$

$^*$ $P_\mathrm{T} = H_3PO_4 + H_2PO_4^- + HPO_4^{2-} + PO_4^{3-}$. In seawater most of $P_\mathrm{T}$ is in the form of $HPO_4^{2-}$, while $PO_4^{3-}$ makes up only $\sim$10% at pH 8.

$^\dagger$ $Si_\mathrm{T} = \left[Si(OH)_4\right] + \left[SiO(OH)_3^-\right]$, i.e. the sum of silicic acid and silicate

$^\ddagger$ modeled dissolved iron includes all simulated dissolved species, both free and organically complexed

$^\S$ sum of chlorophyll from all phytoplankton group concentrations. In most models this is equal to chldiat+chlmisc.

**Table 6.** Annual-mean biogeochemical output: Priority 2 (concentrations).

| Symbol | Variable name | Units | Shape | Priority | Long name |
|---|---|---|---|---|---|
| DOC | dissoc | mol m$^{-3}$ | XYZ | 2 | Dissolved Organic Carbon Concentration |
| | phyc | mol m$^{-3}$ | XYZ | 2 | Phytoplankton Carbon Concentration |
| | zooc | mol m$^{-3}$ | XYZ | 2 | Zooplankton Carbon Concentration |
| | detoc | mol m$^{-3}$ | XYZ | 2 | Detrital Organic Carbon Concentration |
| $[CaCO_3]_{calc}$ | calc | mol m$^{-3}$ | XYZ | 2 | Calcite Concentration |
| $[CaCO_3]_{arag}$ | arag | mol m$^{-3}$ | XYZ | 2 | Aragonite Concentration |
| $[O_2]_{sat}$ | o2sat | mol m$^{-3}$ | XYZ | 2 | Dissolved Oxygen Concentration at Saturation |
| $[NH_4^+]$ | nh4 | mol m$^{-3}$ | XYZ | 2 | Dissolved Ammonium Concentration |
| | chldiat[*] | kg m$^{-3}$ | XYZ | 2 | Mass Concentration of Diatom expressed as Chlorophyll in sea water |
| | chldiaz[†] | kg m$^{-3}$ | XYZ | 2 | Mass Concentration of Diazotrophs expressed as Chlorophyll in Seawater |
| | chlcalc[‡] | kg m$^{-3}$ | XYZ | 2 | Mass Concentration of Calcareous Phytoplankton expressed as Chlorophyll in Seawater |
| | chlpico[§] | kg m$^{-3}$ | XYZ | 2 | Mass Concentration of Picophytoplankton expressed as Chlorophyll in sea water |
| | chlmisc[¶] | kg m$^{-3}$ | XYZ | 2 | Mass Concentration of Other Phytoplankton expressed as Chlorophyll in sea water |
| | pon | mol m$^{-3}$ | XYZ | 2 | Mole Concentration of Particulate Organic Matter expressed as Nitrogen in sea water |
| | pop | mol m$^{-3}$ | XYZ | 2 | Mole Concentration of Particulate Organic Matter expressed as Phosphorus in sea water |
| | bfe[1] | mol m$^{-3}$ | XYZ | 2 | Mole Concentration of Particulate Organic Matter expressed as Iron in sea water |
| | bsi[2] | mol m$^{-3}$ | XYZ | 2 | Mole Concentration of Particulate Organic Matter expressed as silicon in sea water |
| | phyn | mol m$^{-3}$ | XYZ | 2 | Mole Concentration of Total Phytoplankton expressed as Nitrogen in sea water |
| | phyp | mol m$^{-3}$ | XYZ | 2 | Mole Concentration of Total Phytoplankton expressed as Phosphorus in sea water |
| | phyfe | mol m$^{-3}$ | XYZ | 2 | Mole Concentration of Total Phytoplankton expressed as Iron in sea water |
| | physi | mol m$^{-3}$ | XYZ | 2 | Mole Concentration of Total Phytoplankton expressed as Silicon in sea water |
| DMS | dms | mol m$^{-3}$ | XYZ | 2 | Mole Concentration of Dimethyl Sulphide in sea water |
| $[CO_3^{2-}]$ | co3 | mol m$^{-3}$ | XYZ | 2 | Carbonate ion Concentration |
| $[CO_3^{2-}]^{nat}$ | co3nat | mol m$^{-3}$ | XYZ | 2 | Natural Carbonate ion Concentration |
| $[CO_3^{2-}]^{abio}$ | co3abio | mol m$^{-3}$ | XYZ | 2 | Abiotic Carbonate ion Concentration |
| $[CO_3^{2-}]_{sat}^{calc}$ | co3satcalc | mol m$^{-3}$ | XYZ | 2 | Carbonate ion Concentration for sea water in equilibrium with pure Calcite |
| $[CO_3^{2-}]_{sat}^{arag}$ | co3satarag | mol m$^{-3}$ | XYZ | 2 | Carbonate ion Concentration for sea water in equilibrium with pure Aragonite |

[*] chlorophyll from diatom phytoplankton component concentration alone

[†] chlorophyll concentration from the diazotrophic phytoplankton component alone

[‡] chlorophyll concentration from the calcite-producing phytoplankton component alone

[§] chlorophyll concentration from the picophytoplankton ($<2\ \mu$m) component alone

[¶] chlorophyll from additional phytoplankton component concentrations alone

[1] sum of particulate organic iron component concentrations

[2] sum of particulate silica component concentrations

**Table 7.** Annual-mean biogeochemical output: Priority 2 (rates).

| Variable name | Units | Shape | Priority | Long name |
|---|---|---|---|---|
| pp | mol m$^{-3}$ s$^{-1}$ | XYZ | 2 | Primary Carbon Production by Total Phytoplankton |
| pnitrate | mol m$^{-3}$ s$^{-1}$ | XYZ | 2 | Primary Carbon Production by Phytoplankton due to Nitrate Uptake Alone |
| pbfe | mol m$^{-3}$ s$^{-1}$ | XYZ | 2 | Biogenic Iron Production |
| pbsi | mol m$^{-3}$ s$^{-1}$ | XYZ | 2 | Biogenic Silica Production |
| pcalc | mol m$^{-3}$ s$^{-1}$ | XYZ | 2 | Calcite Production |
| parag | mol m$^{-3}$ s$^{-1}$ | XYZ | 2 | Aragonite Production |
| expc | mol m$^{-2}$ s$^{-1}$ | XYZ | 2 | Sinking Particulate Organic Carbon Flux |
| expn | mol m$^{-2}$ s$^{-1}$ | XYZ | 2 | Sinking Particulate Organic Nitrogen Flux |
| expp | mol m$^{-2}$ s$^{-1}$ | XYZ | 2 | Sinking Particulate Organic Phosphorus Flux |
| expfe | mol m$^{-2}$ s$^{-1}$ | XYZ | 2 | Sinking Particulate Iron Flux |
| expsi | mol m$^{-2}$ s$^{-1}$ | XYZ | 2 | Sinking Particulate Silica Flux |
| expcalc | mol m$^{-2}$ s$^{-1}$ | XYZ | 2 | Sinking Calcite Flux |
| exparag | mol m$^{-2}$ s$^{-1}$ | XYZ | 2 | Sinking Aragonite Flux |
| remoc | mol m$^{-3}$ s$^{-1}$ | XYZ | 2 | Remineralization of Organic Carbon |
| dcalc | mol m$^{-3}$ s$^{-1}$ | XYZ | 2 | Calcite Dissolution |
| darag | mol m$^{-3}$ s$^{-1}$ | XYZ | 2 | Aragonite Dissolution |
| ppdiat | mol m$^{-3}$ s$^{-1}$ | XYZ | 2 | Diatom Primary Carbon Production |

**Table 8.** Annual-mean biogeochemical output: Priority 3.

| Variable name | Units | Shape | Priority | Long name |
|---|---|---|---|---|
| bacc | mol m$^{-3}$ | XYZ | 3 | Bacterial Carbon Concentration |
| phydiat | mol m$^{-3}$ | XYZ | 3 | Mole Concentration of Diatoms expressed as Carbon in sea water |
| phydiaz | mol m$^{-3}$ | XYZ | 3 | Mole Concentration of Diazotrophs Expressed as Carbon in sea water |
| phycalc | mol m$^{-3}$ | XYZ | 3 | Mole Concentration of Calcareous Phytoplankton expressed as Carbon in sea water |
| phypico[*] | mol m$^{-3}$ | XYZ | 3 | Mole Concentration of Picophytoplankton expressed as Carbon in sea water |
| phymisc[†] | mol m$^{-3}$ | XYZ | 3 | Mole Concentration of Miscellaneous Phytoplankton expressed as Carbon in sea water |
| zmicro[‡] | mol m$^{-3}$ | XYZ | 3 | Mole Concentration of Microzooplankton expressed as Carbon in sea water |
| zmeso[§] | mol m$^{-3}$ | XYZ | 3 | Mole Concentration of Mesozooplankton expressed as Carbon in sea water |
| zmisc[¶] | mol m$^{-3}$ | XYZ | 3 | Mole Concentration of Other Zooplankton expressed as Carbon in sea water |
| dpocdtdiaz | mol m$^{-3}$ s$^{-1}$ | XYZ | 3 | Tendency of Mole Concentration of Organic Carbon in sea water due to NPP by Diazotrophs |
| dpocdtcalc | mol m$^{-3}$ s$^{-1}$ | XYZ | 3 | Tendency of Mole Concentration of Organic Carbon in sea water due to NPP by Calcareous Phytoplankton |
| dpocdtpico | mol m$^{-3}$ s$^{-1}$ | XYZ | 3 | Tendency of Mole Concentration of Organic Carbon in sea water due to NPP by Picophytoplankton |
| ppdiat | mol m$^{-3}$ s$^{-1}$ | XYZ | 3 | Net Primary Organic Carbon Production by Diatoms |
| ppdiaz | mol m$^{-3}$ s$^{-1}$ | XYZ | 3 | Net Primary Mole Productivity of Carbon by Diazotrophs |
| ppcalc | mol m$^{-3}$ s$^{-1}$ | XYZ | 3 | Net Primary Mole Productivity of Carbon by Calcareous Phytoplankton |
| pppico | mol m$^{-3}$ s$^{-1}$ | XYZ | 3 | Net Primary Mole Productivity of Carbon by Picophytoplankton |
| ppmisc | mol m$^{-3}$ s$^{-1}$ | XYZ | 3 | Net Primary Organic Carbon Production by Other Phytoplankton |
| bddtdic | mol m$^{-3}$ s$^{-1}$ | XYZ | 3 | Rate of Change of Dissolved Inorganic Carbon due to Biological Activity |
| bddtdin | mol m$^{-3}$ s$^{-1}$ | XYZ | 3 | Rate of Change of Nitrogen Nutrient due to Biological Activity |
| bddtdip | mol m$^{-3}$ s$^{-1}$ | XYZ | 3 | Rate of Change of Dissolved Phosphorus due to Biological Activity |
| bddtdife | mol m$^{-3}$ s$^{-1}$ | XYZ | 3 | Rate of Change of Dissolved Inorganic Iron due to Biological Activity |
| bddtdisi | mol m$^{-3}$ s$^{-1}$ | XYZ | 3 | Rate of Change of Total Dissolved Inorganic Silicon due to Biological Activity |
| bddtalk | mol m$^{-3}$ s$^{-1}$ | XYZ | 3 | Rate of Change of Alkalinity due to Biological Activity |
| fescav | mol m$^{-3}$ s$^{-1}$ | XYZ | 3 | Nonbiogenic Iron Scavenging |
| fediss | mol m$^{-3}$ s$^{-1}$ | XYZ | 3 | Particle Source of Dissolved Iron |
| graz | mol m$^{-3}$ s$^{-1}$ | XYZ | 3 | Total Grazing of Phytoplankton by Zooplankton |

[*] carbon concentration from the picophytoplankton ($<2$ $\mu$m) component alone

[†] carbon concentration from additional phytoplankton component alone

[‡] carbon concentration from the microzooplankton ($<20$ $\mu$m) component alone

[§] carbon concentration from mesozooplankton (20–200 $\mu$m) component alone

[¶] carbon from additional zooplankton component concentrations alone (e.g. micro, meso). Provides check for model intercomparison since some phytoplankton groups are supersets.

**Table 9.** Monthly mean biogeochemical output: Priority 1.

| Symbol | Variable name | Units | Shape | Priority | Long name |
|---|---|---|---|---|---|
| | dissicos | mol m$^{-3}$ | XY | 1 | Surface Dissolved Inorganic Carbon Concentration |
| | dissicnatos | mol m$^{-3}$ | XY | 1 | Surface Natural Dissolved Inorganic Carbon Concentration |
| | dissicabioos | mol m$^{-3}$ | XY | 1 | Surface Abiotic Dissolved Inorganic Carbon Concentration |
| | dissi14cabioos | mol m$^{-3}$ | XY | 1 | Surface Abiotic Dissolved Inorganic 14Carbon Concentration |
| | dissi13cos | mol m$^{-3}$ | XY | 1 | Surface Dissolved Inorganic 13Carbon Concentration |
| | talkos | mol m$^{-3}$ | XY | 1 | Surface Total Alkalinity |
| | talknatos | mol m$^{-3}$ | XY | 1 | Surface Natural Total Alkalinity |
| | phos | 1 | XY | 1 | Surface pH on total scale |
| | sios | mol m$^{-3}$ | XY | 1 | Surface Total Dissolved Inorganic Silicon Concentration |
| | o2os | mol m$^{-3}$ | XY | 1 | Surface Dissolved Oxygen Concentration |
| | o2satos | mol m$^{-3}$ | XY | 1 | Surface Dissolved Oxygen Concentration at Saturation |
| | po4os | mol m$^{-3}$ | XY | 1 | Surface Total Dissolved Inorganic Phosphorus Concentration |
| | chlos | kg m$^{-3}$ | XY | 1 | Surface Mass Conc. of Total Phytoplankton expressed as Chlorophyll in sea water |
| $C_{\mathrm{T}}$ | dissic | mol m$^{-3}$ | XYZ | 1 | Dissolved Inorganic Carbon Concentration |
| $A_{\mathrm{T}}$ | talk | mol m$^{-3}$ | XYZ | 1 | Total Alkalinity |
| pH | ph | 1 | XYZ | 1 | pH on total scale |
| $P_{\mathrm{T}}$ | po4$^a$ | mol m$^{-3}$ | XYZ | 1 | Total Dissolved Inorganic Phosphorus Concentration |
| | intpp$^*$ | mol m$^{-2}$ s$^{-1}$ | XY | 1 | Primary Organic Carbon Production by All Types of Phytoplankton |
| | expc100$^\dagger$ | mol m$^{-2}$ s$^{-1}$ | XY | 1 | Downward Flux of Particle Organic Carbon |
| | expcalc100$^\dagger$ | mol m$^{-2}$ s$^{-1}$ | XY | 1 | Downward Flux of Calcite |
| | exparag100$^\dagger$ | mol m$^{-2}$ s$^{-1}$ | XY | 1 | Downward Flux of Aragonite |
| $p\mathrm{CO}_2$ | spco2 | Pa | XY | 1 | Surface Aqueous Partial Pressure of CO2 |
| $p\mathrm{CO}_2^{\mathrm{nat}}$ | spco2nat | Pa | XY | 1 | Natural Surface Aqueous Partial Pressure of CO2 |
| $p\mathrm{CO}_2^{\mathrm{abio}}$ | spco2abio | Pa | XY | 1 | Abiotic Surface Aqueous Partial Pressure of CO2 |
| $F_{\mathrm{CO}_2^{\mathrm{tot}}}$ | fgco2 | kg m$^{-2}$ s$^{-1}$ | XY | 1 | Surface Downward Flux of Total CO2 |
| $F_{\mathrm{CO}_2^{\mathrm{nat}}}$ | fgco2nat | kg m$^{-2}$ s$^{-1}$ | XY | 1 | Surface Downward Flux of Natural CO2 |
| $F_{\mathrm{CO}_2^{\mathrm{abio}}}$ | fgco2abio | kg m$^{-2}$ s$^{-1}$ | XY | 1 | Surface Downward Flux of Abiotic CO2 |
| $F_{^{14}\mathrm{CO}_2^{\mathrm{abio}}}$ | fg14co2abio | kg m$^{-2}$ s$^{-1}$ | XY | 1 | Surface Downward Flux of Abiotic 14CO2 |
| $F_{^{14}\mathrm{CO}_2}$ | fg13co2 | kg m$^{-2}$ s$^{-1}$ | XY | 1 | Surface Downward Flux of 13CO2 |
| $F_{\mathrm{O}_2}$ | fgo2 | mol m$^{-2}$ s$^{-1}$ | XY | 1 | Surface Downward Flux of O2 |

$^a$ For models that do not carry $P_{\mathrm{T}}$ as a tracer, compute it from NO$_3^-$ assuming N:P = 16:1

$^*$ Vertically integrated total primary (organic carbon) production by phytoplankton. This should equal the sum of intpdiat+intpphymisc, but those individual components may be unavailable in some models.

$^\dagger$ at 100-m depth

**Table 10.** Monthly mean biogeochemical output: Priority 2 (2-D fields).

| Symbol | Variable name | Units | Shape | Priority | Long name |
|---|---|---|---|---|---|
| | dissocos | mol m$^{-3}$ | XY | 2 | Surface Dissolved Organic Carbon Concentration |
| | phycos | mol m$^{-3}$ | XY | 2 | Surface Phytoplankton Carbon Concentration |
| | zoocos | mol m$^{-3}$ | XY | 2 | Surface Zooplankton Carbon Concentration |
| | detocos | mol m$^{-3}$ | XY | 2 | Surface Detrital Organic Carbon Concentration |
| | calcos | mol m$^{-3}$ | XY | 2 | Surface Calcite Concentration |
| | aragos | mol m$^{-3}$ | XY | 2 | Surface Aragonite Concentration |
| | phnatos | 1 | XY | 2 | Surface Natural pH on total scale |
| | phabioos | 1 | XY | 2 | Surface Abiotic pH on total scale |
| | no3os | mol m$^{-3}$ | XY | 2 | Surface Dissolved Nitrate Concentration |
| | nh4os | mol m$^{-3}$ | XY | 2 | Surface Dissolved Ammonium Concentration |
| | dfeos | mol m$^{-3}$ | XY | 2 | Surface Dissolved Iron Concentration |
| | co3os | mol m$^{-3}$ | XY | 2 | Surface Carbonate ion Concentration |
| | co3natos | mol m$^{-3}$ | XY | 2 | Surface Natural Carbonate ion Concentration |
| | co3abioos | mol m$^{-3}$ | XY | 2 | Surface Abiotic Carbonate ion Concentration |
| | co3satcalcos | mol m$^{-3}$ | XY | 2 | Surface Carbonate ion Conc. for Seawater in equilibrium with pure Calcite |
| | co3sataragos | mol m$^{-3}$ | XY | 2 | Surface Carbonate ion Conc. for Seawater in equilibrium with pure Aragonite |
| | limndiat[1] | 1 | XY | 2 | Nitrogen limitation of Diatoms |
| | limirrdiat[1] | 1 | XY | 2 | Irradiance limitation of Diatoms |
| | limfediat[1] | 1 | XY | 2 | Iron limitation of Diatoms |
| | intppnitrate[*] | mol m$^{-2}$ s$^{-1}$ | XY | 2 | Primary Organic Carbon Production by Phytoplankton Based on Nitrate Uptake Alone |
| | intppdiat[†] | mol m$^{-2}$ s$^{-1}$ | XY | 2 | Primary Organic Carbon Production by Diatoms |
| $\int C_{\mathrm{T}}\,dz$ | intdic[‡] | kg m$^{-2}$ | XY | 2 | Dissolved Inorganic Carbon Content |
| $\int DOC\,dz$ | intdoc[§] | kg m$^{-2}$ | XY | 2 | Dissolved Organic Carbon Content |
| $\int POC\,dz$ | intpoc[¶] | kg m$^{-2}$ | XY | 2 | Particulate Organic Carbon Content |

[1] These 2-D limitation terms should be calculated as the carbon biomass weighted average for the upper 100 m

[*] Vertically integrated primary (organic carbon) production by phytoplankton based on nitrate uptake alone

[†] Vertically integrated primary (organic carbon) production by the diatom phytoplankton component alone

[‡] Vertically integrated $C_{\mathrm{T}}$

[§] Vertically integrated DOC (explicit pools only)

[¶] Vertically integrated POC

**Table 11.** Monthly mean biogeochemical output: Priority 2 (3-D fields).

| Symbol | Variable name | Units | Shape | Priority | Long name |
|---|---|---|---|---|---|
| $C_T^{nat}$ | dissicnat | mol m$^{-3}$ | XYZ | 2 | Natural Dissolved Inorganic Carbon Concentration |
| $C_T^{abio}$ | dissicabio | mol m$^{-3}$ | XYZ | 2 | Abiotic Dissolved Inorganic Carbon Concentration |
| $^{14}C_T^{abio}$ | dissi14cabio | mol m$^{-3}$ | XYZ | 2 | Abiotic Dissolved Inorganic 14Carbon Concentration |
| $^{13}C_T$ | dissi13c | mol m$^{-3}$ | XYZ | 2 | Dissolved Inorganic 13Carbon Concentration |
| $A_T^{nat}$ | talknat | mol m$^{-3}$ | XYZ | 2 | Natural Total Alkalinity |
| pH$^{nat}$ | phnat | 1 | XYZ | 2 | Natural pH |
| pH$^{abio}$ | phabio | 1 | XYZ | 2 | Abiotic pH |
| $[O_2]$ | o2 | mol m$^{-3}$ | XYZ | 2 | Dissolved Oxygen Concentration |
|  | o2sat | mol m$^{-3}$ | XYZ | 2 | Dissolved Oxygen Concentration at Saturation |
| $[NO_3^-]$ | no3 | mol m$^{-3}$ | XYZ | 2 | Dissolved Nitrate Concentration |
| $[NH_4^+]$ | nh4 | mol m$^{-3}$ | XYZ | 2 | Dissolved Ammonium Concentration |
| Fe$^{\ddagger}$ | dfe | mol m$^{-3}$ | XYZ | 2 | Dissolved Iron Concentration |
| $Si_T$ | si | mol m$^{-3}$ | XYZ | 2 | Total Dissolved Inorganic Silicon Concentration |
| Chl | chl | kg m$^{-3}$ | XYZ | 2 | Mass Concentration of Total Phytoplankton expressed as Chlorophyll in sea water |
| DOC | dissoc | mol m$^{-3}$ | XYZ | 2 | Dissolved Organic Carbon Concentration |
|  | phyc | mol m$^{-3}$ | XYZ | 2 | Phytoplankton Carbon Concentration |
|  | zooc | mol m$^{-3}$ | XYZ | 2 | Zooplankton Carbon Concentration |
|  | detoc | mol m$^{-3}$ | XYZ | 2 | Detrital Organic Carbon Concentration |
| $[CaCO_3]_{calc}$ | calc | mol m$^{-3}$ | XYZ | 2 | Calcite Concentration |
| $[CaCO_3]_{arag}$ | arag | mol m$^{-3}$ | XYZ | 2 | Aragonite Concentration |
| $[CO_3^{2-}]$ | co3 | mol m$^{-3}$ | XYZ | 2 | Carbonate ion Concentration |
| $[CO_3^{2-}]^{nat}$ | co3nat | mol m$^{-3}$ | XYZ | 2 | Natural Carbonate ion Concentration |
| $[CO_3^{2-}]^{abio}$ | co3abio | mol m$^{-3}$ | XYZ | 2 | Abiotic Carbonate ion Concentration |
| $[CO_3^{2-}]_{sat}^{calc}$ | co3satcalc | mol m$^{-3}$ | XYZ | 2 | Carbonate ion Concentration for Seawater in equilibrium with pure Calcite |
| $[CO_3^{2-}]_{sat}^{arag}$ | co3satarag | mol m$^{-3}$ | XYZ | 2 | Carbonate ion Concentration for Seawater in equilibrium with pure Aragonite |

**Table 12.** Monthly mean biogeochemical output: Priority 3 (concentrations of surface fields)

| Variable name | Units | Shape | Priority | Long name |
|---|---|---|---|---|
| baccos | mol m$^{-3}$ | XY | 3 | Surface Bacterial Carbon Concentration |
| phydiatos | mol m$^{-3}$ | XY | 3 | Surface Mole Concentration of Diatoms expressed as Carbon in sea water |
| phydiazos | mol m$^{-3}$ | XY | 3 | Surface Mole Concentration of Diazotrophs Expressed as Carbon in sea water |
| phycalcos | mol m$^{-3}$ | XY | 3 | Surface Mole Concentration of Calcareous Phytoplankton expressed as Carbon in sea water |
| phypicoos | mol m$^{-3}$ | XY | 3 | Surface Mole Concentration of Picophytoplankton expressed as Carbon in sea water |
| phymiscos | mol m$^{-3}$ | XY | 3 | Surface Mole Concentration of Miscellaneous Phytoplankton expressed as Carbon in sea water |
| zmicroos | mol m$^{-3}$ | XY | 3 | Surface Mole Concentration of Microzooplankton expressed as Carbon in sea water |
| zmesoos | mol m$^{-3}$ | XY | 3 | Surface Mole Concentration of Mesozooplankton expressed as Carbon in sea water |
| zmiscos | mol m$^{-3}$ | XY | 3 | Surface Mole Concentraiton of Other Zooplankton expressed as Carbon in sea water |
| chldiatos | kg m$^{-3}$ | XY | 3 | Surface Mass Concentration of Diatoms expressed as Chlorophyll in sea water |
| chldiazos | kg m$^{-3}$ | XY | 3 | Surface Mass Concentration of Diazotrophs expressed as Chlorophyll in sea water |
| chlcalcos | kg m$^{-3}$ | XY | 3 | Surface Mass Concentration of Calcareous Phytoplankton expressed as Chlorophyll in sea water |
| chlpicoos | kg m$^{-3}$ | XY | 3 | Surface Mass Concentration of Picophytoplankton expressed as Chlorophyll in sea water |
| chlmiscos | kg m$^{-3}$ | XY | 3 | Surface Mass Concentration of Other Phytoplankton expressed as Chlorophyll in sea water |
| ponos | mol m$^{-3}$ | XY | 3 | Surface Mole Concentration of Particulate Organic Matter expressed as Nitrogen in sea water |
| popos | mol m$^{-3}$ | XY | 3 | Surface Mole Concentration of Particulate Organic Matter expressed as Phosphorus in sea water |
| bfeos | mol m$^{-3}$ | XY | 3 | Surface Mole Concentration of Particulate Organic Matter expressed as Iron in sea water |
| bsios | mol m$^{-3}$ | XY | 3 | Surface Mole Concentration of Particulate Organic Matter expressed as Silicon in sea water |
| phynos | mol m$^{-3}$ | XY | 3 | Surface Mole Concentration of Phytoplankton Nitrogen in sea water |
| phypos | mol m$^{-3}$ | XY | 3 | Surface Mole Concentration of Total Phytoplankton expressed as Phosphorus in sea water |
| phyfeos | mol m$^{-3}$ | XY | 3 | Surface Mass Concentration of Diazotrophs expressed as Chlorophyll in sea water |
| physios | mol m$^{-3}$ | XY | 3 | Surface Mole Concentration of Total Phytoplankton expressed as Silicon in sea water |
| dmsos | mol m$^{-3}$ | XY | 3 | Surface Mole Concentration of Dimethyl Sulphide in sea water |

**Table 13.** Monthly mean biogeochemical output: Priority 3 (concentrations of 3-D fields)

| Variable name | Units | Shape | Priority | Long name |
| --- | --- | --- | --- | --- |
| bacc | mol m$^{-3}$ | XYZ | 3 | Bacterial Carbon Concentration |
| phydiat | mol m$^{-3}$ | XYZ | 3 | Mole Concentration of Diatoms expressed as Carbon in sea water |
| phydiaz | mol m$^{-3}$ | XYZ | 3 | Mole Concentration of Diazotrophs Expressed as Carbon in sea water |
| phycalc | mol m$^{-3}$ | XYZ | 3 | Mole Concentration of Calcareous Phytoplankton expressed as Carbon in sea water |
| phypico | mol m$^{-3}$ | XYZ | 3 | Mole Concentration of Picophytoplankton expressed as Carbon in sea water |
| phymisc | mol m$^{-3}$ | XYZ | 3 | Mole Concentration of Miscellaneous Phytoplankton expressed as Carbon in sea water |
| zmicro | mol m$^{-3}$ | XYZ | 3 | Mole Concentration of Microzooplankton expressed as Carbon in sea water |
| zmeso | mol m$^{-3}$ | XYZ | 3 | Mole Concentration of Mesozooplankton expressed as Carbon in sea water |
| zmisc | mol m$^{-3}$ | XYZ | 3 | Mole Concentraiton of Other Zooplankton expressed as Carbon in sea water |
| chldiat | kg m$^{-3}$ | XYZ | 3 | Mass Concentration of Diatoms expressed as Chlorophyll in sea water |
| chldiaz | kg m$^{-3}$ | XYZ | 3 | Mass Concentration of Diazotrophs expressed as Chlorophyll in sea water |
| chlcalc | kg m$^{-3}$ | XYZ | 3 | Mass Concentration of Calcareous Phytoplankton expressed as Chlorophyll in sea water |
| chlpico | kg m$^{-3}$ | XYZ | 3 | Mass Concentration of Picophytoplankton expressed as Chlorophyll in sea water |
| chlmisc | kg m$^{-3}$ | XYZ | 3 | Mass Concentration of Other Phytoplankton expressed as Chlorophyll in sea water |
| pon | mol m$^{-3}$ | XYZ | 3 | Mole Concentration of Particulate Organic Matter expressed as Nitrogen in sea water |
| pop | mol m$^{-3}$ | XYZ | 3 | Mole Concentration of Particulate Organic Matter expressed as Phosphorus in sea water |
| bfe | mol m$^{-3}$ | XYZ | 3 | Mole Concentration of Particulate Organic Matter expressed as Iron in sea water |
| bsi | mol m$^{-3}$ | XYZ | 3 | Mole Concentration of Particulate Organic Matter expressed as Silicon in sea water |
| phyn | mol m$^{-3}$ | XYZ | 3 | Mole Concentration of Phytoplankton Nitrogen in sea water |
| phyp | mol m$^{-3}$ | XYZ | 3 | Mole Concentration of Total Phytoplankton expressed as Phosphorus in sea water |
| phyfe | mol m$^{-3}$ | XYZ | 3 | Mass Concentration of Diazotrophs expressed as Chlorophyll in sea water |
| physi | mol m$^{-3}$ | XYZ | 3 | Mole Concentration of Total Phytoplankton expressed as Silicon in sea water |
| dmso | mol m$^{-3}$ | XYZ | 3 | Mole Concentration of Dimethyl Sulphide in sea water |

**Table 14.** Monthly mean biogeochemical output: Priority 3 (gas exchange, river, burial, $N_2$ fixation, thresholds)

| Symbol | Variable name | Units | Shape | Priority | Long name |
|---|---|---|---|---|---|
| $\Delta p\text{CO}_2$ | dpco2[*] | Pa | XY | 3 | Delta PCO2 |
| $\Delta p\text{CO}_2^{\text{nat}}$ | dpco2nat[*] | Pa | XY | 3 | Natural Delta PCO2 |
| $\Delta p\text{CO}_2^{\text{abio}}$ | dpco2abio[*] | Pa | XY | 3 | Abiotic Delta PCO2 |
| $\Delta p\text{O}_2$ | dpo2[†] | Pa | XY | 3 | Delta PO2 |
| $F_{\text{DMS}}$ | fgdms | mol m$^{-2}$ s$^{-1}$ | XY | 3 | Surface Upward Flux of DMS |
| | icfriver | mol m$^{-2}$ s$^{-1}$ | XY | 3 | Flux of Inorganic Carbon Into Ocean Surface by Runoff |
| | fric | mol m$^{-2}$ s$^{-1}$ | XY | 3 | Downward Inorganic Carbon Flux at Ocean Bottom |
| | ocfriver | mol m$^{-2}$ s$^{-1}$ | XY | 3 | Flux of Organic Carbon Into Ocean Surface by Runoff |
| | froc | mol m$^{-2}$ s$^{-1}$ | XY | 3 | Downward Organic Carbon Flux at Ocean Bottom |
| | intpn2 | mol m$^{-2}$ s$^{-1}$ | XY | 3 | Nitrogen Fixation Rate in Ocean |
| | fsn | mol m$^{-2}$ s$^{-1}$ | XY | 3 | Surface Downward Net Flux of Nitrogen |
| | frn | mol m$^{-2}$ s$^{-1}$ | XY | 3 | Nitrogen Loss to Sediments and through Denitrification |
| | fsfe | mol m$^{-2}$ s$^{-1}$ | XY | 3 | Surface Downward Net Flux of Iron |
| | frfe | mol m$^{-2}$ s$^{-1}$ | XY | 3 | Iron Loss to Sediments |
| | o2min | mol m$^{-3}$ | XY | 3 | Oxygen Minimum Concentration |
| | zo2min | m | XY | 3 | Depth of Oxygen Minimum Concentration |
| CSH | zsatcalc[‡] | m | XY | 3 | Calcite Saturation Depth |
| ASH | zsatarag[§] | m | XY | 3 | Aragonite Saturation Depth |

[*] Difference between atmospheric and oceanic partial pressure of $CO_2$ (positive meaning ocean > atmosphere)

[†] Difference between atmospheric and oceanic partial pressure of $O_2$ (positive meaning ocean > atmosphere)

[‡] Depth of calcite saturation horizon (0 if < surface, "missing" if > bottom, if two, then the shallower)

[§] Depth of aragonite saturation horizon (0 if < surface, "missing" if > bottom, if two, then the shallower)

**Table 15.** Monthly mean biogeochemical output: Priority 3 (production and rates of change)

| Variable name | Units | Shape | Priority | Long name |
|---|---|---|---|---|
| expn100[*] | mol m$^{-2}$ s$^{-1}$ | XY | 3 | Downward Flux of Particulate Nitrogen |
| expp100[*] | mol m$^{-2}$ s$^{-1}$ | XY | 3 | Downward Flux of Particulate Phosphorus |
| expfe100[*] | mol m$^{-2}$ s$^{-1}$ | XY | 3 | Downward Flux of Particulate Iron |
| expsi100[*] | mol m$^{-2}$ s$^{-1}$ | XY | 3 | Downward Flux of Particulate Silica |
| fddtdic[†] | mol m$^{-2}$ s$^{-1}$ | XY | 3 | Rate of Change of Net Dissolved Inorganic Carbon |
| fddtdin[†,‡] | mol m$^{-2}$ s$^{-1}$ | XY | 3 | Rate of Change of Net Dissolved Inorganic Nitrogen |
| fddtdip[†] | mol m$^{-2}$ s$^{-1}$ | XY | 3 | Rate of Change of Net Dissolved Inorganic Phosphorus |
| fddtdife[†] | mol m$^{-2}$ s$^{-1}$ | XY | 3 | Rate of Change of Net Dissolved Inorganic Iron |
| fddtdisi[†] | mol m$^{-2}$ s$^{-1}$ | XY | 3 | Rate of Change of Net Dissolved Inorganic Silicon |
| fddtalk[†] | mol m$^{-2}$ s$^{-1}$ | XY | 3 | Rate of Change of Total Alkalinity |
| fbddtdic[†] | mol m$^{-2}$ s$^{-1}$ | XY | 3 | Rate of Change of Dissolved Inorganic Carbon due to Biological Activity |
| fbddtdin[†,§] | mol m$^{-2}$ s$^{-1}$ | XY | 3 | Rate of Change of Dissolved Inorganic Nitrogen due to Biological Activity |
| fbddtdip[†] | mol m$^{-2}$ s$^{-1}$ | XY | 3 | Rate of Change of Total Dissolved Inorganic Phosphorus due to Biological Activity |
| fbddtdife[†] | mol m$^{-2}$ s$^{-1}$ | XY | 3 | Rate of Change of Dissolved Inorganic Iron due to Biological Activity |
| fbddtdisi[†] | mol m$^{-2}$ s$^{-1}$ | XY | 3 | Rate of Change of Total Dissolved Inorganic Silicon due to Biological Activity |

[*] at 100-m depth

[†] integral over upper 100 m only

[‡] Net time rate of change of nitrogen nutrients (e.g. $NO_3^- + NH_4^+$)

[§] Vertical integral of net biological terms in time rate of change of nitrogen nutrients (e.g. $NO_3^- + NH_4^+$)

**Table 16.** Monthly mean biogeochemical output: Priority 3 (production, grazing, sinking, limitation)

| Variable name | Units | Shape | Priority | Long name |
|---|---|---|---|---|
| pp | mol m$^{-3}$ s$^{-1}$ | XYZ | 3 | Primary Carbon Production by Phytoplankton |
| graz | mol m$^{-3}$ s$^{-1}$ | XYZ | 3 | Total Grazing of Phytoplankton by Zooplankton |
| expc | mol m$^{-2}$ s$^{-1}$ | XY | 3 | Sinking Particulate Organic Carbon Flux |
| limndiaz | 1 | XY | 3 | Nitrogen limitation of Diazotrophs |
| limncalc | 1 | XY | 3 | Nitrogen limitation of Calcareous Phytoplankton |
| limnpico | 1 | XY | 3 | Nitrogen limitation of Picophytoplankton |
| limnmisc | 1 | XY | 3 | Nitrogen Limitation of Other Phytoplankton |
| limirrdiaz | 1 | XY | 3 | Irradiance limitation of Diazotrophs |
| limirrcalc | 1 | XY | 3 | Irradiance limitation of Calcareous Phytoplankton |
| limirrpico | 1 | XY | 3 | Irradiance limitation of Picophytoplankton |
| limirrmisc | 1 | XY | 3 | Irradiance Limitation of Other Phytoplankton |
| limfediaz | 1 | XY | 3 | Iron limitation of Diazotrophs |
| limfecalc | 1 | XY | 3 | Iron limitation of Calcareous Phytoplankton |
| limfepico | 1 | XY | 3 | Iron limitation of Picophytoplankton |
| limfemisc | 1 | XY | 3 | Iron Limitation of Other Phytoplankton |
| intppdiaz | mol m$^{-2}$ s$^{-1}$ | XY | 3 | Net Primary Mole Productivity of Carbon by Diazotrophs |
| intppcalc | mol m$^{-2}$ s$^{-1}$ | XY | 3 | Net Primary Mole Productivity of Carbon by Calcareous Phytoplankton |
| intpppico | mol m$^{-2}$ s$^{-1}$ | XY | 3 | Net Primary Mole Productivity of Carbon by Picophytoplankton |
| intppmisc | mol m$^{-2}$ s$^{-1}$ | XY | 3 | Net Primary Organic Carbon Production by Other Phytoplankton |
| intpbn | mol m$^{-2}$ s$^{-1}$ | XY | 3 | Nitrogen Production |
| intpbp | mol m$^{-2}$ s$^{-1}$ | XY | 3 | Phosphorus Production |
| intpbfe | mol m$^{-2}$ s$^{-1}$ | XY | 3 | Iron Production |
| intpbsi | mol m$^{-2}$ s$^{-1}$ | XY | 3 | Silica Production |
| intpcalcite | mol m$^{-2}$ s$^{-1}$ | XY | 3 | Calcite Production |
| intparag | mol m$^{-2}$ s$^{-1}$ | XY | 3 | Aragonite Production |

**Table 17.** Daily mean biogeochemical output.

| Variable name | Units | Shape | Priority | Long name |
|---|---|---|---|---|
| chlos | kg m$^{-3}$ | XY | 3 | Surface Mass Conc. of Total Phytoplankton expressed as Chlorophyll seawater |
| phycos | mol m$^{-3}$ | XY | 3 | Surface Phytoplankton Carbon Concentration |

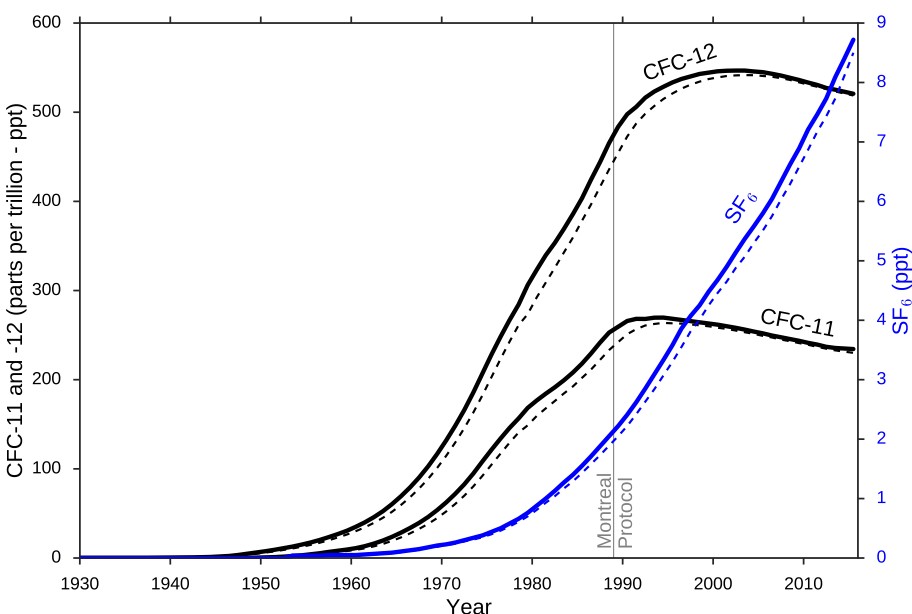

**Figure 1.** Histories of annual-mean tropospheric mixing ratios of CFC-11, CFC-12, and $SF_6$ for the northern hemisphere (solid line) and southern hemisphere (dashed line). Mixing ratios are given in parts per trillion (ppt) from mid-year data provided by Bullister (2015). For the OMIP simulations, these inert chemical tracers need not be included until the 4th CORE-II forcing cycle when they will be initialized to zero on 01 January 1936 (at model date 01 January 0237). The vertical grey line indicates the date when the Montreal protocol entered into force.

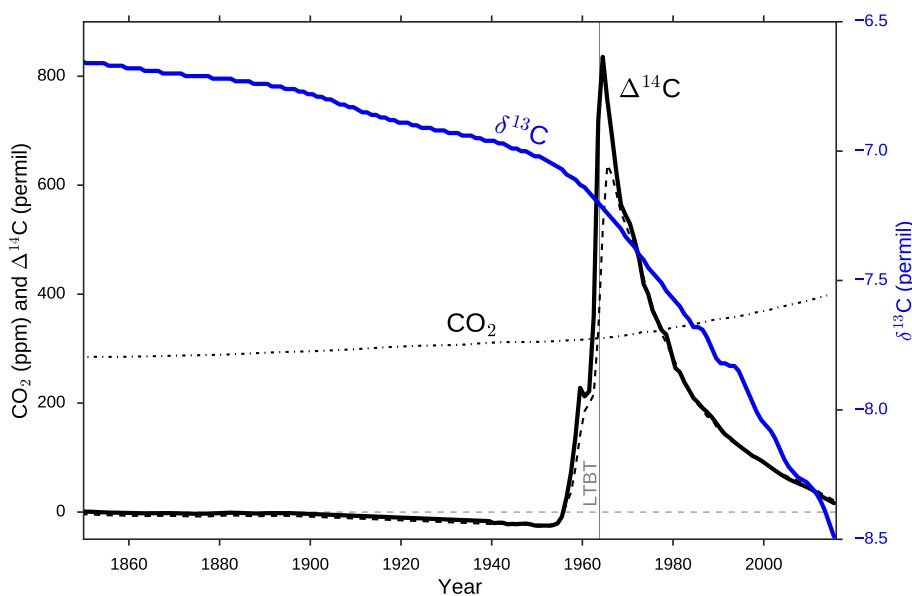

**Figure 2.** Annual-mean atmospheric histories for global-mean $CO_2$ (black dash-dot) and $\delta^{13}C$ (blue) compared to hemispheric means of $\Delta^{14}C$ for the north (black solid) and south (black dashes). The $CO_2$ data are identical to those used for CMIP6 (Meinshausen et al., 2016) and the carbon isotope data are common with C4MIP (Jones et al., 2016). The $CO_2$ observations are from NOAA (Dlugokencky and Tans, 2016) and Scripps Institution of Oceanography (Keeling et al., 2001), and $\delta^{13}C$ is a compilation of ice-core data (Rubino et al., 2013) and atmospheric measurements at Mauna Loa (Keeling et al., 2001). The $\Delta^{14}C$ data is compiled from Levin et al. (2010) and other sources. Data after 2009 are not used in OMIP Phase 1, but will be used in subsequent phases. Beyond 2009, atmospheric $\Delta^{14}C$ is unpublished data from the University of Heidelberg. Between the beginning of both OMIP simulations on 1 January 1700 and the same date in 1850, the atmospherc concentrations of $CO_2$, $\delta^{13}C$, and $\Delta^{14}C$ are to be held constant at are 285.375 ppm, -6.8‰ and 0‰, respectively. Also indicated are the preindustrial reference (0 permil) for atmospheric $\Delta^{14}C$ (horizontal grey dashed) and when the Limited Test Ban Treaty (LTBT) went into effect (vertical grey solid).

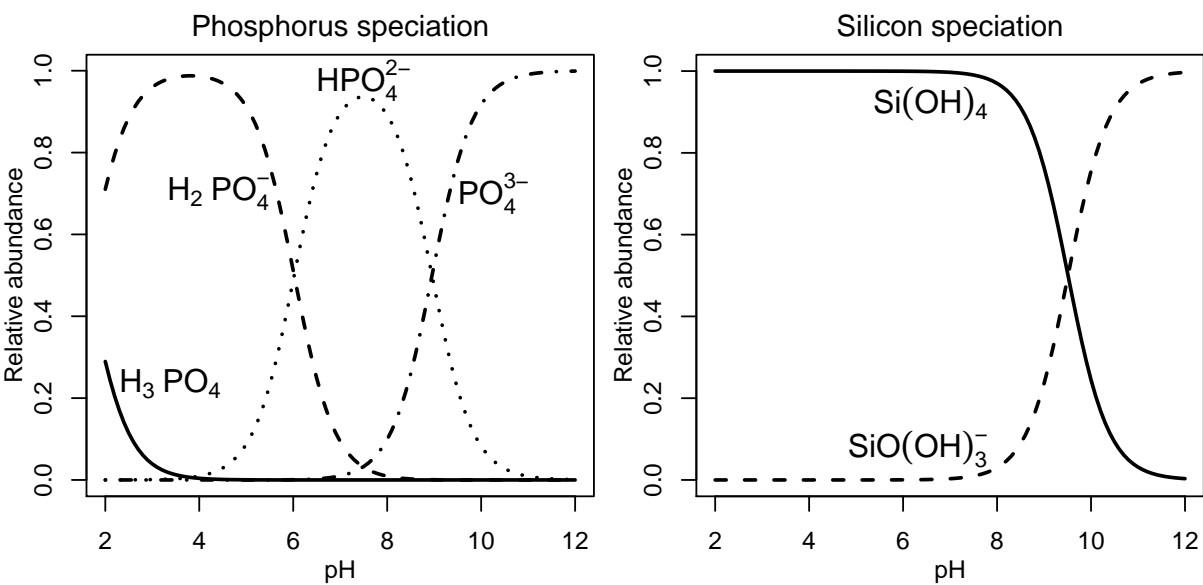

**Figure 3.** Relative molar abundance of inorganic species of phosphorus (left) and silicon (right) as a function of pH (total scale) in seawater at a temperature of 18°C and salinity of 35.