# Peer review of "Biogeochemical protocols and diagnostics for the CMIP6 Ocean Model Intercomparison Project (OMIP)"

_Geoscientific Model Development, 2016_

## Short Comment (SC1) · 8 Sep 2016

Firstly, thanks to the authors for coordinating the OMIP-BGC effort and putting together this very comprehensive documenting paper. I have a few comments, which I think will be relevant to modelling centres trying to perform the simulations and diagnostics you describe:

- For the diagnostic output (section 3 and tables 9 to 14), no indication is given on which variables are expected only for the OMIP-BGC ocean model runs; and which variables are expected from the Earth System Models doing the "full" CMIP6 runs (historical etc), unless I missed this somewhere. It would be good to mark clearly if there are variables expected only for OMIP-BGC runs, but not the rest of CMIP6.

[Figure]

- Table 5 and 9 mark all variables as "priority 1". My understanding of a priority 1 variable is from the CMIP6 data request: "all participating groups must commit to supplying all priority 1 variables". Thus, priority 1 variables should be the lowest common denominator that all groups can provide. In tables 5 and 9, there are multiple variables (like 13C, and all the "abiotic" terms) which are not carried in all (or even most) BGC models. Thus, groups face the significant coding and computation expense of adding 7+ new tracers if they are "required" to provide all these terms to participate in OMIP-BGC. If these variables are not indeed "required", please mark them as priority 2, in which case it is clear they can be provided optionally, if available. Otherwise, as I understanding it, any group not providing all these variables will end up not "participating" [at least officially].

Neil Swart, Canadian Centre for Climate Modelling and Analysis, Victoria, BC, Canada.

---

## Referee Comment (RC1) · Anonymous Referee #1 · 9 Sep 2016

This paper lays out a strategy for the biogeochemistry component (OMIP-BGC) of the Ocean Model Intercomparison Project under the umbrella of the 6th Coupled Model Intercomparison Project (CMIP6). The paper is well written and mostly does a good job of outlining the experimental design for potential participants in clear and unambiguous terms. The promised OMIP-BGC web page (bottom p. 19) does not appear to be operational yet (and rather more than 4 weeks have passed).

Major comments:

Overall strategy and ordering of priorities

The weakest point of the strategy is that it is vague about the priority of experiments.

[Figure]

In CMIP5, there were tiers of experiments (Core/Tier1/Tier2), and tiers of output fields (Priority 1/2/3). This document does not really separate the two, implicitly treating all experiments as Core. The list of Priority 1 output fields is expansive and probably unrealistic.

Some tracers are referred to as "level=1" and "level=2" (5/25), but the term is not defined. It appears to refer to output fields in which case it is synonymous with "Priority" (Tables 4-14) and there is no real conceptual problem. But the most important thing the authors need to do is to separate the x (experiments) and y (output fields) axes in a fashion similar to the CMIP5 data request, and pare down the list of Priority 1 fields to a more realistic level.

The list of 3D monthly fields is long. In CMIP5 no 3D ocean biogeochemistry fields were monthly. According to Moore's Law, computing power should have increased about 16-fold since CMIP5, but in practice the gain is probably much less. Making previously annual fields monthly requires a twelvefold increase just to break even in terms of the time and storage capacity it will take to access and download data, even assuming no increase in model resolution.

For any model that includes feedbacks between saturation state and biology, the duplication of tracers is likely to make the 'natural $CO_2$' experiments prohibitively expensive. Yet these are said in the Conclusion to be critical (19/14) and to be "required" on 6/19. Even if one only considers the effect of saturation state on dissolution of $CaCO_3$, that is a minimum of three additional tracers (or maybe two if $CaCO_3$ is parameterized). I agree with the authors that these experiments are important, but the rather superficial consideration given to the actual cost (bottom p. 6) simply underscores that the strategy does not include a clear hierarchy of priorities for the different experiments proposed.

Alkalinity and speciation

I agree that using a truncated expression for alkalinity causes large errors, but the pa-

per could spell out in detail exactly what they envision alkalinity as including (since the full formal definition includes lots of species that are not provided in a model simulation), rather than simply referring the reader to the provided codes.

Similarly, the authors could clarify exactly what they mean by N speciation (18/14). I agree that the alkalinity sources and sinks associated with biological transformations (e.g., nitrification) of N species should be accounted for. But I think it is better if the word speciation is not used here.

For models that have N but not P it is recommended that the PO4 contribution to alkalinity be calculated as the P/N Redfield ratio times the total inorganic N concentration (16/29), which is appropriate. But then on 18/15-16 this is referred to as the effect of nitrate on alkalinity. But really what is being referred to here is the effect of phosphate on alkalinity, parameterized as DIN/16. How much N is present as e.g. NO3 vs NH4 is not relevant.

It might be useful to include a table of the net alkalinity change associated with biological transformations of N (phytoplankton uptake, remineralization, nitrification, denitrification, N2 fixation), to help ensure that this is done consistently across models. These numbers can be found in Wolf-Gladrow et al 2007 Mar Chem 106: 287. I think such a table would be more useful than Figure 3, which I do not think is necessary.

I also note that Fe speciation is far more complex than implied in Footnote 2 to Table 5, so that it might be better to simply state that modelled dFe includes all dissolved species. Also Fe* (column 1) is not defined, assuming that the * is not just an erroneously placed footnote marker.

Minor comments

Is "online" rather than "in line" the proper terminology? In any case the authors should define it at first occurrence.

2/23 change "model-predicted" to "modelled"

7/32 "Carbon-13 is typically included in ocean models as a biotic variable influenced by fractionation effects during photosynthesis that depend on growth rate and phytoplankton type." could use some literature references.

8/25-30 Might want to mention here that while the equilibration times for Fe and DOC are much longer than for e.g. phytoplankton biomass, they are much shorter than for DIC or alkalinity.

9/18 not clear what the stray < means

12/20 "polynomical"???

12/27-28 change "pH2O is the water vapor pressure at saturation" to "pH2O is the saturation vapor pressure at sea surface temperature and salinity" (see 15/26)

16/18-21 The authors recommend that carbon chemistry calculations follow the Best Practices Guide (Dickson et al 2007). They might also consider mentioning that the BPG also gives formulae for the coefficients in equation (26). Interestingly, the definition of R used here differs slightly (1e-6 level) from that in the BPG.

17/5 delete the '*' in equation 27

Table 2 is the second footnote really necessary?

Table 5 "Mole concentration of phytoplankton expressed as chlorophyll" I think just "Concentration of chlorophyll" is more accurate (see Table 9). I also don't think the second sentence of the footnote is necessary.

---

## Short Comment (SC2) · 12 Sep 2016

This manuscript documents the experimental protocol for the biogeochemical and inert chemical tracers under the CMIP6 Ocean Model Intercomparison Project (OMIP), here referred as OMIP-BGC. The description of simulations protocols, preferred parameterizations, and diagnostics is very thorough and it provides a good guidance for all the groups involved in this intercomparison exercise. I have few comments on aspects related to the protocol definition.

1- At the beginning of section 2, authors strongly advice to use the constants recommended in best practices described by Dickson (2010). However, Orr and Epitalon (2015) clearly pointed out that some exceptions to the best practices might become

relevant when dealing with numerical models. I think that the protocol could be revised by pointing out the use of more suitable parameterizations, like e.g. K1-K2 from Millero (2010) and KF formulation in Dickson and Riley (1979).

2- The protocol for simulations indicates that initial conditions for DIC an TA are based on the recent GLODAPv2 (see section 2.2). In particular, these data are provided over two distinguished time periods, namely from 1986-1999 for the WOCE era and from 2000-2013 for the CLIVAR one (see Kay et al., 2015). It would be very useful to report in the manuscript how these data will be handled to create the initial conditions (use only one period, data blending, etc.), especially if one consider that DIC is remarkably time-dependent over long time windows.

3- In section 2.5.3, it is indicated that in-situ temperature and salinity in permil units have to be used in the computations related to the carbonate system. I think that the preferred type of these two variables could be addressed more precisely, e.g. in-situ temperature as ITS-90 and Practical Salinity as PSS-78, also to comply with the routines used in mocsy.

4- In the companion paper on OMIP physical experiments (Griffies et al., 2016) it is considered also the use of most recent Equation of State for ocean physics (TEOS-10), which relies on Conservative Temperature and Absolute Salinity. This might represent a critical issue since equilibrium constants were all derived using practical salinity (Millero, 2007; Dickson, 2010). I guess that some guidelines on the use of the most appropriate conversions tools between different formulations of temperature and salinity should be addressed in the protocol description.

5- I think it would be very useful to have a table that summarizes the requested variables for each Tier and link them to the specific experiments of both OMIP-BGC and DECK.

Key, R.M., A. Olsen, S. van Heuven, S. K. Lauvset, A. Velo, X. Lin, C. Schirnick, A. Kozyr, T. Tanhua, M. Hoppema, S. Jutterström, R. Steinfeldt, E. Jeansson, M. Ishi, F.

F. Perez, and T. Suzuki. 2015. Global Ocean Data Analysis Project, Version 2 (GLO-DAPv2), ORNL/CDIAC-162, ND-P093. Carbon Dioxide Information Analysis Center, Oak Ridge National Laboratory, US Department of Energy, Oak Ridge, Tennessee. doi: 10.3334/CDIAC/OTG.NDP093_GLODAPv2.

———————————————————

---

## Short Comment (SC3) · 12 Sep 2016

Dear OMIP BGC authors,

The CMIP Panel is undertaking a review of the CMIP6 GMD special issue papers to ensure a level of consistency in answering the key questions that were outlined in our request to submit a paper to all co-chairs of CMIP6-Endorsed MIPs. These questions are outline in the overview paper (Eyring et al, GMD, 2016) and the relevant section is summarised below:

'Each of the 21 CMIP6-Endorsed MIPs is described in a separate invited contribution to this Special Issue. These contributions will detail the goal of the MIP and the major scientific gaps the MIP is addressing, and will specify what is new compared to CMIP5 and previous CMIP phases. The contributions will include a description of the experimental design and scientific justification of each of the experiments for Tier 1 (and possibly beyond), and will link the experiments and analysis to the DECK and CMIP6 historical simulations. They will additionally include an analysis plan to fully justify the resources used to produce the various requested variables, and if the analysis plan is to compare model results to observations, the contribution will highlight possible model diagnostics and performance metrics specifying whether the comparison entails any particular requirement for the simulations or outputs (e.g. the use of observational simulators). In addition, possible observations and reanalysis products for model evaluation are discussed and the MIPs are encouraged to help facilitate their use by contributing them to the obs4MIPs/ana4MIPs archives at the ESGF (see Section 3.3). In some MIPs additional forcings beyond those used in the DECK and CMIP6 historical simulations are required, and these are described in the respective contribution as well.'

We very much welcome the OMIP BGC contribution and the hugely valuable detailing of the desired formulations for gas exchange and carbonate chemistry, diagnostic tracers and their initialisation that you currently cover in section 2. This is nicely consistent with the leadership that the other OMIP paper (Griffies et al) is also providing on the physical ocean diagnostics and together these will provide an important protocol for CMIP6.

However we would like to suggest that for consistency with the other papers in the GMD special issue, you consider moving much of section 2 to an appendix rather than in the main body of the paper. A similar suggestion was made on the Griffies at al paper for documentation of diagnostics. Some parts of section 2 such as detailing the tier 1 experiments, their length and initialisation should remain in the main paper and perhaps this section can be re-organised around these including the justification of these runs. I think much of this is already in the paper but could be better structured.

Additionally, we would like to see some more detail on some of the issues raised above,

notably;

a. More discussion on the science goals of the OMIP BGC in CMIP6 and what science gaps it is attempting to fill to be outlined in the introduction. You mention that OMIP BGC is focussed on the CMIP6 question on 'understanding systematic biases' but give no detail on what OMIP BGC is hoping to achieve that is new.

b. All MIPs have been asked to demonstrate connectivity to the DECK experiments and the CMIP6 historical simulations as one of the 10 endorsement criteria (see Table 1 in Eyring et al., 2016). Please document this for OMIP BGC.

c. You have not provided an analysis plan for the science community engaged in OMIP BGC. How are you going to use the experiments and diagnostics? Are you committing to analyse all the data that you are requesting (or can you point to other MIPs that will do so)?

d. You describe observations of e.g. CFC-11, CFC-12, SF6 etc in the introduction that might be used for evaluation of the models. Are/Could any new observations be made easily available to the modelling community (e.g. through Obs4MIPs?)

We hope you agree that some level of consistency across the MIP papers in this special issue is valuable and that the above suggestions can be accommodated in your paper.

Other comments:

- For the diagnostic section (3 and tables 4-14), what is the link to the CMIP6 data request? Perhaps you need to clarify where is the definitive documentation of what is actually being output from the models (e.g. via a link to the actual data request) and to reference the GMD paper by Martin Jukes?

With many thanks for your ongoing efforts in the CMIP6 process.

The CMIP Panel

---

## Referee Comment (RC2) · Anonymous Referee #2 · 31 Oct 2016

This manuscript presents a plan for simulations and diagnostics of biogeochemical tracers during the CMIP6 simulations, including carbon cycle and biological tracers. Model intercomparison projects are somewhat unwieldy beasts, when comparing models it is often difficult to know what to say beyond "the models differ". Reasons for this include subtle differences in model construction and parameter values as well as more fundamental issues about which processes are represented. Ideally, model intercomparison projects will try to keep as many things as similar as possible, so as to narrow down the range of possible differences between simulations. The strategy taken in this version of CMIP seems to be to make sure the different models all use the same gas exchange, atmospheric concentration, gas chemistry and carbon chemistry parameters, while using different ecosystem models. This seems a very sensible approach to me.

It would be good at the end of the introduction for the authors to define what the principal scientific goals are. Right now it appears that a principal goal is to quantify the change in ocean carbon inventory under global warming and to attribute this uptake to passive uptake by a changing circulation vs. changes in the natural storage of carbon by biology. It would be good to say a little more about why this is challenging, in particular that the long equilibration time for carbon dioxide means that the (poorly known) disequilibrium component of anthropogenic $CO_2$ is the same size as the actual signal we are trying to detect. As a result, different estimates of anthropogenic $CO_2$ differ by large amounts. Combining C14 with SF6 and CFC12 tracers offers us a way of not only testing the models, but of narrowing the observational uncertainty on anthropogenic $CO_2$. However, doing this right requires not only getting inventories but fluxes right, which in turn requires standardizing carbon chemistry (the recent paper by Lovenduski et al in Global Biogeochemical Cycles would be a good one to reference here as it shows that systematic model bias dominates regional carbon fluxes). This would motivate the discussion later in the paper.

Additionally however, biogeochemical tracers can serve as useful constraints on ocean circulation, in particular the ventilation of the deep ocean, as they average over long periods of time and exhibit strong contrasts between different regions of deep water formation. A great example of this is Broecker et al. (JGR-Oceans) use of C14 and PO4* to derive ventilation rates for Antarctic Bottom Water and North Atlantic Deep Water- the former of which is likely still better than anything that physical oceanographers have been able to quantify directly. Radiocarbon is also useful for getting at upwelling pathways, as different overturning schemas can produce vastly different distributions of surface radiocarbon with very similar overall hydrography (see Gnanadesikan et al., GBC, 2004 for an example of this).

Finally, though, there's the issue of biological variability under climate change. This will

be an important area going forward and I do not feel that the diagnostics for it have been properly prioritized. One of the main "consumers" of this work are going to be marine ecologists looking for changes in community structure and ecology. Because of this I would strongly recommend prioritizing monthly 100m-integrated biomass measurements for as many classes as exist in the model as a top priority. I disagree with the reviewer who worried about having too many diagnostics- frankly the field as a whole suffers from having too few diagnostics saved out to actually understand differences. In the current version of CMIP5, for example, many of the models save out minimum oxygen rather than the three-dimensional fields. In analysing this my group is finding that this limits the signals of changes in climate to convective zones, rather than allowing them to be tracked more broadly.

I have only one other quibble about the standards. The OCMIP2 standards for carbon and radiocarbon were generally for models that did not have interannual variability but were being forced towards a mean climate. When climate variability is included (and as noted by de Lavergne et al., 2015 this variability can have large amplitude) one can easily see variability accounting for differences of 5 GtC over the course of a century. I strongly recommend that the authors either raise the threshold for carbon trends or soften the requirement in some way.

---

## Author Comment (AC1) · 28 Nov 2016

**Response to Short Comment by T. Lovato**

This Short Comment is repeated below in gray; our response follows in black. We thank Tomas Lovato for these helpful comments.

This manuscript documents the experimental protocol for the biogeochemical and inert chemical tracers under the CMIP6 Ocean Model Intercomparison Project (OMIP), here referred as OMIP-BGC. The description of simulations protocols, preferred

parameterizations, and diagnostics is very thorough and it provides a good guidance for all the groups involved in this intercomparison exercise. I have few comments on aspects related to the protocol definition.

1- At the beginning of section 2, authors strongly advise to use the constants recommended in best practices described by Dickson (2010). However, Orr and Epitalon (2015) clearly pointed out that some exceptions to the best practices might become relevant when dealing with numerical models. I think that the protocol could be revised by pointing out the use of more suitable parameterizations, like e.g. K1-K2 from Millero (2010) and KF formulation in Dickson and Riley (1979).

These points will be addressed more thoroughly in the revised manuscript. Orr and Epitalon (2015) do discuss the Millero (2010) formulations for $K_1$ and $K_2$ relative to those recommended for best practices. However, the companion paper (Orr et al., 2015) identified internal discrepancies with the Millero (2010) formulations, recommending to remain with the best-practice formulations until those discrepancies are resolved. We will provide an updated analysis of this situation for $K_1$ and $K_2$ in the revised manuscript. Regarding $K_F$, the choice between the two available formulations (Dickson and Riley, 1979; Perez and Fraga, 1987) does not make a significant difference in computed variables (Dickson et al., 2007; Orr et al., 2015). Indeed some of the public software packages that make these carbonate chemistry calculations (e.g., various versions of CO2SYS) do not even offer a choice.

2- The protocol for simulations indicates that initial conditions for DIC an TA are based on the recent GLODAPv2 (see section 2.2). In particular, these data are provided over two distinguished time periods, namely from 1986–1999 for the WOCE era and from 2000-2013 for the CLIVAR one (see Kay et al., 2015). It would be very useful to report in the manuscript how these data will be handled to create the initial conditions (use

only one period, data blending, etc.), especially if one consider that DIC is remarkably time-dependent over long time windows.

In theory, either the first or second period could be used for initial conditions if pre-treatment of the GLODAPv2 data would include removing the anthropogenic DIC component (or part of it) based on data-based estimates (e.g., Khatiwala et al., 2009). The revised manuscript will discuss this point and stipulate the preferred option.

3- In section 2.5.3, it is indicated that in-situ temperature and salinity in permil units have to be used in the computations related to the carbonate system. I think that the preferred type of these two variables could be addressed more precisely, e.g. in situ temperature as ITS-90 and Practical Salinity as PSS-78, also to comply with the routines used in mocsy.

These specifications of the $T$ and $S$ scales will be added to the revised manuscript.

4- In the companion paper on OMIP physical experiments (Griffies et al., 2016) it is considered also the use of most recent Equation of State for ocean physics (TEOS-10), which relies on Conservative Temperature and Absolute Salinity. This might represent a critical issue since equilibrium constants were all derived using practical salinity (Millero, 2007; Dickson, 2010). I guess that some guidelines on the use of the most appropriate conversions tools between different formulations of temperature and salinity should be addressed in the protocol description.

For these conversions, the revised manuscript will recommend that model groups should use the routines from the TEOS-10 Fortran library, available from http://www.teos-10.org/software.htm.

5- I think it would be very useful to have a table that summarizes the requested variables for each Tier and link them to the specific experiments of both OMIP-BGC and DECK.

We will consider this as an option for the revised manuscript while weighing the concern of excessive duplication. The publicly available CMIP6 data-request tables for OMIP will contain the same information and can be sorted by individual modeling groups according to their needs.

Key, R.M., A. Olsen, S. van Heuven, S. K. Lauvset, A. Velo, X. Lin, C. Schirnick, A. Kozyr, T. Tanhua, M. Hoppema, S. Jutterström, R. Steinfeldt, E. Jeansson, M. Ishi, F. F. Perez, and T. Suzuki. 2015. Global Ocean Data Analysis Project, Version 2 (GLODAPv2), ORNL/CDIAC-162, ND-P093. Carbon Dioxide Information Analysis Center, Oak Ridge National Laboratory, US Department of Energy, Oak Ridge, Tennessee. doi:10.3334/CDIAC/OTG.NDP093_GLODAPv2.

**References**

Dickson, A. G. and Riley, J. P.: The estimation of acid dissociation constants in seawater media from potentiometric titrations with strong base. I. The ionic product of water - $K_W$, Mar. Chem., 7, 89–99, 1979.

Dickson, A. G., Sabine, C. L., and Christian, J. R.: Guide to best practices for ocean $CO_2$

measurements, , PICES Special Publication 3, 191 pp., http://aquaticcommons.org/1443/, 2007.

Khatiwala, S., Primeau, F., and Hall, T.: Reconstruction of the history of anthropogenic CO2 concentrations in the ocean, Nature, 462, 346–349, 2009.

Millero, F. J.: Carbonate constants for estuarine waters, Mar. Freshwater Res., 61, 139–142, doi:10.1071/MF09254, 2010.

Orr, J. C. and Epitalon, J.-M.: Improved routines to model the ocean carbonate system: mocsy 2.0, Geosci. Model Dev., 8, 485–499, doi:10.5194/gmd–8–485–2015, 2015.

Orr, J. C., Gattuso, J.-P., and Epitalon, J.-M.: Comparison of ten packages that compute ocean carbonate chemistry, Biogeosciences, 12, 1483–1510, doi:10.5194/bg–12–1483–2015, 2015.

Perez, F. F. and Fraga, F.: Association constant of fluoride and hydrogen ions in seawater, Mar. Chem., 21, 161–168, 1987.

---

## Author Comment (AC4) · 28 Nov 2016

**Response to Comment from Anonymous Referee #1**

Referee #1's detailed comments are much appreciated. They are repeated below in gray, while our responses are given in black.

This paper lays out a strategy for the biogeochemistry component (OMIP-BGC) of the Ocean Model Intercomparison Project under the umbrella of the 6th Coupled Model Intercomparison Project (CMIP6). The paper is well written and mostly

[Figure]

does a good job of outlining the experimental design for potential participants in clear and unambiguous terms. The promised OMIP-BGC web page (bottom p. 19) does not appear to be operational yet (and rather more than 4 weeks have passed).

Thanks. It is now planned that the OMIP-BGC web page be made available at about the time that the revised manuscript is submitted.

Major comments:

Overall strategy and ordering of priorities

The weakest point of the strategy is that it is vague about the priority of experiments. In CMIP5, there were tiers of experiments (Core/Tier1/Tier2), and tiers of output fields (Priority 1/2/3). This document does not really separate the two, implicitly treating all experiments as Core. The list of Priority 1 output fields is expansive and probably unrealistic.

We agree that a weak point of the manuscript is that the priority of experiments is not as clear as it should be. In the revised manuscript, we will strive to improve this deficiency, following these comments and Short Comments from others.

Some tracers are referred to as "level=1" and "level=2" (5/25), but the term is not defined. It appears to refer to output fields in which case it is synonymous with "Priority" (Tables 4-14) and there is no real conceptual problem. But the most important thing the authors need to do is to separate the x (experiments) and y (output fields) axes in a fashion similar to the CMIP5 data request, and pare down the list of Priority 1 fields

to a more realistic level.

Separating out the experiments and output fields in a clearer fashion is a good idea that we will try to implement in the revised manuscript. In the light of these comments, the coauthors will rediscuss the list for Priority 1 fields.

The list of 3D monthly fields is long. In CMIP5 no 3D ocean biogeochemistry fields were monthly. According to Moore's Law, computing power should have increased about 16-fold since CMIP5, but in practice the gain is probably much less. Making previously annual fields monthly requires a twelvefold increase just to break even in terms of the time and storage capacity it will take to access and download data, even assuming no increase in model resolution.

The lack of 3-D monthly fields below the surface in CMIP5 was an impediment to analysis. We do not wish to repeat that mistake during CMIP6. Monthly fields will be valuable to the community, and cutting back on requested fields will limit the science that can be done. See also the comments from Referee #2 along the same lines. Those that do not wish to analyze the monthly fields may choose to download and analyze only the annual mean fields.

For any model that includes feedbacks between saturation state and biology, the duplication of tracers is likely to make the 'natural CO2' experiments prohibitively expensive. Yet these are said in the Conclusion to be critical (19/14) and to be "required" on 6/19. Even if one only considers the effect of saturation state on dissolution of CaCO3, that is a minimum of three additional tracers (or maybe two if CaCO3 is parameterized). I agree with the authors that these experiments are important, but the rather superficial consideration given to the actual cost (bottom p. 6) simply underscores that the

strategy does not include a clear hierarchy of priorities for the different experiments proposed.

In defense of the original manuscript, adding three passive tracers in an online coupled physical-biogeochemical simulation does not typically increase costs greatly, far less than a factor of two. In such models, the computational cost of running the dynamical model typically dominates. If a separate simulation needs to be run for the natural component, then of course that would double the computational time. But the only required simulation (ocmip1) is initialized from data and run for 310 years. These forced ocean simulations being proposed for OMIP are much less computationally expensive than are the Earth System Model simulations at the same resolution. Nonetheless, we agree that a clearer strategy needs to be elaborated, an effort we will take on with the revised manuscript.

Alkalinity and speciation

I agree that using a truncated expression for alkalinity causes large errors, but the paper could spell out in detail exactly what they envision alkalinity as including (since the full formal definition includes lots of species that are not provided in a model simulation), rather than simply referring the reader to the provided codes.

The revised manuscript will include the equation for total alkalinity that is provided already in the common code for computing the carbonate chemistry (mocsy). This equation is also provided in the publication which describes that code (Orr and Epitalon, 2015, Equation 7).

Similarly, the authors could clarify exactly what they mean by N speciation (18/14). I

agree that the alkalinity sources and sinks associated with biological transformations (e.g., nitrification) of N species should be accounted for. But I think it is better if the word speciation is not used here.

Good point. The revised manuscript will clarify this issue, avoiding the word speciation.

For models that have N but not P it is recommended that the PO4 contribution to alkalinity be calculated as the P/N Redfield ratio times the total inorganic N concentration (16/29), which is appropriate. But then on 18/15-16 this is referred to as the effect of nitrate on alkalinity. But really what is being referred to here is the effect of phosphate on alkalinity, parameterized as DIN/16. How much N is present as e.g. NO3 vs NH4 is not relevant.

Thanks for pointing out this confusing passage. It will be clarified in the revised manuscript.

It might be useful to include a table of the net alkalinity change associated with biological transformations of N (phytoplankton uptake, remineralization, nitrification, denitrification, N2 fixation), to help ensure that this is done consistently across models. These numbers can be found in Wolf-Gladrow et al 2007 Mar Chem 106: 287. I think such a table would be more useful than Figure 3, which I do not think is necessary.

Thank you for this idea to include a table of net alkalinity changes due to biological transformations of nitrogen. We will consider this for the revised manuscript. As for Fig. 3, that presents the speciation of phosphoric acid and silicic acid species as a function of pH. It is not directly related to transformations of nitrogen. We think it adds value to the manuscript because it clarifies what species are important. There is a

common misconception among many ocean scientists who refer to phosphate ($PO_4^{3-}$) when what they really should refer to is the total dissolved inorganic phosphorus ($P_T$). Even the publications and web pages that describe the World Ocean Atlas incorrectly refer to phosphate when in fact the discrete and objectively mapped data is actually $P_T$. Eliminating this confusion seems to us to be valuable for the ocean modeling community as well as others.

I also note that Fe speciation is far more complex than implied in Footnote 2 to Table 5, so that it might be better to simply state that modelled dFe includes all dissolved species. Also Fe* (column 1) is not defined, assuming that the * is not just an erroneously placed footnote marker.

The revised manuscript will attempt to clarify these concerns about dissolved iron.

Minor comments

Is "online" rather than "in line" the proper terminology? In any case the authors should define it at first occurrence.

We may be confused about the meaning this comment. As we do not use the term "in line", is the Referee suggesting that we replace all occurrences of "online" with "in line"? This is certainly not something we are willing to do. Online has a particular meaning in the ocean modeling world. Perhaps though, the Referee is suggesting that we should define "online" when it is first used. Such will be done in the revised manuscript.

2/23 change "model-predicted" to "modelled"

This change will be made in the revised manuscript.

7/32 "Carbon-13 is typically included in ocean models as a biotic variable influenced by fractionation effects during photosynthesis that depend on growth rate and phytoplankton type." could use some literature references.

In the revised manuscript, one or more references will be added.

8/25-30 Might want to mention here that while the equilibration times for Fe and DOC are much longer than for e.g. phytoplankton biomass, they are much shorter than for DIC or alkalinity.

We will consider making such a statement in the revised manuscript.

9/18 not clear what the stray < means

This is a typo. It will be removed in the revised manuscript.

12/20 "polynomical"???

Another typo. It will be corrected to "polynomial".

12/27-28 change "pH2O is the water vapor pressure at saturation" to "pH2O is the saturation vapor pressure at sea surface temperature and salinity" (see 15/26)

We will clarify this phrase in the revised manuscript.

16/18-21 The authors recommend that carbon chemistry calculations follow the Best Practices Guide (Dickson et al 2007). They might also consider mentioning that the BPG also gives formulae for the coefficients in equation (26). Interestingly, the definition of R used here differs slightly (1e-6 level) from that in the BPG.

The original reference is Weiss (1974). We may add a reference to the BPG in the revised manuscript. The value of R has been updated since the best-practices guide was published 10 years ago. A reference for that will be provided in the revised manuscript.

17/5 delete the '*' in equation 27

This sign is indeed unnecessary and will be removed in the revised manuscript.

Table 5 "Mole concentration of phytoplankton expressed as chlorophyll" I think just "Concentration of chlorophyll" is more accurate (see Table 9). I also don't think the second sentence of the footnote is necessary.

In the revised manuscript, Table 5 will list the name of this variable exactly as it is given in the CMIP6 data request.

**References**

Orr, J. C. and Epitalon, J.-M.: Improved routines to model the ocean carbonate system: mocsy 2.0, Geosci. Model Dev., 8, 485–499, doi:10.5194/gmd–8–485–2015, 2015.

Weiss, R. F.: Carbon dioxide in water and seawater: the solubility of a non-ideal gas, Mar. Chem., 2, 203–215, 1974.

―――――――――――――――――――――

---

## Author Comment (AC5) · 29 Nov 2016

**Response to Comment from Anonymous Referee #2**

The suggestions from Referee #2 will prove valuable to improve the OMIP-BGC manuscript. They are repeated below in gray, while our responses are given in black.

This manuscript presents a plan for simulations and diagnostics of biogeochemical tracers during the CMIP6 simulations, including carbon cycle and biological tracers. Model intercomparison projects are somewhat unwieldy beasts, when comparing

models it is often difficult to know what to say beyond "the models differ". Reasons for this include subtle differences in model construction and parameter values as well as more fundamental issues about which processes are represented. Ideally, model intercomparison projects will try to keep as many things as similar as possible, so as to narrow down the range of possible differences between simulations. The strategy taken in this version of CMIP seems to be to make sure the different models all use the same gas exchange, atmospheric concentration, gas chemistry and carbon chemistry parameters, while using different ecosystem models. This seems a very sensible approach to me.

We agree. Physical forcing of the ocean only model simulations will also be identical.

It would be good at the end of the introduction for the authors to define what the principal scientific goals are. Right now it appears that a principal goal is to quantify the change in ocean carbon inventory under global warming and to attribute this uptake to passive uptake by a changing circulation vs. changes in the natural storage of carbon by biology. It would be good to say a little more about why this is challenging, in particular that the long equilibration time for carbon dioxide means that the (poorly known) disequilibrium component of anthropogenic $CO_2$ is the same size as the actual signal we are trying to detect. As a result, different estimates of anthropogenic $CO_2$ differ by large amounts. Combining C14 with SF6 and CFC12 tracers offers us a way of not only testing the models, but of narrowing the observational uncertainty on anthropogenic $CO_2$. However, doing this right requires not only getting inventories but fluxes right, which in turn requires standardizing carbon chemistry (the recent paper by Lovenduski et al in Global Biogeochemical Cycles would be a good one to reference here as it shows that systematic model bias dominates regional carbon fluxes). This would motivate the discussion later in the paper.

In the revised manuscript, the main scientific goals of the project will be better detailed as also requested in the short comment from the CMIP Panel. Referee #2 provides valuable suggestions, including the interesting reference to the work by Lovenduski et al. (2016), which we plan to address in the revised manuscript.

Additionally however, biogeochemical tracers can serve as useful constraints on ocean circulation, in particular the ventilation of the deep ocean, as they average over long periods of time and exhibit strong contrasts between different regions of deep water formation. A great example of this is Broecker et al. (JGR-Oceans) use of C14 and PO4* to derive ventilation rates for Antarctic Bottom Water and North Atlantic Deep Water—the former of which is likely still better than anything that physical oceanographers have been able to quantify directly. Radiocarbon is also useful for getting at upwelling pathways, as different overturning schemas can produce vastly different distributions of surface radiocarbon with very similar overall hydrography (see Gnanadesikan et al., GBC, 2004 for an example of this).

The utility of the tracers that will be modeled in the OMIP-BGC simulations to help constrain ocean circulation, particularly in the deep ocean, will be brought forward in the Introduction to help emphasize these objectives. The publications suggested by Referee #2 (Broecker et al., 1998; Gnanadesikan et al., 2004) are excellent examples that we will consider mentioning in the revised Introduction.

Finally, though, there's the issue of biological variability under climate change. This will be an important area going forward and I do not feel that the diagnostics for it have been properly prioritized. One of the main "consumers" of this work are going to be marine ecologists looking for changes in community structure and ecology. Because of this I would strongly recommend prioritizing monthly 100m-integrated biomass measurements for as many classes as exist in the model as a top priority. I disagree

with the reviewer who worried about having too many diagnostics- frankly the field as a whole suffers from having too few diagnostics saved out to actually understand differences. In the current version of CMIP5, for example, many of the models save out minimum oxygen rather than the three-dimensional fields. In analysing this my group is finding that this limits the signals of changes in climate to convective zones, rather than allowing them to be tracked more broadly.

Agreed. The requested monthly fields will be valuable contributions that OMIP and CMIP6 can provide to help offer a better understanding of biological and chemical variability and potential changes under climate change and rising levels of $CO_2$.

I have only one other quibble about the standards. The OCMIP2 standards for carbon and radiocarbon were generally for models that did not have interannual variability but were being forced towards a mean climate. When climate variability is included (and as noted by de Lavergne et al., 2015 this variability can have large amplitude) one can easily see variability accounting for differences of 5 GtC over the course of a century. I strongly recommend that the authors either raise the threshold for carbon trends or soften the requirement in some way.

We are not sure that we fully understand the meaning of this comment nor the exact publication that is being referred to. The OMIP-BGC models will account for climate variability, either when forced by reanalysis data or when coupled within an Earth System Model framework. The boundary condition for atmospheric $CO_2$ does include interannual variability, being based on annual-mean observations. The same may be said for the atmospheric $^{14}C/C$ ratio. Furthermore given the corresponding air-sea equilibration times (1 yr and 10 yr, respectively) it is not clear to us that the atmospheric records that are to be used to force the OMIP-BGC simulations are inadequate to study interannual variability. More clarification from Referee #2 on the

nuanced meaning of this comment would be most welcome.

**References**

Broecker, W., Peacock, S., Walker, S., Weiss, R., Fahrbach, E., Schröder, M., Mikolajewicz, U., Heinze, C., Key, R., Peng, T.-H., et al.: How much deep water is formed in the Southern Ocean?, Journal of Geophysical Research: Oceans, 103, 15 833–15 843, 1998.

Gnanadesikan, A., Dunne, J. P., Key, R. M., Matsumoto, K., Sarmiento, J. L., Slater, R. D., and Swathi, P.: Oceanic ventilation and biogeochemical cycling: Understanding the physical mechanisms that produce realistic distributions of tracers and productivity, Global Biogeochemical Cycles, 18, doi:10.1029/2003GB002 097, 1944–9224, 2004.

Lovenduski, N. S., McKinley, G. A., Fay, A. R., Lindsay, K., and Long, M. C.: Partitioning uncertainty in ocean carbon uptake projections: Internal variability, emission scenario, and model structure, Global Biogeochemical Cycles, 30, doi:10.1002/2016GB005 426, 1276–1287, 2016.

---

## Author Response (AR1)

**Author Response on "Biogeochemical protocols and diagnostics for the CMIP6 Ocean Model Intercomparison Project (OMIP)"**

James C. Orr[1], Raymond G. Najjar[2], Olivier Aumont[3], Laurent Bopp[1], John L. Bullister[4], Gokhan Danabasoglu[5], Scott C. Doney[6], John P. Dunne[7], Jean-Claude Dutay[1], Heather Graven[8], Stephen M. Griffies[7], Jasmin G. John[7], Fortunat Joos[9], Ingeborg Levin[10], Keith Lindsay[5], Richard J. Matear[11], Galen A. McKinley[12], Anne Mouchet[13,14], Andreas Oschlies[15], Anastasia Romanou[16], Reiner Schlitzer[17], Alessandro Tagliabue[18], Toste Tanhua[15], and Andrew Yool[19]

[1]LSCE/IPSL, Laboratoire des Sciences du Climat et de l'Environnement, CEA-CNRS-UVSQ, Gif-sur-Yvette, France
[2]Dept. of Meteorology, Pennsylvania State University, University Park, Pennsylvania, USA
[3]Laboratoire d'Océanographie et de Climatologie: Expérimentation et Approches Numériques, IPSL, Paris, France
[4]Pacific Marine Environmental Laboratory, NOAA, Seattle, Washington, USA
[5]National Center for Atmospheric Research, Boulder, Colorado, USA
[6]Marine Chemistry & Geochemistry, Woods Hole Oceanographic Institution, USA
[7]NOAA Geophysical Fluid Dynamics Laboratory, Princeton, New Jersey, USA
[8]Dept. of Physics, Imperial College, London, UK
[9]Climate and Environmental Physics, Physics Inst. & Oeschger Center for Climate Change Res., Univ. of Bern, Switzerland
[10]Institut fuer Umweltphysik, Universitaet Heidelberg, Heidelberg, Germany
[11]CSIRO Oceans and Atmosphere, Hobart, Tasmania 7000, Australia
[12]Atmospheric and Oceanic Sciences, University of Wisconsin-Madison, Wisconsin, USA
[13]Max Planck Institute for Meteorology, Hamburg, Germany
[14]Astrophysics, Geophysics and Oceanography Department, University of Liege, Liege, Belgium
[15]GEOMAR Helmholtz Centre for Ocean Research Kiel, Kiel, Germany
[16]Columbia University and NASA-Goddard Institute for Space Studies, New York, NY, USA
[17]Alfred Wegener Institute, Bremerhaven, Germany
[18]Earth, Ocean and Ecological Sciences, University of Liverpool, Liverpool, UK
[19]National Oceanographic Centre, Southampton, UK

*Correspondence to:* James Orr (james.orr@lsce.ipsl.fr)

We thank the two referees and the three contributors of short comments for their many thoughtful remarks, which together have improved the manuscript. These comments are repeated below in gray, our first responses follow in black, and our final response and changes to the manuscript are indicated in blue.

5    **Response to Comment from Anonymous Referee #1**

This paper lays out a strategy for the biogeochemistry component (OMIP-BGC) of the Ocean Model Intercomparison Project under the umbrella of the 6th Coupled Model Intercomparison Project (CMIP6). The paper is well written and mostly does a good job of outlining the experimental design for potential participants in clear and unambiguous terms. The promised OMIP-

BGC web page (bottom p. 19) does not appear to be operational yet (and rather more than 4 weeks have passed).

Thanks. There now exists an unreleased OMIP-BGC web page. It will be made available once the initialization data sets are finalized, within 15 days after this revised manuscript is submitted. We are sorry for this delay, but the OMIP-BGC web page will go online before this revised paper has a chance to be published.

Major comments:

Overall strategy and ordering of priorities

The weakest point of the strategy is that it is vague about the priority of experiments. In CMIP5, there were tiers of experiments (Core/Tier1/Tier2), and tiers of output fields (Priority 1/2/3). This document does not really separate the two, implicitly treating all experiments as Core. The list of Priority 1 output fields is expansive and probably unrealistic.

We agree that a weak point of the manuscript is that the priority of experiments is not as clear as it should be. In the revised manuscript, we will strive to improve this deficiency, following these comments and Short Comments from others.

The revised manuscript is clearer about the experiments, using Tier 1 to indicate the required simulation *(omip1)* and Tier 2 to indicate the optional simulation *(omip1-spunup)*. Furthermore, Priorities 1, 2, and 3 are indicated in the diagnostic tables.

Some tracers are referred to as "level=1" and "level=2" (5/25), but the term is not defined. It appears to refer to output fields in which case it is synonymous with "Priority" (Tables 4-14) and there is no real conceptual problem. But the most important thing the authors need to do is to separate the x (experiments) and y (output fields) axes in a fashion similar to the CMIP5 data request, and pare down the list of Priority 1 fields to a more realistic level.

Separating out the experiments and output fields in a clearer fashion is a good idea that we will try to implement in the revised manuscript. In the light of these comments, the coauthors will rediscuss the list for Priority 1 fields.

We have changed "level" to "priority" in this passage. We no longer use level to indicate the priority, anywhere in the revised manuscript. The approved CMIP6 terms to designate the importance of the experiments (Tier) and the output fields (Priority) are now used rigourously throughout the text.

The list of 3D monthly fields is long. In CMIP5 no 3D ocean biogeochemistry fields were monthly. According to Moore's Law, computing power should have increased about 16-fold since CMIP5, but in practice the gain is probably much less. Making previously annual fields monthly requires a twelvefold increase just to break even in terms of the time and storage capacity

it will take to access and download data, even assuming no increase in model resolution.

The lack of 3-D monthly fields below the surface in CMIP5 was an impediment to analysis. We do not wish to repeat that mistake during CMIP6. Monthly fields will be valuable to the community, and cutting back on requested fields will limit the science that can be done. See also the comments from Referee #2 along the same lines. Those that do not wish to analyze the monthly fields may choose to download and analyze only the annual mean fields.

After discussion among coauthors, we now list only four required 3-D fields to be stored at monthly frequency (Priority 1). All other 3-D fields have been demoted to Priority 2 (optional). To partially compensate, we ask for surface monthly concentrations (Priority 1) for the demoted 3-D tracer fields.

For any model that includes feedbacks between saturation state and biology, the duplication of tracers is likely to make the 'natural CO2' experiments prohibitively expensive. Yet these are said in the Conclusion to be critical (19/14) and to be "required" on 6/19. Even if one only considers the effect of saturation state on dissolution of CaCO3, that is a minimum of three additional tracers (or maybe two if CaCO3 is parameterized). I agree with the authors that these experiments are important, but the rather superficial consideration given to the actual cost (bottom p. 6) simply underscores that the strategy does not include a clear hierarchy of priorities for the different experiments proposed.

In defense of the original manuscript, adding three passive tracers in an online coupled physical-biogeochemical simulation does not typically increase costs greatly, far less than a factor of two. In such models, the computational cost of running the dynamical model typically dominates. If a separate simulation needs to be run for the natural component, then of course that would double the computational time. But the only required simulation (ocmip1) is initialized from data and run for 310 years. These forced ocean simulations being proposed for OMIP are much less computationally expensive than are the Earth System Model simulations at the same resolution. Nonetheless, we agree that a clearer strategy needs to be elaborated, an effort we will take on with the revised manuscript.

We have adapted the wording of the two sentences to be less emphatic; however, we still consider the natural carbon tracer to be crucial to eliminate model drift.

Alkalinity and speciation

I agree that using a truncated expression for alkalinity causes large errors, but the paper could spell out in detail exactly what they envision alkalinity as including (since the full formal definition includes lots of species that are not provided in a model simulation), rather than simply referring the reader to the provided codes.

The revised manuscript will include the equation for total alkalinity that is provided already in the common code for computing the carbonate chemistry (mocsy). This equation is also provided in the publication which describes that code (Orr and Epitalon, 2015, Equation 7).

5    The revised manuscript now includes an equation that lists all components of total alkalinity that are recommended to be included in the OMIP simulations (equations 32–38

Similarly, the authors could clarify exactly what they mean by N speciation (18/14). I agree that the alkalinity sources and sinks associated with biological transformations (e.g., nitrification) of N species should be accounted for. But I think it is better
10   if the word speciation is not used here.

Good point. The revised manuscript will clarify this issue, avoiding the word speciation.

We avoid the term "nitrogen speciation" in the revised manuscript. Instead we use the term "different inorganic forms of
15   nitrogen"
For models that have N but not P it is recommended that the PO4 contribution to alkalinity be calculated as the P/N Redfield ratio times the total inorganic N concentration (16/29), which is appropriate. But then on 18/15-16 this is referred to as the effect of nitrate on alkalinity. But really what is being referred to here is the effect of phosphate on alkalinity, parameterized as DIN/16. How much N is present as e.g. NO3 vs NH4 is not relevant.

Thanks for pointing out this confusing passage. It will be clarified in the revised manuscript.

Our intent concerning the sentence on 18/15-16 was not to account for the effect of phosphate. Rather it was to account for the effects of the different inorganic forms of nitrogen on alkalinity. Indeed, their effects differ as a function of their form as
25   detailed by Wolf-Gladrow et al. (2007). That sentence has now been replaced with "Models with $P_T$ as the sole macronutrient tracer should consider accounting for the effect of nitrate assimilation and remineralization on alkalinity, effects that are 16 times larger than for those for $P_T$ (Wolf-Gladrow et al., 2007)."

It might be useful to include a table of the net alkalinity change associated with biological transformations of N (phyto-
30   plankton uptake, remineralization, nitrification, denitrification, N2 fixation), to help ensure that this is done consistently across models. These numbers can be found in Wolf-Gladrow et al 2007 Mar Chem 106: 287. I think such a table would be more useful than Figure 3, which I do not think is necessary.

Thank you for this idea to include a table of net alkalinity changes due to biological transformations of nitrogen. We will
35   consider this for the revised manuscript. As for Fig. 3, that presents the speciation of phosphoric acid and silicic acid species

as a function of pH. It is not directly related to transformations of nitrogen. We think it adds value to the manuscript because it clarifies what species are important. There is a common misconception among many ocean scientists who refer to phosphate ($PO_4^{3-}$) when what they really should refer to is the total dissolved inorganic phosphorus ($P_T$). Even the publications and web pages that describe the World Ocean Atlas incorrectly refer to phosphate when in fact the discrete and objectively mapped data

5  is actually $P_T$. Eliminating this confusion seems to us to be valuable for the ocean modeling community as well as others.

The Table by Wolf-Gladrow et al. (2007) is interesting and we now cite that publication in the revised manuscript when mentioning the need to account for effects from nitrogen on alkalinity. We stop short though of republishing the same table in our revised manuscript. Yet we do not wish to remove Figure 3 from the revised manuscript because our bjerrum plot for

10  the phosphoric acid system differs from that provided by Wolf-Gladrow et al. (2007). Indeed our curve for $HPO_4^-$ shows a peak around the pH of surface seawater, while the same curve in Figure 3 of Wolf-Gladrow et al. (2007) exhibits a plateau that extends from pH 8 to pH 12. That difference arises because Wolf-Gladrow et al. uses values for pK1, pK2, and pK3 of the phosphiric acid system that are appropriate for pure water; conversely, we use values that are appropriate for seawater.

15  I also note that Fe speciation is far more complex than implied in Footnote 2 to Table 5, so that it might be better to simply state that modelled dFe includes all dissolved species. Also Fe* (column 1) is not defined, assuming that the * is not just an erroneously placed footnote marker.

The revised manuscript will attempt to clarify these concerns about dissolved iron.

We have changed the footnote to "modeled dissolved iron includes all simulated dissolved species, both free and organically complexed". The "*" should have pointed to the footnote that describes the meaning of dissolved iron; it now does so in the revised manuscript.

25  Minor comments

Is "online" rather than "in line" the proper terminology? In any case the authors should define it at first occurrence.

We may be confused about the meaning this comment. As we do not use the term "in line", is the Referee suggesting that

30  we replace all occurrences of "online" with "in line"? This is not what we wish to do. Online has a particular meaning in the ocean modeling world. Perhaps though, the Referee is suggesting that we should define "online" when it is first used. Such will be done in the revised manuscript.

We now define online when it is first used (in the Introduction).

2/23 change "model-predicted" to "modelled"

In the revised manuscript, we have changed "model-predicted" to "simulated".

7/32 "Carbon-13 is typically included in ocean models as a biotic variable influenced by fractionation effects during photosynthesis that depend on growth rate and phytoplankton type." could use some literature references.

In the revised manuscript, we have added one reference (Tagliabue and Bopp, 2008).

8/25-30 Might want to mention here that while the equilibration times for Fe and DOC are much longer than for e.g. phytoplankton biomass, they are much shorter than for DIC or alkalinity.

We prefer not to make this statement in the revised manuscript. Although true, it distracts from the paragraph's topic, which is about how to initialize tracer fields, not their equilibration time.

9/18 not clear what the stray < means

This was a typo. It has been removed in the revised manuscript.

12/20 "polynomical"???

Another typo. It has been corrected to "polynomial".

12/27-28 change "pH2O is the water vapor pressure at saturation" to "pH2O is the saturation vapor pressure at sea surface temperature and salinity" (see 15/26)

In the revised manuscript, we have changed the wording to the "pH2O is the vapor pressure of water (also in atm) at sea surface temperature and salinity". The term "vapor pressure of water" is exactly the phrasing used by Weiss and Price (1980).

16/18-21 The authors recommend that carbon chemistry calculations follow the Best Practices Guide (Dickson et al., 2007). They might also consider mentioning that the BPG also gives formulae for the coefficients in equation (26). Interestingly, the definition of R used here differs slightly (1e-6 level) from that in the BPG.

The original reference is Weiss (1974). We may add a reference to the BPG in the revised manuscript. The value of R has been updated since the best-practices guide was published 10 years ago.

We cite the original reference (Weiss, 1974); the BPG refers to the same coefficients (from Weiss) for B and their version of equation (26) is identical. The value of R that we refer to here is from CODATA(2006) of the NIST. The slight difference does not lead to significant differences in model calculations.

17/5 delete the '*' in equation 27

This sign was indeed unnecessary. It has been removed in the revised manuscript.

Table 5 "Mole concentration of phytoplankton expressed as chlorophyll" I think just "Concentration of chlorophyll" is more accurate (see Table 9). I also don't think the second sentence of the footnote is necessary.

In the revised manuscript, Table 5 will list the name of this variable exactly as it is given in the CMIP6 data request.

That entry in Table 5 has now been changed to "Mass Concentration of Chlorophyll in Seawater". More generally, much work has gone into revising the tables to make descriptions clearer and consistent with the CMIP6 MIP Table for OMIP.

**Response to Comment from Anonymous Referee #2**

This manuscript presents a plan for simulations and diagnostics of biogeochemical tracers during the CMIP6 simulations, including carbon cycle and biological tracers. Model intercomparison projects are somewhat unwieldy beasts, when comparing models it is often difficult to know what to say beyond "the models differ". Reasons for this include subtle differences in model construction and parameter values as well as more fundamental issues about which processes are represented. Ideally, model intercomparison projects will try to keep as many things as similar as possible, so as to narrow down the range of possible differences between simulations. The strategy taken in this version of CMIP seems to be to make sure the different models all use the same gas exchange, atmospheric concentration, gas chemistry and carbon chemistry parameters, while using different ecosystem models. This seems a very sensible approach to me.

We agree. Physical forcing of the ocean only model simulations will also be identical.

It would be good at the end of the introduction for the authors to define what the principal scientific goals are. Right now it appears that a principal goal is to quantify the change in ocean carbon inventory under global warming and to attribute this uptake to passive uptake by a changing circulation vs. changes in the natural storage of carbon by biology. It would be good to say a little more about why this is challenging, in particular that the long equilibration time for carbon dioxide means that the (poorly known) disequilibrium component of anthropogenic CO2 is the same size as the actual signal we are trying to detect.

As a result, different estimates of anthropogenic CO2 differ by large amounts. Combining C14 with SF6 and CFC12 tracers offers us a way of not only testing the models, but of narrowing the observational uncertainty on anthropogenic CO2. However, doing this right requires not only getting inventories but fluxes right, which in turn requires standardizing carbon chemistry (the recent paper by Lovenduski et al in Global Biogeochemical Cycles would be a good one to reference here as it shows that

5   systematic model bias dominates regional carbon fluxes). This would motivate the discussion later in the paper.

In the revised manuscript, the main scientific goals of the project will be better detailed as also requested in the short comment from the CMIP Panel. Referee #2 provides valuable suggestions, including the interesting reference to the work by Lovenduski et al. (2016), which we plan to address in the revised manuscript.

To better detail the scientific goals of OMIP-BGC, the final paragraph of the Introduction of the submitted manuscript has been modified, and two new paragraphs have been added just afterwards. The reference suggested by Referee #2 has been cited.

Additionally however, biogeochemical tracers can serve as useful constraints on ocean circulation, in particular the ventila-

15   tion of the deep ocean, as they average over long periods of time and exhibit strong contrasts between different regions of deep water formation. A great example of this is Broecker et al. (JGR-Oceans) use of C14 and PO4* to derive ventilation rates for Antarctic Bottom Water and North Atlantic Deep Water—the former of which is likely still better than anything that physical oceanographers have been able to quantify directly. Radiocarbon is also useful for getting at upwelling pathways, as different overturning schemas can produce vastly different distributions of surface radiocarbon with very similar overall hydrography

20   (see Gnanadesikan et al., GBC, 2004 for an example of this).

The utility of the tracers that will be modeled in the OMIP-BGC simulations to help constrain ocean circulation, particularly in the deep ocean, will be brought forward in the Introduction to help emphasize these objectives. The publications suggested by Referee #2 (Broecker et al., 1998; Gnanadesikan et al., 2004) are excellent examples that we will consider mentioning in

25   the revised Introduction.

In the submitted manuscript, the Introduction already mentioned that $^{14}$C is used to assess subsurface ventilation times. However, we follow Referre #2's advice, emphasizing this point further in ther revised manuscript by also mentioning the study by (Broecker et al., 1998).

Finally, though, there's the issue of biological variability under climate change. This will be an important area going forward and I do not feel that the diagnostics for it have been properly prioritized. One of the main "consumers" of this work are going to be marine ecologists looking for changes in community structure and ecology. Because of this I would strongly recommend prioritizing monthly 100m-integrated biomass measurements for as many classes as exist in the model as a top

35   priority. I disagree with the reviewer who worried about having too many diagnostics- frankly the field as a whole suffers from

having too few diagnostics saved out to actually understand differences. In the current version of CMIP5, for example, many of the models save out minimum oxygen rather than the three-dimensional fields. In analysing this my group is finding that this limits the signals of changes in climate to convective zones, rather than allowing them to be tracked more broadly.

5     Agreed. The requested monthly fields will be valuable contributions that OMIP and CMIP6 can provide to help offer a better understanding of biological and chemical variability and potential changes under climate change and rising levels of $CO_2$.

    In the revised manuscript, we still request monthly tracer fields also in subsurface waters (i.e., monthly 3-D fields), but in response to Referee #1 the Priority for most of those fields has been lowered from 1 (required) to 2 (optional).

    I have only one other quibble about the standards. The OCMIP2 standards for carbon and radiocarbon were generally for models that did not have interannual variability but were being forced towards a mean climate. When climate variability is included (and as noted by de Lavergne et al., 2015 this variability can have large amplitude) one can easily see variability accounting for differences of 5 GtC over the course of a century. I strongly recommend that the authors either raise the threshold

15 for carbon trends or soften the requirement in some way.

    We are not sure that we fully understand the meaning of this comment nor the exact publication that is being referred to. The OMIP-BGC models will account for climate variability, either when forced by reanalysis data or when coupled within an Earth System Model framework. The boundary condition for atmospheric $CO_2$ does include interannual variability, being based on

20 annual-mean observations. The same may be said for the atmospheric $^{14}$C/C ratio. Furthermore given the corresponding air-sea equilibration times (1 yr and 10 yr, respectively) it is not clear to us that the atmospheric records that are to be used to force the OMIP-BGC simulations are inadequate to study interannual variability. More clarification from Referee #2 on the nuanced meaning of this comment would be most welcome.

25     No changes were made to the submitted manuscript regarding this point because we already account for climate variability in the OMIP protocols, unlike for OCMIP2.

**Response to Short Comment by N. Swart**

Firstly, thanks to the authors for coordinating the OMIP-BGC effort and putting together this very comprehensive document-

30 ing paper. I have a few comments, which I think will be relevant to modelling centres trying to perform the simulations and diagnostics you describe:

- For the diagnostic output (section 3 and tables 9 to 14), no indication is given on which variables are expected only for the OMIP-BGC ocean model runs; and which variables are expected from the Earth System Models doing the "full" CMIP6 runs (historical etc), unless I missed this somewhere. It would be good to mark clearly if there are variables expected only for OMIP-BGC runs, but not the rest of CMIP6.

Conceptually there is no difference in output requirements for the forced ocean simulations made for OMIP and the coupled simulations made with the Earth System Models that are participating in CMIP6. These simulations differ in forcing but not in the types of output requested. However, we will further consider this point and clarify when distinguishing the two types of simulations in the revised manuscript.

The first two sentences in our response just above are now included in Section 3 of the revised manuscript.

- Table 5 and 9 mark all variables as "priority 1". My understanding of a priority 1 variable is from the CMIP6 data request: "all participating groups must commit to supplying all priority 1 variables". Thus, priority 1 variables should be the lowest

15 common denominator that all groups can provide. In tables 5 and 9, there are multiple variables (like 13C, and all the "abiotic" terms) which are not carried in all (or even most) BGC models. Thus, groups face the significant coding and computation expense of adding 7+ new tracers if they are "required" to provide all these terms to participate in OMIP-BGC. If these variables are not indeed "required", please mark them as priority 2, in which case it is clear they can be provided optionally, if available. Otherwise, as I understanding it, any group not providing all these variables will end up not "participating" [at least officially].

These are excellent points. Priorities will be clarified following the CMIP6 data request for OMIP that has been refined since the original manuscript was submitted. Priorities will be adjusted and explained in detail, following the CMIP6 guidelines.

Motivated by these remarks, much discussion ensued among coauthors. This has led to substantial changes to the Diagnostic

25 tables. The priorities for the monthly 3-D fields has generally been lowered from 1 to 2. Tables 5 and 9 list fewer tracers, including only those that all model groups will be able to contribute.

In the revised manuscript we will also make it clearer that $\delta^{13}C$ simulations are recommended only for those who already have experience modeling this tracer. The abiotic tracers are highly recommended but not required for participation in OMIP.

30 Fortunately, many modeling groups already have experience simulating abiotic dissolved inorganic carbon and radiocarbon, and for those that do not, their addition as new tracers is straightforward.

We have now added the following 3 sentences to the subsection on C-13: "Groups that have experience modeling $^{13}C$ in their biogeochemical model are requested to include it as a tracer in the OMIP-BGC simulations. Groups without experience should avoid adding it. Groups may participate in OMIP without including $^{13}C$ as a tracer."

**Response to Short Comment by T. Lovato**

This Short Comment is repeated below in gray; our response follows in black. We thank Tomas Lovato for these helpful comments.

5    This manuscript documents the experimental protocol for the biogeochemical and inert chemical tracers under the CMIP6 Ocean Model Intercomparison Project (OMIP), here referred as OMIP-BGC. The description of simulations protocols, preferred parameterizations, and diagnostics is very thorough and it provides a good guidance for all the groups involved in this intercomparison exercise. I have few comments on aspects related to the protocol definition.

10    1- At the beginning of section 2, authors strongly advise to use the constants recommended in best practices described by Dickson (2010). However, Orr and Epitalon (2015) clearly pointed out that some exceptions to the best practices might become relevant when dealing with numerical models. I think that the protocol could be revised by pointing out the use of more suitable parameterizations, like e.g. K1-K2 from Millero (2010) and KF formulation in Dickson and Riley (1979).

15    These points will be addressed more thoroughly in the revised manuscript. Orr and Epitalon (2015) do discuss the Millero (2010) formulations for $K_1$ and $K_2$ relative to those recommended for best practices. However, the companion paper (Orr et al., 2015) identified internal discrepancies with the Millero (2010) formulations, recommending to remain with the best-practice formulations until those discrepancies are resolved. We will provide an updated analysis of this situation for $K_1$ and $K_2$ in the revised manuscript. Regarding $K_F$, the choice between the two available formulations (Dickson and Riley, 1979; Perez and
20   Fraga, 1987) does not make a significant difference in computed variables (Dickson et al., 2007; Orr et al., 2015). Indeed some of the public software packages that make these carbonate chemistry calculations (e.g., various versions of CO2SYS) do not even offer a choice.

   Given the statements in our first response above, we stand by our choice to recommend that modelers use all the constants
25   recommended for best practices Dickson et al. (2007); Dickson (2010). We have not changed the revised manuscript in this regard.

   2- The protocol for simulations indicates that initial conditions for DIC an TA are based on the recent GLODAPv2 (see section 2.2). In particular, these data are provided over two distinguished time periods, namely from 1986–1999 for the WOCE
30   era and from 2000-2013 for the CLIVAR one (see Key et al., 2015). It would be very useful to report in the manuscript how these data will be handled to create the initial conditions (use only one period, data blending, etc.), especially if one consider that DIC is remarkably time-dependent over long time windows.

In theory, either the first or second period could be used for initial conditions if pre-treatment of the GLODAPv2 data would include removing the anthropogenic DIC component (or part of it) based on data-based estimates (e.g., Khatiwala et al., 2009). The revised manuscript will discuss this point and stipulate the preferred option.

In the revised manuscript, we now write, "For greater consistency with GLODAPv1, OMIP-BGC model groups will use the $C_T$ and $A_T$ fields from GLODAPv2's first period (1986–1999, the WOCE era)."

3- In section 2.5.3, it is indicated that in-situ temperature and salinity in permil units have to be used in the computations related to the carbonate system. I think that the preferred type of these two variables could be addressed more precisely, e.g. in situ temperature as ITS-90 and Practical Salinity as PSS-78, also to comply with the routines used in mocsy.

These specifications of the $T$ and $S$ scales have been added to the revised manuscript.

4-In the companion paper on OMIP physical experiments (Griffies et al., 2016) it is considered also the use of most recent Equation of State for ocean physics (TEOS-10), which relies on Conservative Temperature and Absolute Salinity. This might represent a critical issue since equilibrium constants were all derived using practical salinity (Millero, 2007; Dickson, 2010). I guess that some guidelines on the use of the most appropriate conversions tools between different formulations of temperature and salinity should be addressed in the protocol description.

For these conversions, the revised manuscript will recommend that model groups should use the routines from the TEOS-10 Fortran library, available from http://www.teos-10.org/software.htm.

In section 2.6 of the revised manuscript, we have added the following "Although by default *mocsy* uses older scales for temperature and salinity (ITS90 and PSS78, respectively) for input, the latest version now includes a new option so that modelers can choose to use the new TEOS-10 standards (Conservative Temperature and Absolute Salinity)."

5- I think it would be very useful to have a table that summarizes the requested variables for each Tier and link them to the specific experiments of both OMIP-BGC and DECK.

We will consider this as an option for the revised manuscript while weighing the concern of excessive duplication. The publicly available CMIP6 data-request tables for OMIP will contain the same information and can be sorted by individual modeling groups according to their needs.

As we already have 16 Tables in the revised manuscript, it would be unwise to add more. However, we do plan to add the link fot the final OMIP MIP tables (Excel spreadsheets) to the OMIP-BGC web page, so that modelers can download those

tables and organize the data request as they prefer.

Key, R.M., A. Olsen, S. van Heuven, S. K. Lauvset, A. Velo, X. Lin, C. Schirnick, A. Kozyr, T. Tanhua, M. Hoppema, S. Juttterström, R. Steinfeldt, E. Jeansson, M. Ishi, F. F. Perez, and T. Suzuki. 2015. Global Ocean Data Analysis Project, Version 2 (GLODAPv2), ORNL/CDIAC-162, ND-P093. Carbon Dioxide Information Analysis Center, Oak Ridge National Laboratory, US Department of Energy, Oak Ridge, Tennessee. doi:10.3334/CDIAC/OTG.NDP093_GLODAPv2.

**Response to Short Comment by C. Senior, representing the CMIP Panel**

The CMIP Panel is undertaking a review of the CMIP6 GMD special issue papers to ensure a level of consistency in answering the key questions that were outlined in our request to submit a paper to all co-chairs of CMIP6-Endorsed MIPs. These questions are outlined in the overview paper (Eyring et al, GMD, 2016) and the relevant section is summarised below:

Each of the 21 CMIP6-Endorsed MIPs is described in a separate invited contribution to this Special Issue. These contributions will detail the goal of the MIP and the major scientific gaps the MIP is addressing, and will specify what is new compared to CMIP5 and previous CMIP phases. The contributions will include a description of the experimental design and scientific justification of each of the experiments for Tier 1 (and possibly beyond), and will link the experiments and analysis to the DECK and CMIP6 historical simulations. They will additionally include an analysis plan to fully justify the resources used to produce the various requested variables, and if the analysis plan is to compare model results to observations, the contribution will highlight possible model diagnostics and performance metrics specifying whether the comparison entails any particular requirement for the simulations or outputs (e.g. the use of observational simulators). In addition, possible observations and reanalysis products for model evaluation are discussed and the MIPs are encouraged to help facilitate their use by contributing them to the obs4MIPs/ana4MIPs archives at the ESGF (see Section 3.3). In some MIPs additional forcings beyond those used in the DECK and CMIP6 historical simulations are required, and these are described in the respective contribution as well.

We very much welcome the OMIP BGC contribution and the hugely valuable detailing of the desired formulations for gas exchange and carbonate chemistry, diagnostic tracers and their initialisation that you currently cover in section 2. This is nicely consistent with the leadership that the other OMIP paper (Griffies et al) is also providing on the physical ocean diagnostics and together these will provide an important protocol for CMIP6.

Thank you.

However we would like to suggest that for consistency with the other papers in the GMD special issue, you consider moving much of section 2 to an appendix rather than in the main body of the paper. A similar suggestion was made on the Griffies at

al paper for documentation of diagnostics. Some parts of section 2 such as detailing the tier 1 experiments, their length and initialisation should remain in the main paper and perhaps this section can be re-organised around these including the justification of these runs. I think much of this is already in the paper but could be better structured.

5    The question of moving some of information on protocols to an appendix will be considered. However, we fear that a reorganization that would break apart the protocols into two major sections (one being an appendix), separated by other sections, would force readers to need to repeatedly move back and forth between sections, degrading flow. However during the revision process, we will more fully consider this option.

10    We agree that moving a specialized section from the main body to an appendix is often a good way to to streamline a paper. We have made an attempt to do this, but the result was unsatisfactory. The problem is that the section that is mentioned (Protocols) is the main section of our paper. If all of it were moved to an appendix, only 3 or 4 pages of text would remain for the main body. If part of it were moved, the targeted readers (mostly ocean biogeochemical modelers) would need to move back and forth across many other sections many times when studying the protocols. Furthermore, we fear that relegating the
15    protocols to an Appendix might be taken as a sign by some readers that it is optional material that does not need close attention. Therefore we have kept the Protocols section in the main body of the paper.

Additionally, we would like to see some more detail on some of the issues raised above, notably;

20    a. More discussion on the science goals of the OMIP BGC in CMIP6 and what science gaps it is attempting to fill to be outlined in the introduction. You mention that OMIP BGC is focussed on the CMIP6 question on 'understanding systematic biases' but give no detail on what OMIP BGC is hoping to achieve that is new.

In the revised manuscript we will include more detail on the OMIP-BGC science goals as well as the gaps to be addressed.
25    We will then further address how OMIP-BGC aims to assess fundamental concerns about systematic biases.

In the revised manuscript, more information on the science goals, the gaps to be addressed, and new achievments are provided in the final three paragraphs of the Introduction. Two of them are new; the other has been revised.

30    b. All MIPs have been asked to demonstrate connectivity to the DECK experiments and the CMIP6 historical simulations as one of the 10 endorsement criteria (see Table 1 in Eyring et al., 2016). Please document this for OMIP BGC.

The connectivity to the CMIP6 historical and DECK experiments will be made clearer in the revised manuscript.

We have inserted a paragraph in the first part of the Protocols section, that describes the connectivity between the OMIP simulations and CMIP6 DECK and historical simulations.

c. You have not provided an analysis plan for the science community engaged in OMIP BGC. How are you going to use the experiments and diagnostics? Are you committing to analyse all the data that you are requesting (or can you point to other MIPs that will do so)?

The OMIP-BGC effort aims to provide a central forum to promote discussion, facilitate analysis, and prompt wide participation of the ocean biogeochemical modeling community in the related analysis effort. In this sense then, speaking for the community, the goal is indeed to analyze all of the model output requested. An analysis plan will be included in the revised manuscript. Other MIPs under the CMIP6 umbrella such as C4MIP will certainly take on some analysis of ocean output for which OMIP has provided diagnostics.

A short paragraph has been added at the end of the Diagnostics section in order to provide a glimpse of how we aim to proceed in order to promote analysis within the international community.

d. You describe observations of e.g. CFC-11, CFC-12, SF6 etc in the introduction that might be used for evaluation of the models. Are/Could any new observations be made easily available to the modelling community (e.g. through Obs4MIPs?)

Discrete and gridded observations of CFC-11, CFC-12, and $SF_6$ will be used extensively to evaluate the OMIP models. Whether we have the right, as a modeling community, to submit new observations through Obs4MIPs is an open question that we have not adequately considered. Certainly existing observations that are already available publicly could also be added through Obs4MIPS, assuming approval can be obtained from the data providers. It is an aim of OMIP-BGC to facilitate access to the relevant observational data that is used for model evaluation, as done previously during the Ocean Carbon Cycle Model Intercomparison Project.

We intend to explore these possibilities in the future, but for now our understanding is too preliminary concerning the possibilities and legal impediments to contributing to contributing to Obs4MIPs. Hence we have added nothing to the manuscript about this future possibility.

We hope you agree that some level of consistency across the MIP papers in this special issue is valuable and that the above suggestions can be accommodated in your paper.

Consistency across the contributions to the CMIP6 special issue in GMD is important, and we will do our part to help.

Other comments:

- For the diagnostic section (3 and tables 4-14), what is the link to the CMIP6 data request? Perhaps you need to clarify where is the definitive documentation of what is actually being output from the models (e.g. via a link to the actual data request) and to reference the GMD paper by Martin Jukes?

In the revised document, we plan to cite the GMD paper by Jukes and provide links to the CMIP6 data request, while assuring consistency with revisions to the Tables.

Unfortunately, we were unable to find any publication in GMD by Martin Juckes et al. However, we have indicated the link to the CMIP6 data request in the Diagnostics section of our revised manuscript.

Other comments: With many thanks for your ongoing efforts in the CMIP6 process.

The CMIP Panel

Your comments are much appreciated.

[revised manuscript text omitted]

* carbon concentration from the picophytoplankton ($<2$ $\mu$m) component alone

† carbon concentration from additional phytoplankton component alone

‡ carbon concentration from the microzooplankton ($<20$ $\mu$m) component alone

§ carbon concentration from mesozooplankton (20–200 $\mu$m) component alone

[revised manuscript text omitted]

‡ Vertically integrated $C_T$

§ Vertically integrated DOC (explicit pools only)

¶ Vertically integrated POC

**Table 11.** Monthly mean biogeochemical output: Priority 2 (3-D fields).

| Symbol | Variable name | Units | Shape |
|---|---|---|---|
| $C_T^{nat}$ | dissicnat | mol m⁻³ | XYZ |
| $pC_T^{abio}$ | dissicabio | mol m⁻³ | XYZ |
| $F_{CO_2}^{nat}-{}^{14}C_T^{abio}$ | dissi14cabio | mol m⁻³ | XYZ |
| $F_{CO_2}^{nat}-{}^{13}C_T$ | dissi13c | mol m⁻³ | XYZ |
| $F_{CO_2}^{abio}-A_T^{nat}$ | talknat | mol m⁻³ | XYZ |
| pH | ph | 1 | Surface Downward Flux of Abiotic |
| $F_{14CO_2}^{abio}-pH^{nat}$ | phnat | 1 | XYZ |
| pH$^{abio}$ | phabio | 1 | Surface Downward Flux of Abiotic |
| $F_{14CO_2}-O_2$ | o2 | mol m⁻³ | XYZ |
| $F_{O_2}$ | o2sat | mol m⁻³ | XYZ |

Monthly mean biogeochemical output: Priority 2. Symbol $NO_3^-$ | Variable name no3 | Units mol m⁻³ | Shape XYZ

[revised manuscript text omitted]

---

## Author Response (AR2)

**Author Response on Editor Decision on "Biogeochemical protocols and diagnostics for the CMIP6 Ocean Model Intercomparison Project (OMIP)"**

James C. Orr[1], Raymond G. Najjar[2], Olivier Aumont[3], Laurent Bopp[1], John L. Bullister[4], Gokhan Danabasoglu[5], Scott C. Doney[6], John P. Dunne[7], Jean-Claude Dutay[1], Heather Graven[8], Stephen M. Griffies[7], Jasmin G. John[7], Fortunat Joos[9], Ingeborg Levin[10], Keith Lindsay[5], Richard J. Matear[11], Galen A. McKinley[12], Anne Mouchet[13,14], Andreas Oschlies[15], Anastasia Romanou[16], Reiner Schlitzer[17], Alessandro Tagliabue[18], Toste Tanhua[15], and Andrew Yool[19]

[1]LSCE/IPSL, Laboratoire des Sciences du Climat et de l'Environnement, CEA-CNRS-UVSQ, Gif-sur-Yvette, France
[2]Dept. of Meteorology, Pennsylvania State University, University Park, Pennsylvania, USA
[3]Laboratoire d'Océanographie et de Climatologie: Expérimentation et Approches Numériques, IPSL, Paris, France
[4]Pacific Marine Environmental Laboratory, NOAA, Seattle, Washington, USA
[5]National Center for Atmospheric Research, Boulder, Colorado, USA
[6]Marine Chemistry & Geochemistry, Woods Hole Oceanographic Institution, USA
[7]NOAA Geophysical Fluid Dynamics Laboratory, Princeton, New Jersey, USA
[8]Dept. of Physics, Imperial College, London, UK
[9]Climate and Environmental Physics, Physics Inst. & Oeschger Center for Climate Change Res., Univ. of Bern, Switzerland
[10]Institut fuer Umweltphysik, Universitaet Heidelberg, Heidelberg, Germany
[11]CSIRO Oceans and Atmosphere, Hobart, Tasmania 7000, Australia
[12]Atmospheric and Oceanic Sciences, University of Wisconsin-Madison, Wisconsin, USA
[13]Max Planck Institute for Meteorology, Hamburg, Germany
[14]Astrophysics, Geophysics and Oceanography Department, University of Liege, Liege, Belgium
[15]GEOMAR Helmholtz Centre for Ocean Research Kiel, Kiel, Germany
[16]Columbia University and NASA-Goddard Institute for Space Studies, New York, NY, USA
[17]Alfred Wegener Institute, Bremerhaven, Germany
[18]Earth, Ocean and Ecological Sciences, University of Liverpool, Liverpool, UK
[19]National Oceanographic Centre, Southampton, UK

*Correspondence to:* James Orr (james.orr@lsce.ipsl.fr)

**Response to "Topical Editor Decision: Publish subject to technical corrections (07 Mar 2017)"**

We are grateful to the Editor for his careful handling of this manuscript and thank him for his final comments and of course for the positive decision. Those comments are provided below in gray, and our responses are given in black.

5    I have now completed my reading of your revised manuscript in the light of the two referees' comments and the short comments. I see that you have made very comprehensive replies to all of the comments. Thank you. Thank you also for the detailed and clear "Author's response". I very much appreciate the careful preparation of that document.

Our pleasure.

I am pleased to inform you that I accept your manuscript for publication subject to technical corrections.

Many thanks. Your comments have been very helpful to get this work in proper shape.

I have spotted a few minor things. In the following, the line numbers refer to the version of the manuscript from the "authors' response" file. The line numbers in the final manuscript are not usable from page 12 onwards – not sure what happened there (perhaps one extra LaTeXing would help?).

The line numbers should appear correctly in this response to your Final comments.

p. 5, l. 6: "earth system models" is capitalized everywhere else. Do not forget that you have defined "ESM" before ...

We now capitalize Earth System Models on its first use and afterwards we employ the abbreviation ESMs throughout the text.

p. 5, l. 13: the "2" in CO2 to be set as index

Fixed. The "2" has been made a subscript.

p. 6, l. 2: "initialilzed" should read "initialized"

This typo has been corrected.

p. 13, l. 29: please discard "in permil" as the salinity on the PSS-78 is a dimensionless number (see e.g., UNESCO Technical Paper in Marine Science 45, The International System of Units (SI) in Oceanography, 1985)

Done.

In the references (throughout): long dashes in DOIs make the DOIs invalid (they cannot be used in DOI resolvers such as http://dx.doi.org). Long dashes must be replaced by normal ones, and then the resolvers work fine.

Thanks for pointing out this important point, which we would have never noticed without you keen eyes. It has been corrected. We learned that we should always avoid putting a DOI number containing dashes in the "pages" field of a BibTeX entry,

because it converts a single hyphen into an en-dash, at least with the copernicus.bst.

   p. 29, l. 24: the "2" in CO2 to be set as index

5   Done.

   During the access review stage, you agreed to add the old OCMIP2 documents as supplementary material with the final revised manuscript. I have not found any supplement in the submitted document. Perhaps you have changed your mind. For readers convenience, I would still be pleased to have those reference documents bundled with this paper. An alternative would
10  be to add dedicated links to the OMIP-BGC web page, that will hopefully accept connections soon.

   Many thanks for this reminder. This supplement is included along with this response and the other files when loaded on the EGU/Copernicus page for submission of files for the producttion phase. Links will also be made to the online versions of these protocols on the OMIP-BGC web page.

   Your comments are greatly appreciated.

[revised manuscript text omitted]